# Endocycle-related tubular cell hypertrophy and progenitor proliferation recover renal function after acute kidney injury

Elena Lazzeri[1,2], Maria Lucia Angelotti[1,2], Anna Peired [1,2], Carolina Conte[1,2], Julian A. Marschner[3], Laura Maggi[2], Benedetta Mazzinghi[4], Duccio Lombardi [1,2], Maria Elena Melica[1], Sara Nardi[1,2], Elisa Ronconi[1,2], Alessandro Sisti[1,4], Giulia Antonelli[1,2], Francesca Becherucci[4], Letizia De Chiara[4], Ricardo Romero Guevara[1,2], Alexa Burger[5], Beat Schaefer[6], Francesco Annunziato[2], Hans-Joachim Anders[3], Laura Lasagni[1,2] & Paola Romagnani [1,2,4]

Acute kidney injury (AKI) is considered largely reversible based on the capacity of surviving tubular cells to dedifferentiate and replace lost cells via cell division. Here we show by tracking individual tubular cells in conditional Pax8/Confetti mice that kidney function is recovered after AKI despite substantial tubular cell loss. Cell cycle and ploidy analysis upon AKI in conditional Pax8/FUCCI2aR mice and human biopsies identify endocycle-mediated hypertrophy of tubular cells. By contrast, a small subset of Pax2+ tubular progenitors enriches via higher stress resistance and clonal expansion and regenerates necrotic tubule segments, a process that can be enhanced by suitable drugs. Thus, renal functional recovery upon AKI involves remnant tubular cell hypertrophy via endocycle and limited progenitor-driven regeneration that can be pharmacologically enhanced.

[1] Department of Clinical and Experimental Biomedical Sciences, University of Florence, Florence, Italy. [2] Excellence Centre for Research, Transfer and High Education for the development of DE NOVO Therapies (DENOTHE), Florence, Italy. [3] Division of Nephrology, Medizinische Klinik and Poliklinik IV, Klinikum der LMU München, Munich, Germany. [4] Nephrology Unit and Meyer Children's University Hospital, Florence, Italy. [5] Institute of Molecular Life Sciences, University of Zurich, Zurich, Switzerland. [6] Department of Oncology and Children's Research Center, University Children's Hospital, Zurich, Switzerland. These authors contributed equally: Elena Lazzeri, Maria Lucia Angelotti. Correspondence and requests for materials should be addressed to P.R. (email: paola.romagnani@unifi.it)

cute kidney injury (AKI) is a global health concern impacting ~13.3 million patients[1] and 1.7 million deaths per year[2,3]. AKI is defined by an acute deterioration of renal excretory function[1–3]. If not lethal in the acute phase, AKI is considered reversible as implied by recovery of urine production and biomarkers of renal function[3]. However, even mild AKI episodes imply a substantial risk for subsequent chronic kidney disease (CKD)[1], but the pathophysiological basis for this

phenomenon remains uncertain[4]. Indeed, the current patho-physiological concept involves the assumption that every tubular epithelial cell (TEC) surviving the injury phase has the potential to dedifferentiate and proliferate to replace lost cells or even re-epithelialize denuded tubule segments[5,6]. This concept has been evidenced by immunolabelling for cell cycle markers, such as Ki-67, proliferating cell nuclear antigen (PCNA) or 5-bromo-2′-deoxyuridine (BrdU) uptake[7]. As a second concept, tubule

regeneration may also involve a specific subset of TECs, referred to as tubular progenitors[8–10]. We set three hypotheses: (1) the overall capacity of tubular regeneration after injury is largely overestimated; (2) cell cycle markers may not consistently represent cell division; (3) regeneration is limited to tubular progenitors and other TECs entering the cell cycle after AKI undergo endocycle-related hypertrophy.

## Results

**Function recovery upon AKI masks a substantial TEC loss**. To evaluate TEC loss and regeneration after AKI, we applied a lineage tracing approach using conditional *Pax8.rtTA;TetO.Cre;R26.Confetti* (Pax8/Confetti) mice[11], enabling a doxycycline-induced random labeling of all TECs by permanent recombination of a single-color-encoding gene (red, yellow, green, or blue fluorescent proteins, RFP, YFP, GFP, and CFP; Supplementary Fig. 1a)[12]. Transient unilateral ischemia reperfusion injury (IRI) was then induced as detailed in Supplementary Fig. 1b, c. Tubular necrosis at day 2 was partially restored at day 30 and associated with some focal interstitial fibrosis (Supplementary Fig. 1d). Blood urea nitrogen (BUN) was unchanged, even if at day 30 a significant loss-of-kidney weight had occurred (Supplementary Fig. 1e, f). Since BUN was too insensitive to detect the decline of kidney function, we directly measured glomerular filtration rate (GFR). GFR strongly declined at day 1 and partially recovered at day 14 remaining stable thereafter indicating CKD after AKI (Fig. 1a). Lineage tracing up to day 30 showed the presence of single-colored clones in outer stripe of the outer medulla (OSOM) (Fig. 1b, c). Therefore, all further analyses focused on this area. Quantitative analysis revealed a substantial and sustained loss-of-30.5 ± 2.8% of total Confetti-labelled TECs (Fig. 1d). Similar results were obtained when TEC loss was evaluated after immunostaining for aquaporin-2 (AQP2) to exclude from the count collecting ducts (23.8 ± 5.9%; Fig. 1d), or for aquaporin-1 (AQP1), to limit the analysis to proximal TECs up to the thin descending limb of the Henle's loop (32.5 ± 7.1%; Fig. 1d and Supplementary Fig. 1g–i). No transgene leakage was observed in healthy or ischemic mice (Supplementary Fig. 1j, k). Similar data were obtained in glycerol-induced AKI, a model of toxic tubule necrosis, either when we quantified total Confetti or AQP2− Confetti TECs (Fig. 1e–h). Thus, function recovery upon AKI masks a substantial and sustained TEC loss.

**A small TEC subset proliferates after AKI**. To test the current dogma of kidney regeneration via mitotic cell division of surviving TECs, we quantified their progeny. Administration of a lower doxycycline dose to label few TECs enabled precise clone counts at day 30 after IRI (Supplementary Fig. 1l, m). Healthy mice (T0), mice that underwent a prolonged washout for 30 days (age-matched controls, T30) and sham-operated mice presented a similar clone size frequency (Fig. 1j; NS, Eq. 1 in Methods). Of note, the percentage of single cell clones decreased from 92.4 ± 0.9% at T0 to 78.9 ± 0.9% at day 30 after IRI. By contrast, clones consisting of 2 or more cells increased from 7.4 ± 0.9% at T0 (6.4 ± 0.7 of doublets and 1 ± 0.2% of triplets) to 21.1 ± 0.9% at T30 after IRI (11.9 ± 1.5% of doublets, 4.3 ± 0.2% of triplets and 4.9 ± 1.4% ≥4 cells, with single colored clones of up to 11 cells; Fig. 1i, j, Supplementary Fig. 2, Eq. 1 in Methods). Newly generated cells, had replaced only 54.1 ± 10.1% of lost AQP2− TECs (Fig. 1k, Eq. 3 in Methods). The clones observed at day 30 after IRI represented 21.1 ± 0.9% of those AQP2− survived, but were the progeny of only 8.6 ± 1.6% of AQP2− TECs present at T0 (Fig. 1l, Supplementary Fig. 2, Eq. 2 in Methods). Similar results were obtained in the nephrotoxic AKI model, where 35.4 ± 5.7% of lost cells were replaced in each kidney (Fig. 1k, Eq. 3 in Methods) and ultimately derived from 3.5 ± 0.1% of AQP2− TECs present at T0 (Fig. 1l, Eq. 2 in Methods). Together, these results show that upon AKI only a small TEC subset undergoes mitosis to replace no more than ~50% of injured cells.

**Kidney tubules contain a distinct Pax2+ cell subset**. Using conditional *Pax2.rtTA;TetO.Cre;R26.Confetti* (Pax2/Confetti) mice (Supplementary Fig. 3a), we recently identified Pax2+ cells of the Bowman's capsule as progenitors regenerating podocytes upon glomerular injury[13]. These mice exhibited no leakage and Pax2 promoter fidelity as showed in Supplementary Fig. 3b–e and already previously reported[14,15]. Pax2+ cells were also found in a scattered pattern within tubules (Fig. 2a–h) along specific segments of the nephron (Fig. 2d). In particular, they represented 1.6 ± 0.5% of megalin+ TECs in S1 and S2 segments (Fig. 2h), 9.8 ± 0.9% of AQP1+ TECs in S3 segment (Fig. 2a, e) and 12.3 ± 1.2% of Tamm–Horsfall Protein+ (THP+) distal TECs (Fig. 2b, f).

Induction of cell labeling followed by 30 days washout or continued doxycycline exposure for the same period of time revealed stable numbers of Pax2 cells (Fig. 2i–l). In addition, induction of cell labeling at 12 weeks of age (Supplementary Fig. 3f) revealed a number of Pax2+ cells similar to that observed in mice induced at 5 weeks of age (68.8 ± 4.9 vs. 62 ± 1.2; NS). Thus, Pax2+ TECs are a distinct and stable cell population in mice.

**Pax2+ cells show increased survival and clonogenicity after AKI**. To see if and how the Pax2+ TEC subset contribute to tubule regeneration upon AKI, injured kidneys were subjected to lineage tracing over 30 days after IRI and nephrotoxic injury

**Fig. 1** Only a small TEC subset proliferates after AKI and partially replaces lost TECs. **a** GFR in ischemic mice (n = 13) normalized on the GFR at baseline and on sham-operated control group (n = 5). One-way ANOVA post hoc Tukey. **b, c** Juxtaposed images of a Pax8/Confetti mouse kidney at day 0 (T0, n = 5) (**b**) and 30 after IRI (IRI T30, n = 4) (**c**). Arrows indicate single-colored clones. OSOM outer stripe of outer medulla; ISOM inner stripe of outer medulla. **d** Number of Pax8+, Pax8+AQP2−, and Pax8+AQP1+ cells in Pax8/Confetti mice at day 0 (T0, white column, n = 5) and at day 30 after IRI (IRI T30, gray column, n = 4). Mann–Whitney test. *p < 0.05, **p < 0.01 IRI T30 vs. T0. **e** BUN in healthy Pax8/Confetti mice (n = 5) and in Pax8/Confetti mice after nephrotoxic AKI (n = 6). One-way ANOVA post hoc Tukey *p < 0.05 glycerol-treated mice vs. healthy. **f** GFR in Pax8/Confetti mice after nephrotoxic AKI (n = 7) normalized on the GFR at baseline and on healthy mice (n = 5). One-way ANOVA post hoc Tukey. **g** Number of Pax8+ cells in Pax8/Confetti mice at day 0 (T0, white column, n = 5) and day 30 after nephrotoxic AKI (Gly T30, gray with sparse pattern column, n = 4). Mann–Whitney test *p < 0.05 Gly T30 vs. T0. **h** Number of Pax8+AQP2− cells in Pax8/Confetti mice at day 0 (T0, white column, n = 5) and day 30 after nephrotoxic AKI (Gly T30, gray with sparse pattern column, n = 4). Mann–Whitney test *p < 0.05 Gly T30 vs. T0. **i** Single-colored clones in AQP2− tubules in Pax8/Confetti mice at day 30 after IRI (n = 4). AQP2 staining is white. **j** Clone frequency analysis of Pax8+AQP2− cells in Pax8/Confetti mice at day 0 (T0, n = 5), in age-matched controls (T30, n = 4), in sham-operated mice (n = 4), at day 30 after IRI (IRI T30, n = 4) and at day 30 after nephrotoxic AKI (Gly T30, n = 4). Mann–Whitney test *p < 0.05 IRI T30 vs. T0, T30, sham and Gly T30 vs. T0 and T30. **k** Percentage of Pax8+AQP2− cells in Pax8/Confetti mice at day 0 (T0, n = 5), at day 30 after IRI (IRI T30, n = 4) and at day 30 after nephrotoxic AKI (Gly T30, n = 4). Mann–Whitney test *p < 0.05 IRI T30 and Gly T30 vs. T0. **l** Percentage of Pax8+AQP2− clonogenic cells (gray column) in Pax8/Confetti mice at day 30 after IRI (IRI T30, n = 4) and at day 30 after nephrotoxic AKI (Gly T30, n = 4) vs. day 0 (n = n = 5). The white column is the percentage of Pax8+AQP2− that did not generate clones. Data are mean ± SEM. Scale bars 40 μm. Pax8+ = Pax8 lineage-positive cells

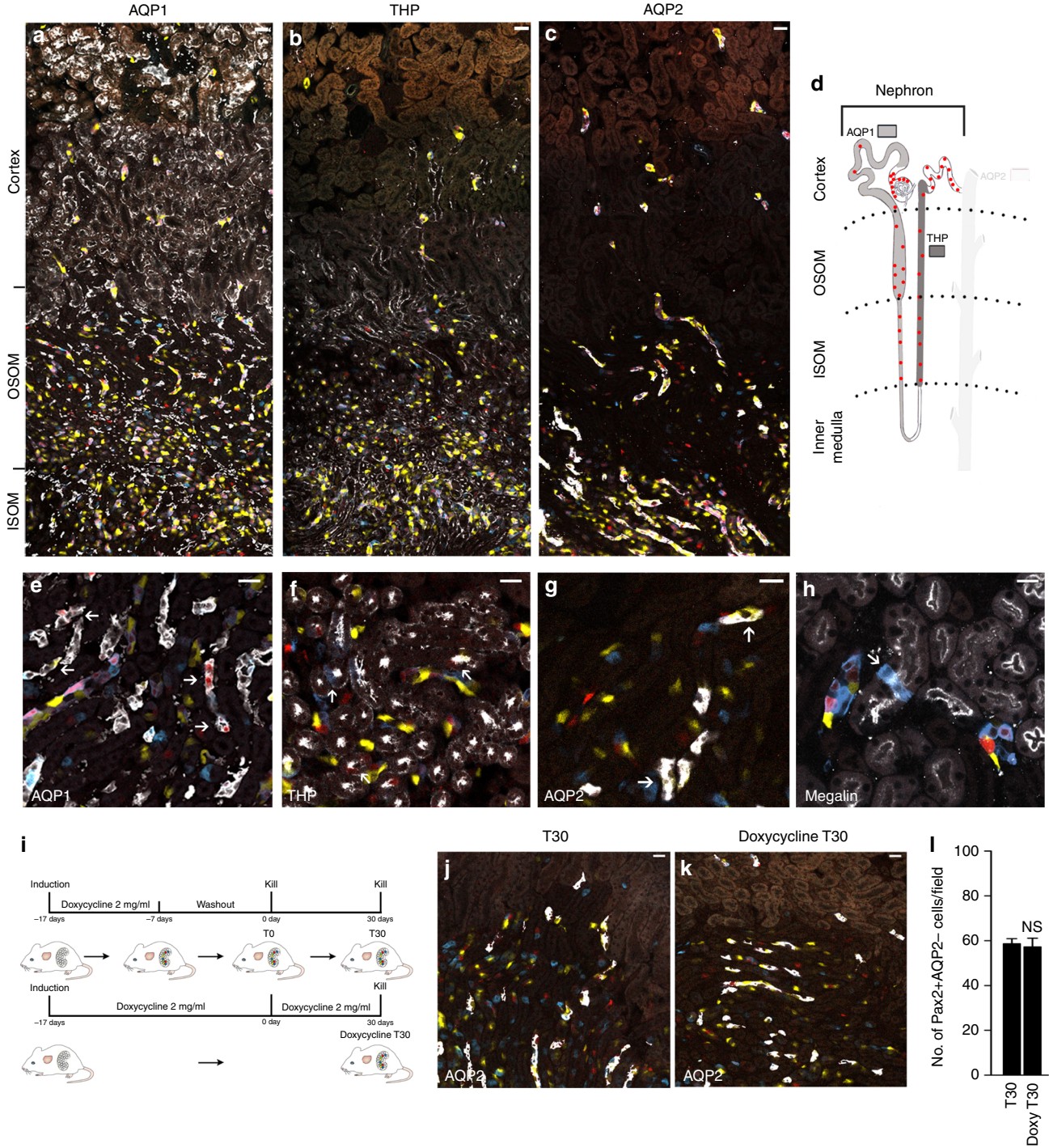

**Fig. 2** Kidney tubules contain a distinct, predefined Pax2 lineage-positive tubular cell subset. **a–c** Juxtaposed confocal images of a kidney section from cortex to inner stripe of outer medulla in adult Pax2/Confetti mice ($n = 4$). Confetti reporter shows Pax2 lineage-positive single cells scattered within the proximal tubules, distal tubules and collecting ducts as demonstrated by immunolabelling for AQP1 (**a**), THP (**b**) and AQP2 (**c**), respectively (white). OSOM outer stripe of outer medulla; ISOM inner stripe of outer medulla. Scale bars 40 μm. **d** Schematic localization of Pax2− lineage-positive cells (red dots) in the proximal tubule, stained by AQP1 (light gray), in the thick ascending limb and in the distal tubule, stained by THP (dark gray), of the nephron. **e–h** Representative images of a kidney section in healthy Pax2/Confetti mice showing ($n = 4$) the distribution of Pax2 lineage-positive cells in S3 segment of proximal tubules (AQP1+, white) (**e**), in thick ascending limbs and distal tubules (THP+, white) (**f**), in collecting ducts (AQP2+, white) (**g**) and in S1–S2 segment of proximal tubules (Megalin+, white) (**h**). Scale bars 20 μm. **i** Experimental schemes. **j**, **k** Representative images of a kidney section in the OSOM of Pax2/Confetti mice showing Pax2+AQP2− cells after 30 days of washout (T30, $n = 5$) (**j**) and after 30 days with doxycycline (doxycycline T30, $n = 4$) (**k**). AQP2 staining is white. **l** Number of Pax2+AQP2− cells in OSOM of Pax2/Confetti mice at T30 ($n = 5$) and after 30 days with doxycycline (doxy T30, $n = 4$). Mann–Whitney test NS. Data are mean ± SEM. Scale bars 20 μm. Pax2+ = Pax2 lineage-positive cells

(Supplementary Fig. 4a). BUN and GFR measurements gave similar results to those obtained in Pax8/Confetti mice (Supplementary Fig. 4b and Fig. 3a). This study revealed single-colored clones (Fig. 3b–d) up to 10 cells within S3 segments resulting in an increase number of Pax2+AQP2− TECs per field (Fig. 3e; $p < 0.05$) after IRI. Clone size frequency analysis at T0, T30, and in sham-operated mice showed similar percentages of clones (Fig. 3f; NS, Eq. 1 in Methods). Of note, the percentage of single cell clones at T0 decreased from $97.7 \pm 0.2\%$ to $76.4 \pm 1.8\%$ at day 30 after IRI (Fig. 3f, Eq. 1 in Methods). By contrast, clones consisting of 2 or more cells increased from $2.2 \pm 0.2\%$ (only doublets) to $23.6 \pm 1.8\%$ ($15.2 \pm 0.6\%$ of doublets, $5.1 \pm 0.8\%$ of triplets and $3.3 \pm 0.7\%$ ≥4 cells, with single colored clones of up to 10 cells; Fig. 3f, Eq. 1 in Methods section). Accordingly, 30 days after IRI newly generated Pax2+AQP2− TECs, appeared and significantly increased the total number of Pax2+AQP2− TECs

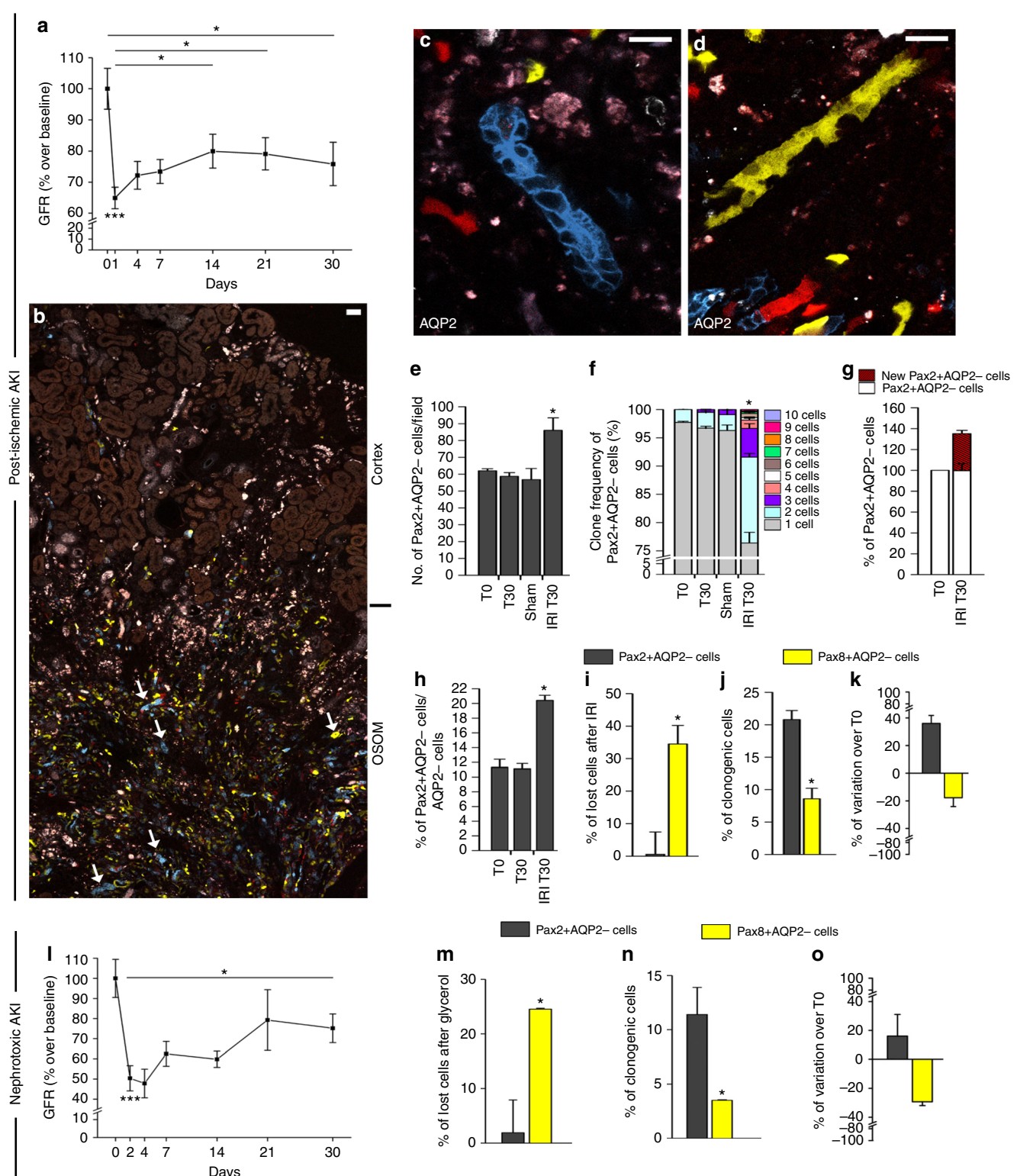

in comparison to T0 (Fig. 3g, Eq. 3 in Methods). Following IRI and expansion of Pax2+ cells, Pax2 immunolabelling was present also on many TECs not expressing the Confetti reporter, implying TECs could acquire Pax2 expression upon injury (Supplementary Fig. 4c). Consistently, many Pax2− immunolabelled cells in Pax8/Confetti mice were not found clonogenic (Supplementary Fig. 4d).

The enrichment of Pax2+ TECs labeled by the Confetti reporter (Fig. 3h) was related to both an increased survival capacity (Fig. 3i, Eq. 4 in Methods), as well as to a higher clonogenic capacity (Fig. 3j, Eq. 2 in Methods) in comparison to other TECs. Indeed, while Pax8+ cells were significantly reduced, Pax2+ cells were expanded in comparison to day 0 (Fig. 3k, Eq. 5 in Methods). Similar results were obtained in nephrotoxic AKI (Fig. 3l–o and Supplementary Fig. 4e, Eqs. 4, 2 and 5 in Methods). No transgene leakage was observed in Pax2/Confetti ischemic mice (Supplementary Fig. 4f). These results demonstrate that during AKI Pax2+ TECs display increased survival and capacity to undergo mitosis in comparison to other TECs.

**Pax2+ cells regenerate long tubule segments**. We then checked the distribution of Pax2+ clones in the different tubule segments after immunolabelling for megalin, AQP1, THP, and AQP2 (Fig. 4a–d). The great majority of Pax2-derived clones formed in the S3 segment and less in the TAL of Henle's loop (Fig. 4e, Eq. 7 in Methods), the two tubule segments of the outer medulla where tubule necrosis occurs. By contrast, only few clones formed in the S1–S2 segments in the cortex or in collecting ducts (Fig. 4e, Eq. 7 in Methods). The percentage of clonogenic cells was not significantly different between a 2D and 3D analysis performed in 40 μm thick sections (Fig. 4f, Eq. 2 in Methods). However, the 2D analysis slightly overestimated the percentage of clonogenic cells due to long clones being mistakenly counted as several smaller ones (Fig. 4f). Indeed, 2D assessment allowed identification of clones of up to 10 cells, while 3D analysis revealed single colored Pax2+ clones up to 98 cells regenerating long tubule segments (Fig. 4g–i and Supplementary Movie 1). These results demonstrate that pre-existing populations of Pax2+ cells regenerate necrotic tubule segments in the injured outer medulla.

**Drugs stimulating regeneration after AKI act selectively on Pax2+ TECs**. We then evaluated if proliferation is limited to Pax2+ cells or involves also other TECs. To this aim, we treated Pax2/Confetti mice with drugs that were previously reported to increase TEC proliferation and to improve tissue regeneration following AKI[16–19], i.e., the histone deacetylase inhibitors (HDACi) trichostatin (TSA) and 4-phenylbutyrate (4-PBA) (Supplementary Fig. 5a–d). Treatment starting 24 h after IRI resulted in a sustained recovery of GFR in comparison to mice treated with vehicle, which remained with persistently impaired GFR, i.e., CKD (Fig. 5a, b). A better reconstitution of tubular integrity was also observed (Fig. 5c, Supplementary Fig. 5e–g), consistent with previous results[16–19]. TSA and 4-PBA both increased TEC number in comparison to vehicle. Of note, $86.4 \pm 4.5\%$ and $64.7 \pm 5.5\%$ of TEC loss were restored in TSA- and 4-PBA- treated mice, respectively (Fig. 5d).

Strikingly, the increased TEC number was related to a selective expansion of Pax2+ TECs (Fig. 5e, Eq. 6 in Methods), that proliferated extensively, as demonstrated by the significant increase of Pax2+ cells included in clones (Fig. 5f). By contrast, the number of other TECs was unchanged (Fig. 5e). Representative images are shown (Fig. 5g–k). These results show that HDAC inhibitors, that stimulate tubular regeneration after AKI, act selectively on Pax2+ cells.

**Cell cycle markers do not predict cell division after AKI**. The finding that only Pax2+ TECs divide upon IRI was in conflict with immunolabelling for Ki-67 (expressed from late G1 to mitosis[7], Fig. 6a–d), or PCNA, (expressed from late G1 to early G2[20], Fig. 6e–h). Indeed, we found diffuse tubular Ki-67 (Fig. 6b, d), as well as PCNA (Fig. 6f, h) positivity 2 days after injury suggesting many more TECs proliferate than actually proven by clone analysis in Pax8/Confetti mice. To clarify this apparent inconsistency we crossed *Pax8*-, as well as *Pax2.rtTA;TetO.Cre;* mice with mice harboring the fluorescent ubiquitin-based cell cycle indicator (FUCCI2) Cre-dependent reporter (Supplementary Fig. 6a, b), which consists of two fluorescent proteins whose expression alternates based on cell cycle phase: mCherry-hCdt1 (red color), expressed in G1, and mVenus-hGem (green color), expressed in S/G2/M[21] (Supplementary Fig. 6c). Cells can also appear as yellow at the G1/S boundary.[22] Experimental designs are detailed in Supplementary Fig. 6d, e.

Based on Pax8/FUCCI2 mice evaluated by confocal microscopy (Fig. 6i–k and Supplementary Fig. 6f–h) few TECs appeared in S/G2/M (mVenus+) at day 2 after IRI despite widespread tubular Ki-67 positivity ($13.2 \pm 3.4\%$ vs. $47.1 \pm 9.2\%$; $p < 0.05$ Mann–Whitney test; Fig. 6j, l vs. Fig. 6b, d). Interestingly, when Pax2/FUCCI2 mice were evaluated by confocal microscopy (Fig. 6m–o and Supplementary Fig. 6i–k), the percentage of cycling cells over the total of Pax2+ FUCCI2+ cells was higher ($40.6 \pm 1.7\%$; Fig. 6n, p), but the number per field was comparable to that of the Pax8/FUCCI mice ($10.7 \pm 0.4$ in Pax2/FUCCI2 mice

**Fig. 3** Pax2 lineage-positive cells show increased survival and proliferative capacity in comparison to other TECs after AKI. **a** GFR in Pax2/Confetti mice after ischemic AKI ($n = 9$) normalized on the GFR at baseline and on sham-operated control group ($n = 5$). One-way ANOVA post hoc Tukey. **b** Juxtaposed confocal images of a kidney section in Pax2/Confetti mice at day 30 after IRI ($n = 5$). Arrows indicate single colored clones. Scale bars 40 μm. **c**, **d** Single colored clones in AQP2− tubules in Pax2/Confetti mice at day 30 after IRI ($n = 5$). AQP2 staining is white. Scale bars 20 μm. **e** Number of Pax2 +AQP2− TECs in Pax2/Confetti mice at day 0 (T0, $n = 4$), in age-matched controls (T30; $n = 5$), in sham-operated mice (sham; $n = 5$) and at day 30 after IRI (IRI T30, $n = 5$). Mann–Whitney test *$p < 0.05$ IRI T30 vs. T0, T30 and sham. **f** Clone frequency analysis of Pax2+AQP2− cells in Pax2/Confetti mice at day 0 (T0, $n = 4$), in age-matched controls (T30, $n = 5$), in sham-operated mice (sham; $n = 5$) and at day 30 after IRI (IRI T30, $n = 5$). Mann–Whitney test *$p < 0.05$ IRI T30 vs. T0, T30, and sham. **g** Percentage of Pax2+AQP2− cells in Pax2/Confetti mice at day 0 (T0, $n = 4$) and 30 after IRI (IRI T30, $n = 5$). **h** Percentage of Pax2+AQP2− TECs vs. AQP2− TECs in Pax2/Confetti mice at day 0 (T0, $n = 4$), in age-matched controls (T30, $n = 5$), and at day 30 after IRI (IRI T30, $n = 5$). Mann–Whitney test *$p < 0.05$ IRI T30 vs. T0 and T30. **i** Percentage of lost Pax2+AQP2− TECs ($n = 5$) and Pax8+AQP2− TECs ($n = 5$) at IRI T30 vs. T0. Mann–Whitney test *$p < 0.05$. **j** Percentage of Pax2+AQP2− ($n = 5$) and Pax8+AQP2− TECs ($n = 5$) that generated clones at IRI T30 vs. T0. Mann–Whitney test *$p < 0.05$. **k** Percentage of Pax2+AQP2− ($n = 5$) and Pax8+AQP2− TECs ($n = 5$) at IRI T30 vs. T0. **l** GFR in Pax2/Confetti mice after nephrotoxic AKI ($n = 8$) normalized on the GFR at baseline and on healthy mice ($n = 5$). One-way ANOVA post-hoc Tukey. **m** Percentage of lost Pax2+AQP2− TECs ($n = 6$) in Pax2/Confetti and of lost Pax8+AQP2− ($n = 4$) in Pax8/Confetti mice at day 30 after nephrotoxic AKI vs. T0. Mann–Whitney test *$p < 0.05$. **n** Percentage of Pax2+AQP2− ($n = 6$) and Pax8+AQP2− TECs ($n = 4$) that generated clones at day 30 after nephrotoxic AKI vs. T0. Mann–Whitney test *$p < 0.05$. **o** Percentage of Pax2+AQP2− ($n = 6$) and Pax8+AQP2− TECs ($n = 4$) at day 30 after nephrotoxic AKI vs. T0. Data are mean ± SEM. Pax2+ = Pax2 lineage-positive cells, Pax8+ = Pax8 lineage-positive cells. For calculation of figures **i–k**, **m–o** see Methods section and representative calculations in Pax2/Confetti mice in Supplementary Methods

vs. 10.9 ± 0.3 in Pax8/FUCCI2 mice, NS) (Fig. 6n vs. Fig. 6j). These results suggested that numerous TECs entered the cell cycle but only few progressed toward the G2/M phase, i.e., mitosis. Both in Pax8/FUCCI2 and Pax2/FUCCI2 mice TECs were selectively labeled, as shown in Fig. 6i–k and Fig. 6m–o with Phalloidin staining and as previously reported[23,24]. No transgene leakage was observed in healthy or ischemic mice (Supplementary Fig. 6l–q). We thus performed immunolabelling for a marker commonly considered to label only cells in the G2/M phase,

serine 10-phosphorylated histone H3 (p-H3)[25–27]. As expected, a lower percentage of TECs at day 2 was p-H3+ in comparison to Ki-67 (11.9 ± 1.1% vs. 47.1 ± 9.2%, $p < 0.05$, Supplementary Fig. 6r and Fig. 6d). P-H3+ TECs were mostly mVenus+, confirming that they were in G2/M (Fig. 6q, s). However, unexpectedly, also some Pax8+ TECs in G1 (mCherry+) were labeled (Fig. 6q). At day 30, mCherry+p-H3+ cells strongly increased, representing about a half of FUCCI2+p-H3+ TECs (Fig. 6r, s). By contrast, in Pax2/FUCCI2 mice, p-H3 mostly co-labeled mVenus-expressing

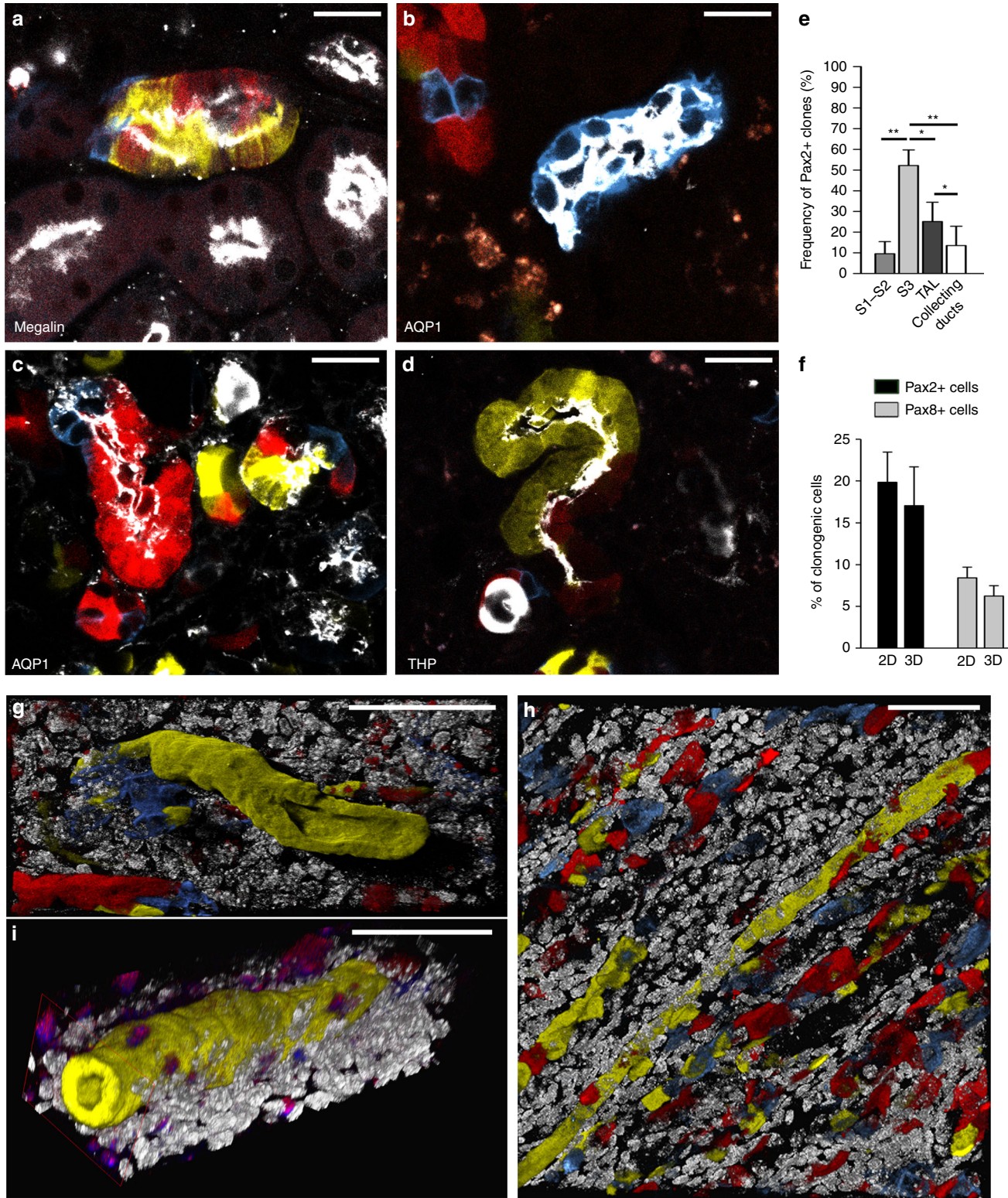

cells (Fig. 6t–v). This implies that Pax2+p-H3+ cells were truly in G2/M phase. These results document that Ki-67, PCNA, and p-H3 do not reliably indicate TEC proliferation after AKI.

**Pax8+ TECs endocycle while Pax2+ cells complete mitosis.** The aforementioned data, together with the results obtained with the Confetti models, consistently suggested that most TECs might undergo alternative cell cycles without cell division. Indeed such alternative cycles, e.g., endoreplication cycles or endocycles, have recently been reported to drive polyploidy and hypertrophy upon tissue injury[28]. To evaluate this possibility in AKI, we combined DNA content analysis with detection of FUCCI2 fluorescence by flow cytometry. Indeed, using this approach, G2/M diploid cells, which express mVenus and represent truly cycling cells can be distinguished from G1 polyploid cells which express mCherry and instead represent mononucleated endoreplicating cells (Fig. 6w–z). Pax8/FUCCI2 mice 2 days after IRI showed 9.8 ± 0.7% of Pax8+ TECs in S phase, but only 2 ± 0.5% in G2/M phase (Fig. 6w, y). Lineage tracing of mice for 30 days showed a considerable increase of Pax8+ TECs cells that expressed mCherry but displayed a polyploid DNA content (≥4C) (2.1 ± 0.1% at day 2 vs. 13.5 ± 1.4% at day 30, p < 0.05, Mann–Whitney test; Fig. 6w, y). By contrast, cells with an abnormal DNA content were not observed in Pax2/FUCCI2 mice, either at 2 or 30 days (Fig. 6x, z). Rather, 2 days after IRI 35 ± 4.8% of Pax2+ cells were in the S phase, suggesting a higher proliferation capacity than Pax8+ cells (Fig. 6y, z). The gating strategy used to analyse FUCCI2+ cells and to avoid counting cell doublets as a potential source of artifacts is shown in Supplementary Fig. 7. Given our extremely conservative gating strategy, we certainly underestimated the percentage of endocycling cells. Similar results were obtained after deleting AQP2+ cells from the analysis (Supplementary Fig. 7b–e). These results show that Pax2+ cells frequently and efficiently divide, while other TECs endocycle into polyploid mononuclear TECs.

**Pax2+ cells proliferate while other TECs undergo hypertrophy.** To validate these results, we used two further transgenic mouse lines, based on the FUCCI2aR reporter, a recent advancement of the FUCCI2 which produces iso-stoichiometric quantities of both FUCCI probes expressed during all the cell cycle, with higher sensitivity and labeling of the cells during all the cell cycle phases[29].

The generation of *Pax8.rtTA;TetO.Cre;FUCCI2aR*, as well as *Pax2.rtTA;TetO.Cre;FUCCI2aR* transgenic mouse lines, experimental design and controls are detailed in Methods and in Supplementary Fig. 8. Both transgenic lines showed a percentage of induction higher than 90%.

The experimental procedure is summarized in Fig. 7a. Total renal cell suspensions were analysed by MacsQuant flow cytometry at time 0, day 2 and day 30 after IRI. As expected, the percentage of Pax8/FUCCI2aR-labeled tubular cells over total renal cells was about 10 times higher than that of Pax2/FUCCI2aR cells (Fig. 7b–j and Supplementary Fig. 9). However, this was only related to a higher percentage of mCherry+ cells, as the percentage of mVenus+ cells was similar (Fig. 7b–j and Supplementary Fig. 9), suggesting that virtually only Pax2+ cells divide after AKI. The MacsQuant flow cytometer also automatically quantified the number of cells/μl of the different populations contained in the total renal cell suspensions. Counts confirmed the massive TEC loss in Pax8/FUCCI2aR mice 2 days after IRI, that was only partially recovered after 30 days as found before (Fig. 7k). Interestingly, the number of newly generated cells at day 30 in comparison to day 2 in Pax8/FUCCI2aR and Pax2/FUCCI2aR mice was similar (Fig. 7l). Pax2/FUCCI2aR cell numbers at day 30 were significantly expanded in comparison to day 2 (Fig. 7k), passing from 12.6 ± 1.4% of Pax8/FUCCI2aR total cells at day 0 to 19.5 ± 2.2% at day 2 and to 28.9 ± 3.4% at day 30 (Fig. 7m). Strikingly, absolute counts showed that the number of proliferating cells (mVenus+ and mCherry+mVenus+) in Pax8/FUCCI2aR and in Pax2/FUCCI2aR mice was virtually identical at all time points analysed (Fig. 7n), demonstrating that only Pax2+ cells divide after AKI. Confocal microscopy showed expression of KIM-1 protein at day 30 by 71.7 ± 4.7% of Pax8+mCherry+ TECs, vs. only 2 ± 1.2% of Pax2+mCherry+ TECs, further underlining the different survival capacity of Pax2+ cells in comparison to other TECs (Supplementary Fig. 8i, m).

We then analysed the occurrence and distribution of endocycle. Consistent with results obtained in FUCCI2 mice, endocycling cells were only observed in Pax8/FUCCI2aR mice (Fig. 7o–q). In addition, 27.9 ± 4.6% of LTA+ proximal tubular cells in S1–S2 segments were p-H3+ mCherry+, representing endocycling cells, in comparison to 1.8 ± 0.8 of LTA+ proximal tubular cells that were mVenus+, the only truly dividing cells (Fig. 7r–t). Immunolabelling for AQP2 showed that the great majority of endocycling cells localized in the cortex (22.3 ± 3.4% in the cortex vs. 7.8 ± 2.2% in the OSOM), while the majority of truly proliferating cells localized in the OSOM (9.2 ± 1.6% in the OSOM vs. 3.1 ± 1.1% in the cortex, Fig. 7u). Endocycling cells in G1 (Pax8+mCherry+p-H3+TECs) displayed an increased size in comparison to cycling cells in G1 (Pax8+mCherry+p-H3− TECs) as confirmed by comparing their cell surface area by confocal microscopy (Fig. 7v, w).

Taken together, Pax2+ cells in the OSOM are the cells that regenerate injured tubules after AKI by mitotic cell division, while other TECs of the S1 and S2 segments rather enter endocycles to undergo hypertrophy. Both mechanisms may compensate for the irreversible loss-of-TECs occurring during AKI.

**Endocycle is detected in kidneys of patients with CKD after AKI.** To verify the presence of endocycling cells in human, we examined kidney biopsies from 10 cases of CKD after AKI and 5 healthy controls (Table 1). To detect endocycling cells in human, we applied the same strategy previously validated in the mouse, and searched for p-H3+ cells in G1 phase, the latter identified by

**Fig. 4** Pax2 lineage-positive cells regenerate long tubule segments. **a** Representative image of a kidney section showing single-colored clones in S1–S2 segments of proximal tubule as demonstrated by staining with anti-megalin antibody (white) in Pax2/Confetti mice at day 30 after IRI (n = 5). Scale bar 20 μm. **b, c** Representative images of a kidney section showing single-colored clones in S3 segment of proximal tubule as demonstrated by staining with anti-AQP1 antibody (white) in OSOM of Pax2/Confetti mice at day 30 after IRI (n = 5). Scale bars 20 μm. **d** Representative image of a kidney section showing single-colored clones in thick ascending limb as demonstrated by staining with anti-THP antibody (white) in Pax2/Confetti mice at day 30 after IRI (n = 5). Scale bar 20 μm. **e** Frequency of Pax2+ clones in S1–S2 segment of proximal tubules (Pax2+ Megalin+ clones, n = 5), in S3 segment of proximal tubules (Pax2+AQP1+ clones, n = 7), in thick ascending limbs (TAL, Pax2+THP+ clones, n = 7) and in the collecting ducts (Pax2+AQP2+ clones, n = 7) at day 30 after IRI. Mann–Whitney test. **p < 0.01 S3 vs. S1–S2 and collecting ducts, *p < 0.05 S3 vs. TAL and TAL vs. collecting ducts. **f** Percentage of clonogenic cells per field in 2D vs. 3D analysis in Pax2/Confetti and in Pax8/Confetti mice at day 30 after IRI vs. T0. (n = 5 Pax2/Confetti mice and n = 5 Pax8/Confetti mice). Mann–Whitney test NS. **g–i** 3D reconstruction of single-colored clones in Pax2/Confetti mice at day 30 after IRI (n = 5). DAPI counterstains nuclei (white). Scale bars 50 μm. Data are mean ± SEM. Pax2+ = Pax2 lineage-positive cells, Pax8+ = Pax8 lineage-positive cells

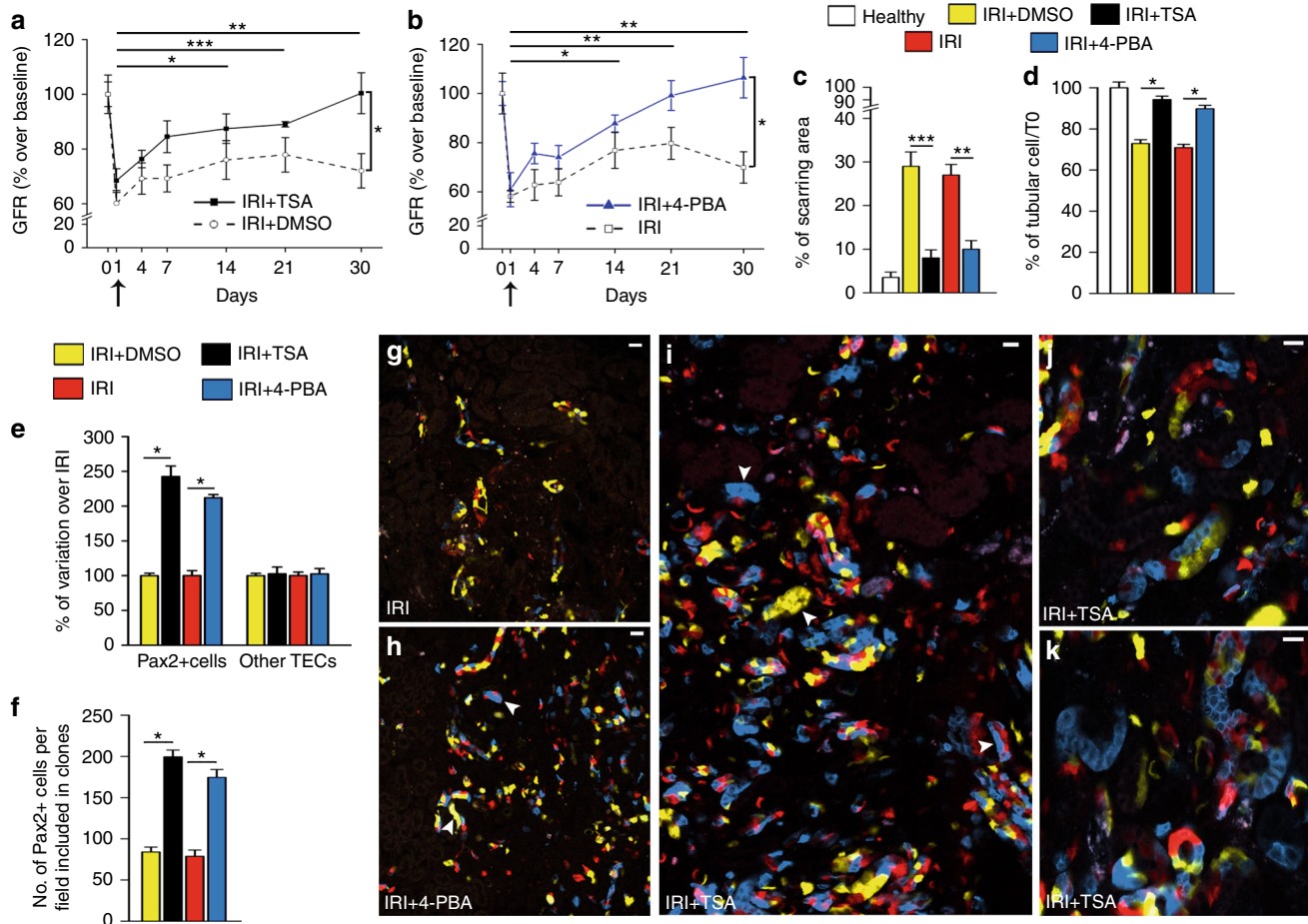

**Fig. 5** Drugs that promote tubular regeneration after AKI selectively enhance Pax2 lineage-positive cell mitosis. **a, b** GFR after IRI ($n = 6$), IRI + DMSO ($n = 7$), TSA ($n = 7$) or 4-PBA treatment ($n = 6$). One-way ANOVA post hoc Tukey. Arrows indicate starting of treatment. **c** Percentage of kidney area with tubular injury in kidney sections of Pax2/Confetti mice at day 0 (healthy, $n = 4$), at day 30 after IRI + DMSO ($n = 4$) or IRI ($n = 6$) and at day 30 after IRI + TSA ($n = 4$) or 4-PBA ($n = 6$) treatment ($n = 4$ images per mouse). Mann–Whitney test. \*\*\*$p < 0.001$ IRI + TSA vs. IRI + DMSO, \*\*$p < 0.01$ IRI vs. IRI + 4-PBA. **d** Percentage of TECs in Pax2/Confetti mice in healthy ($n = 4$), at day 30 after IRI + DMSO ($n = 4$) or IRI ($n = 5$) and at day 30 after IRI + TSA or 4-PBA treatment ($n = 4$ per each group) vs. T0. Mann–Whitney test. \*$p < 0.05$ IRI + TSA vs. IRI + DMSO and IRI vs. IRI + 4-PBA. **e** Percentage of variation of Pax2+ and Pax2− TECs in Pax2/Confetti mice at day 30 after IRI + TSA or 4-PBA treatment ($n = 4$ per each group) vs. IRI + DMSO ($n = 4$) or IRI ($n = 5$). Mann–Whitney test. \*$p < 0.05$ IRI + TSA vs. IRI + DMSO and IRI vs. IRI + 4-PBA. **f** Number of cells included in clones in Pax2/Confetti mice at day 30 after IRI + DMSO ($n = 4$) or IRI ($n = 5$) and at day 30 after IRI + TSA or 4-PBA treatment ($n = 4$). Mann–Whitney test. \*$p < 0.05$ IRI + TSA vs. IRI + DMSO and IRI vs. IRI + 4-PBA. **g–k** Clones in Pax2/Confetti mice at day 30 after IRI ($n = 4$) (**g**), and after IRI+ 4-PBA treatment ($n = 4$) (**h**) or +TSA treatment ($n = 4$) (**i–k**). Arrows indicate clones. Data are mean ± SEM. Scale bars 20 µm. Pax2+ = Pax2 lineage-positive cells

CDK4 positivity, a kinase selectively expressed along the nuclear membrane in cells in G1 cell cycle phase[30]. Triple positivity of nuclear membrane CDK4, nuclear p-H3, and the proximal tubule marker Lotus Tetragonolobus Lectin (LTA) revealed that 44.9 ± 7.6% of proximal TECs in biopsies vs. 8.5 ± 3.5% in control kidneys were endocycling cells (Fig. 8a–e). By contrast, 7.9 ± 1.3% of proximal TECs in biopsies were single labeled for p-H3 vs. 3.8 ± 1.7% in control kidneys suggesting these were the only truly cycling cells. When endocycling cells were evaluated on all the tubular cells by colabelling for Phalloidin, the percentage declined to 18 ± 3.1% in biopsies vs. 2.9 ± 2.2% in control kidneys, suggesting endocycling cells are highly enriched in cortical proximal tubules (Fig. 8f). Consistently, fluorescence in situ hybridization for the Y chromosome on Phalloidin-stained biopsies detected 19.1 ± 4.3% of TECs with two, and 1.9 ± 0.7% of TECs with three or more Y chromosomes but a single nucleus (Fig. 8g, h). These cells were not observed in healthy controls.

These observations validate the experimental data from mice in patients with CKD after AKI and show that the majority of TECs

stained by "proliferation markers" undergo endocycle-mediated hypertrophy.

## Discussion

In this study, we questioned the current paradigm that functional recovery after AKI relates to a regenerative capacity of all TECs[5,6]. Indeed, our data demonstrate that: (1) AKI involves a permanent loss of TECs even when GFR recovery occurs; (2) Pax2+ TECs are endowed with higher resistance to death and clonogenic capacity and are responsible for both spontaneous and drug-enhanced regeneration of necrotic tubule segments after AKI; (3) Only Pax2+ TECs complete mitosis, while other TECs rather undergo endoreplication-mediated hypertrophy; (4) Endocycle is a dominant TEC response upon AKI also in humans.

Previous studies concluded on a diffuse proliferative response of surviving TECs based on immunolabelling for cell cycle proliferation markers, a technical approach obviously unable to verify cell division[5–7,25]. Indeed, TECs with mitotic figures are

rarely found in AKI kidneys[31]. Using three independent techniques, we found that only a small subset of TECs undergos mitosis and contributes to kidney regeneration. To address the discrepancy between widespread positivity of proliferation markers and limited clonal cell division, we first evaluated the possibility that the tubule contains a progenitor population deputed to

replacement of lost TECs using lineage tracing of Pax2+ cells, a putative intratubular progenitor population. Pax2+ cells are mostly located in the S3 segment and the distal tubule, the tubule segments injured in ischemic or toxic AKI[32], reminiscent of another subset of possible tubular progenitors characterized by Sox9 expression[33,34]. Our lineage tracing strategy unambiguously

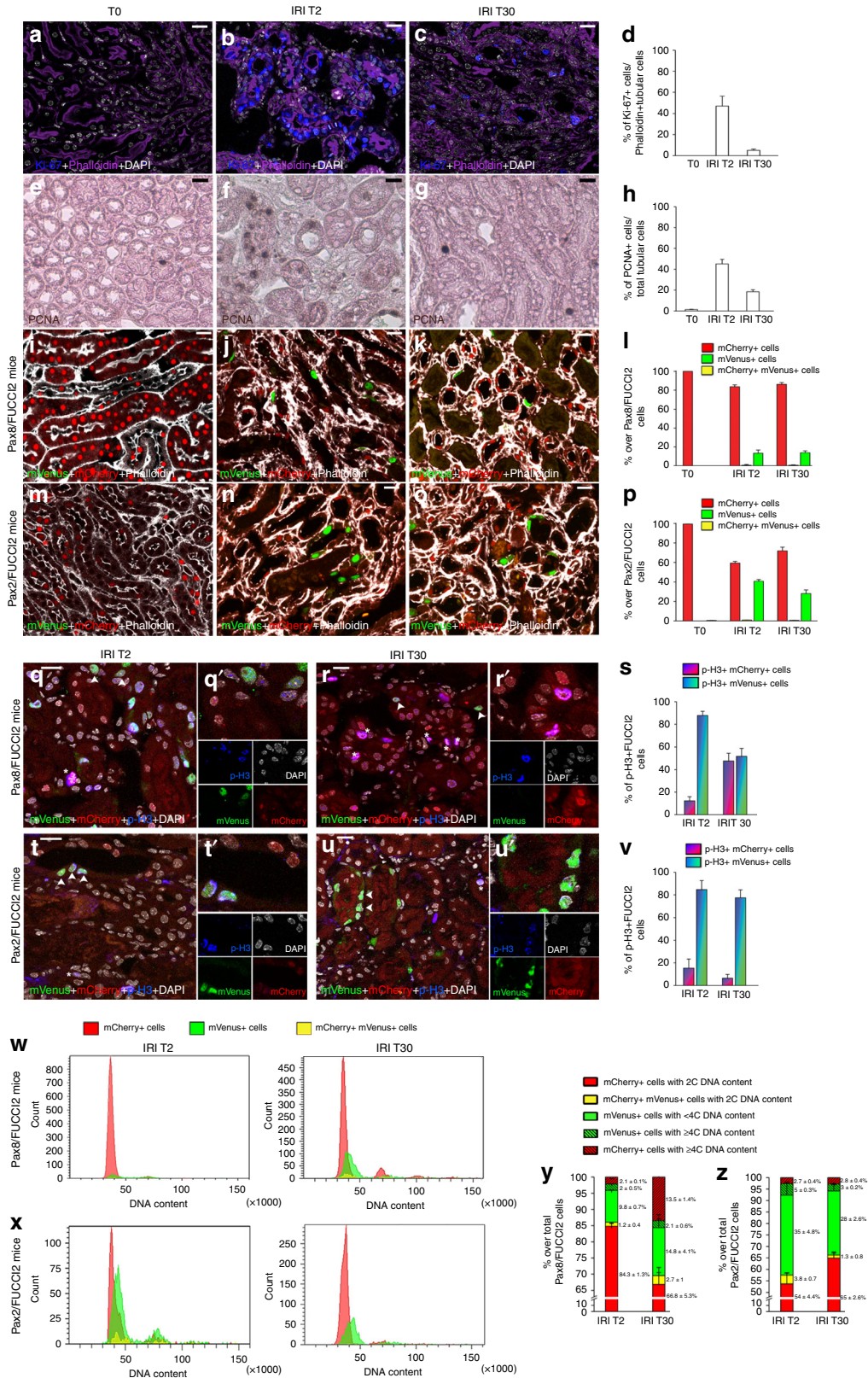

excluded upregulation of Pax2 expression upon AKI as a confounding factor. Rather, Pax2+ cells represented a distinct tubular cell subpopulation and were more resistant to death, explaining why some TECs survive an injury killing most of the other TECs. This was further highlighted by their lack of expression of the TEC injury marker KIM-1 that was instead largely upregulated by other TECs upon IRI. Moreover, Pax2+ cells displayed high clonogenic activity, and were the only TECs to efficiently complete mitosis after AKI, regenerating even long tubule segments. Indeed, 3D evaluation of regenerated tubule segments revealed very long Pax2+ clones that were not visible with the classical 2D analysis that underestimated the length of clones. Interestingly, being unable to detect long clones, the 2D analysis slightly overestimated the percentage of dividing TECs, even if the difference was not statistically significant when 40 μm thick 3D sections were compared with 2D images. This suggests that the percentage of dividing TECs would likely even be lower when analyzing the whole organ. Taken together these results indicate that new tubular cells arise exclusively from pre-existing labeled Pax2+ cells that expand regenerating the entire S3 or distal tubule segment and therefore behaving as progenitor cells[35].

Our results may appear in conflict with those of other studies reporting widespread TEC proliferation with lineage tracing techniques upon IRI[36,37]. However, although we report a percentage of dividing TECs after IRI of only 8.3%, the clone frequency observed in our study in the IRI model at the end of the lineage tracing period is comparable to the one reported by Kusaba et al.[36] for the 26 min IRI model ($21.1 \pm 0.9\%$ vs. 27.4%). The difference in our conclusions is related to our demonstration for the first time that in addition to proliferation, two more phenomena occur after AKI: massive tubular cell loss and endoreplication. Indeed persistent cell loss, enriching dividing clones when calculated over the total number of TECs remained at the end of the lineage tracing period, leads to a profound overestimation of dividing cell number. This phenomenon is explained in detail in Supplementary Fig. 2. In addition, endocycling cells are labeled by cell cycle activation markers such as PCNA or KI-67, representing a further cause of overestimation. Refering to these markers inappropriately as "proliferation markers" now appears as a major reason for invalid conclusions. Indeed, the detection of irreversible TEC loss and endocycle of surviving TECs requires the combination of clonal lineage tracing, as well as cell cycle plus DNA content analysis, as done in this study. Endoreplication cycles (endocycles) are cell cycle variants repeatedly proceeding from G1 to S without passing mitosis and that can occur in differentiated cells[28]. For example, during liver homeostasis, hepatocyte progenitors remain diploid, while differentiated hepatocytes become polyploid via endoreplication[35]. However, polyploid hepatocytes are multinucleated,

while after injury polyploid TECs remain mononucleated and hence cannot be detected with standard tools[38,39]. Combining FUCCI2 transgenic labeling of cell cycle phase with analysis of DNA content by flow cytometry, as reported by Ganem et al.[39], we discovered that upon AKI the majority of TECs undergoes endocycle-mediated hypertrophy. These results challenge the current paradigm of widespread TEC proliferation contributing to kidney regeneration after AKI and rather imply that TEC endocycle-mediated hypertrophy enhances kidney function recovery after injury. Consistently, in *Drosophila melanogaster*, compensatory cellular hypertrophy involves endoreplication[40]. Indeed, the increased DNA content allows a higher transcription capacity that can facilitate an increase in cell size[28,40], as observed in our AKI models.

Thus, the results of this study rather suggest that the majority of TECs entering the cell cycle after AKI undergoes hypertrophy following TEC loss, conceptually similar to hypertrophy of remaining podocytes upon podocyte loss[41]. Kidney regeneration occurs only via a limited clonal response from TEC progenitors resulting in a net TEC loss, implying irreversible nephron loss and subsequent CKD. The more cells are lost, the earlier CKD becomes clinically apparent. Since TEC hypertrophy sustains only function but not tissue regeneration, the presence of endocycling TECs can serve as an indirect indicator of irreversible TEC loss and potentially as a prognostic indicator of the risk for CKD progression, as suggested by their detection in biopsies of patients that developed CKD after an AKI episode.

Finally, the results of this study propose tubular progenitors as a target for treatment of AKI. Indeed, treatment of mice with drugs that were previously reported to improve tissue regeneration following AKI[16–19], showed that their effect resulted from Pax2+ progenitor proliferation that avoids development of tissue fibrosis and CKD.

In conclusion, the renal response to AKI involves two crucial mechanisms: (1) endocycle-mediated hypertrophy of surviving TECs that contributes to recover renal function despite significant loss of renal mass; (2) survival and mitosis of tubular progenitors that provide TEC regeneration. These data challenge the current paradigm of kidney regeneration upon AKI and identify Pax2+ progenitors as the cellular source of a limited intrinsic regenerative capacity of kidney tubules. Instead, endocycle-mediated hypertrophy and persistent tubular cell loss are the predominant features after AKI. Therapeutic targeting of tubule progenitors could be a valuable strategy to improve long-term AKI outcomes.

## Methods

**Pax8/Confetti mice and Pax2/Confetti mice.** The *Pax8.rtTA;TetO.Cre;R26. Confetti* (Pax8/Confetti) mice or the *Pax2.rtTA;TetO.Cre;R26.Confetti* (Pax2/Confetti) mice were developed on a full C57BL/6 background by crossing the

---

**Fig. 6** Cell cycle markers misrepresent proliferation after AKI because numerous TEC endocycle. **a–c** Ki-67+ (blue) Phalloidin+ (purple) tubules at day 0 (**a**), 2 (**b**) and 30 after IRI (**c**) ($n = 3$ per group). **d** Percentage of Ki-67+ cells over Phalloidin+ TECs at T0, IRI T2, and IRI T30 ($n = 3$ per group). **e–g** PCNA + cells at T0 (**e**), IRI T2 (**f**), and IRI T30 (**g**) ($n = 5$ per group). **h** Percentage of PCNA+ cells over TECs at T0, IRI T2, and IRI T30 ($n = 5$ per group). **i–k** mCherry+ cells (red) and mVenus+ cells (green) in Pax8/FUCCI2 mice at T0 (**i**), IRI T2 (**j**), and IRI T30 (**k**) ($n = 4$ per group). Phalloidin staining is white. **l** Percentage of mCherry+ cells, mVenus+ cells and mCherry+mVenus+ cells in Pax8/FUCCI2 mice at T0, IRI T2, and IRI T30. ($n = 4$ per group). **m–o** mCherry+ cells (red) and mVenus+ cells (green) in Pax2/FUCCI2 mice at T0 (**m**), IRI T2 (**n**) and IRI T30 (**o**) ($n = 4$ per group). Phalloidin staining is white. **p** Percentage of mCherry+ cells, mVenus+ cells and mCherry+mVenus+ cells in Pax2/FUCCI2 mice at T0, IRI T2, and IRI T30. ($n = 4$ per group). **q, r** mCherry+ cells (red), mVenus+ cells (green) and p-H3+ cells (blue) in Pax8/FUCCI2 mice at IRI T2 (**q**) and IRI T30 (**r**) ($n = 4$ per group). **q′, r′** Details indicated by arrowheads and asterisks in **q** and **r**. **s** Percentage of p-H3+mCherry+ cells and p-H3+mVenus+ cells in Pax8/FUCCI2 mice at IRI T2 and IRI T30. ($n = 4$ per group). **t, u** mCherry+ cells (red), mVenus+ cells (green) and p-H3+ cells (blue) in Pax2/FUCCI2 mice at IRI T2 (**t**) and IRI T30 (**u**) ($n = 4$ per group). **t′, u′** Details indicated by arrowheads in **t** and **u**. **v** Percentage of p-H3+mCherry+ cells and p-H3+mVenus+ cells in Pax2/FUCCI2 mice at IRI T2 and IRI T30. ($n = 4$ per group). **w, x** Cell cycle distribution of mCherry+, mVenus+, and mCherry+mVenus+ cells in Pax8/FUCCI2 mice (**w**) and Pax2/FUCCI2 mice (**x**) at IRI T2 ($n = 4$, left) and at IRI T30 ($n = 5$, right). Representative experiments are shown. **y, z** Percentage of cells over total Pax8/FUCCI2 cells (**y**) and over total Pax2/FUCCI2 cells (**z**) at IRI T2 ($n = 4$) and IRI T30 ($n = 5$). Data are mean ± SEM. Scale bars 20 μm. DAPI (white) counterstains nuclei. T0 = day 0, IRI T2 = day 2 after IRI, IRI T30 = day 30 after IRI

Confetti strain *Gt(ROSA)26Sor^tm1(CAG-Brainbow2.1)Cle/J* with the TetO.Cre strain *B6. Cg-Tg(TetO-Cre)1Jaw/J*, both purchased from the Jackson Laboratory (Bar Harbor, ME, USA). Double transgenic mice were then crossed with a *Pax8.rtTA* mouse (*B6. Cg-Tg(Pax8-rtTA2S*M2)1Koes/J*, Jackson Laboratory) or with a *Pax2.rtTA* mouse[14], to obtain a triple transgenic inducible mouse model. Mice were genotyped and only triple hemizygous mice were used in this study.

Reporter transgene recombination was induced at 5 weeks of age by administration of 2 mg/ml or 0.25 mg/ml doxycycline hyclate (Sigma-Aldrich, St. Louis, MO, USA) for *Pax8/Confetti* mice and 2 mg/ml doxycycline hyclate for

*Pax2/Confetti* mice in drinking water additioned with 2.5% sucrose (Sigma-Aldrich) for 10 days. Doxycycline administration induces a permanent recombination of a single color encoding gene (red, yellow, green, or blue fluorescent proteins, RFP, YFP, GFP, and CFP), with GFP cells occurring at lower frequency than other colors[12]. Further recombination outcome may also result in no fluorescent reporter labeling in Cre-expressing cells. Following the induction, mice were kept in a washout period of 1 week and were then killed (day 0, T0, $n = 5$ for *Pax8/Confetti* mice induced with 2 mg/ml doxycycline hyclate, $n = 5$ for *Pax8/Confetti* mice induced with 0.25 mg/ml doxycycline hyclate and $n = 4$ for

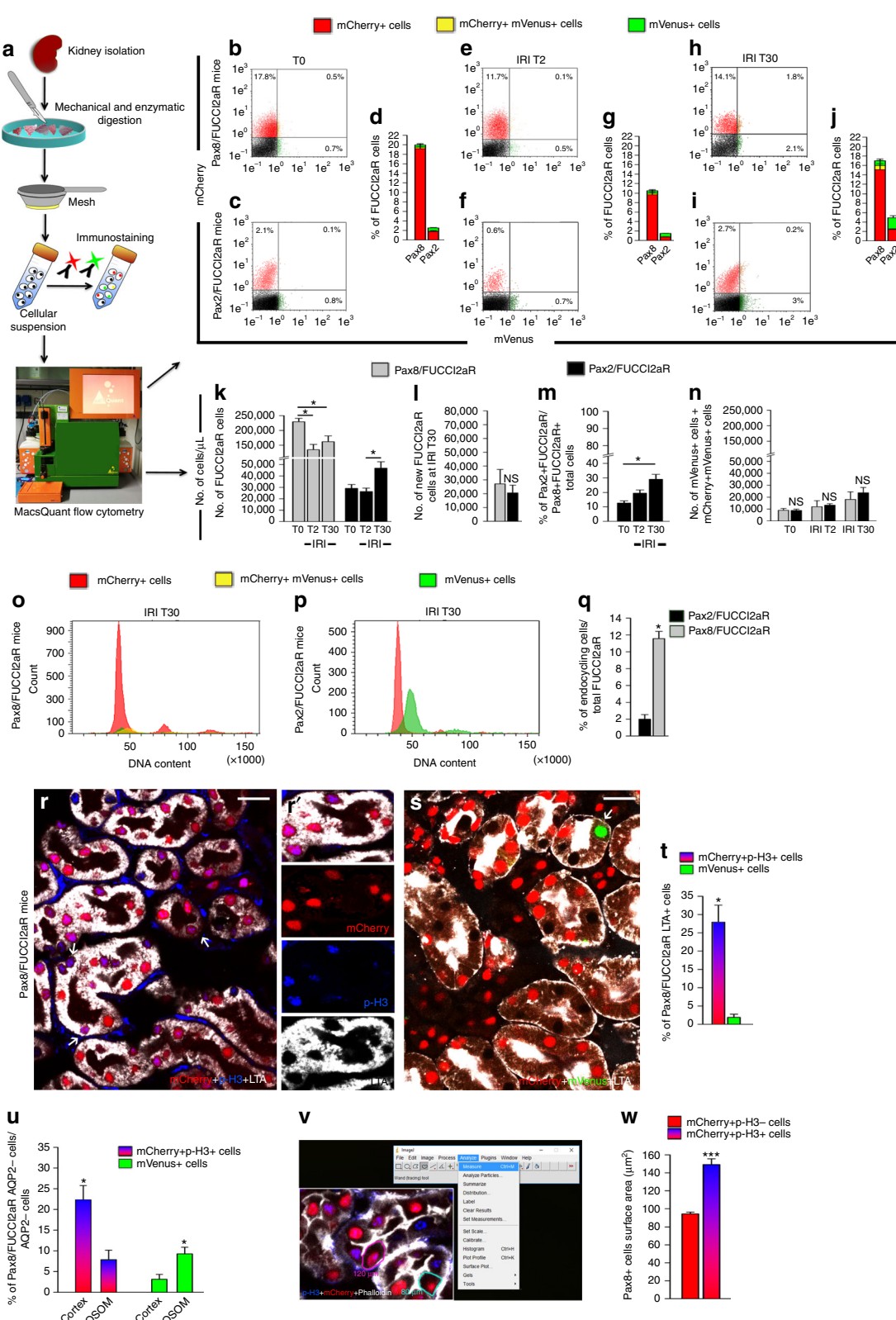

Pax2/Confetti mice) while another group was held in washout for an additional 30 days (T30) and then mice were killed (n = 5 for Pax2/Confetti mice, n = 4 Pax8/Confetti mice induced with 0.25 mg/ml doxycycline hyclate). An additional group was induced with doxycycline for 10 days and was maintained for an additional 37 days in continuous induction (doxycycline T30) (Pax2/Confetti mice n = 4). Another group of Pax2/Confetti mice (n = 3) was induced at 12 weeks of age for 10 days, then it was kept in a washout period of 1 week and then it was killed. In all these groups the number of male and female mice was equally distributed. Another experimental group, following the induction and the washout period, underwent a unilateral IRI of 30′ or an intramuscular injection with hypertonic glycerol and was then killed at day 30 (n = 16 for Pax8/Confetti, n = 11 for Pax2/Confetti). Sham-operated mice (n = 5 Pax2/Confetti, n = 4 Pax8/Confetti mice) underwent the same surgical procedure of ischemic mice, but did not experienced left renal artery clamping and did not suffered of ischemia.

An additional group of Pax2/Confetti mice were induced with doxycycline for 10 days and after the washout period mice underwent a unilateral IRI of 30′. Twenty-four hour after injury, mice were treated with HDACi, trichostatin (TSA, Sigma-Aldrich) 0.5 mg/Kg/day in 40 µL DMSO for 3 weeks i.p. (n = 4) or 4-phenylbutyrate (4-PBA, Sigma-Aldrich) 1 gr/Kg/day for 3 weeks in drinking water (n = 4). In a control group (n = 4), mice were treated with DMSO for 3 weeks. Mice were killed at day 30. Animals with identical genotype and similar age, were assigned to experimental groups in a blinded manner.

To verify if the transgenic system exhibited any type of leakage or non-specific transgene expression, Pax8/Confetti and Pax2/Confetti mice without induction were killed at T0 and at day 30 after IRI (n = 3 for each transgenic mouse at each time point).

All animals were killed by $CO_2$ chamber and kidneys were collected, incubated in 4% paraformaldehyde (PFA, Sigma-Aldrich) in PBS (Sigma-Aldrich) for 2 h at 4 °C followed by immersion in a 15% sucrose solution in PBS for 2 h at 4 °C and, subsequently, in a 30% sucrose solution in PBS overnight at 4 °C, then frozen.

**Pax8/FUCCI2 mice and Pax2/FUCCI2 mice**. To visualize the cell cycle progression of Pax2+ and Pax8+ cells, Pax2.rtTA;TetO.Cre;R26.FUCCI2 (Pax2/FUCCI2) or Pax8.rtTA;TetO.Cre;R26.FUCCI2 (Pax8/FUCCI2) mouse models were employed. To create the two strains, we crossed Pax2.rtTA;TetO.Cre or Pax8.rtTA;TetO.Cre mice with both R26R-mCherry-hCdt1(30/120) (CDB0229K) and R26R-mVenus-hGem(1/110) (CDB0230K) mice (both obtained from CDB Laboratory for Animal Resources and Genetic Engineering, RIKEN Kobe, Japan, http://www.cdb.riken.jp/arg/reporter_mice.html), which combined constitute the FUCCI2 system. We thus obtained quadruple transgenic mice in a full C57BL/6 background that were genotyped as reported in "Genotyping" paragraph.

In the resulting mice, following doxycycline administration, Pax2 or Pax8 promoter drives the expression of the fluorescent protein mCherry-hCdt1 (30/120) (red) in nuclei of cells in G1 phase, and of the fluorescent protein mVenus-hGem (1/110) (green) in nuclei of cells in S/G2/M phase. Cells can also appear as yellow at the G1/S boundary, but this is a very short event and thus yellow cells are rare. To induce reporter expression, at 5 weeks of age male mice were treated with 2 mg/ml of doxycycline in 2.5% sucrose water ad libidum, then were kept in a washout period of 1 week (T0, Pax2/FUCCI2 n = 4, Pax8/FUCCI2 n = 4). The percentage of induction was defined in healthy mice as the number of Pax8-lineage-positive cells expressing mCherry or mVenus on the total number of tubular cells per field. After the washout period mice underwent a unilateral IRI of 30 min (same procedure performed in Confetti mice), and were then killed at day 2 (Pax2/FUCCI2 n = 8, Pax8/FUCCI2 n = 8) and at day 30 (Pax2/FUCCI2 n = 9, Pax8/FUCCI2 n = 12). Animals with identical genotype and similar age were assigned to experimental groups in a blinded manner. To verify if the transgenic system exhibited any type of leakage or non-specific transgene expression, Pax8/FUCCI2 and Pax2/FUCCI2

mice without induction were killed at T0 and at day 30 after IRI (n = 3 for each transgenic mouse at each time point).

All animals were killed by $CO_2$ chamber and kidneys were collected and evaluated by the confocal microscopy or by flow cytometry. To perform confocal analysis, kidneys were processed as described for Confetti mice.

**Pax8/FUCCI2aR mice and Pax2/FUCCI2aR mice**. The FUCCI2aR represents a significant advancement on the FUCCI model, since it produces iso-stoichiometric quantities of both FUCCI probes without the existence of the "dark phase"at the end of each cell cycle. This newer model has the advantage that both probes are always expressed in the same ratio, simplifying the detection of green–red transition and making it possible to directly trace and quantify the number of labeled cells by flow cytometry using the MACSQuant software (Miltenyi Biotec S.r.l., Bologna, Italy). For this reason, we crossed Pax8.rtTA;TetO.Cre; as well as Pax2.rtTA;TetO.Cre; mice with mice harboring the Fluorescent Ubiquitin-based Cell cycle Indicator (FUCCI2aR) Cre-dependent reporter (European Mouse Mutant Archive (EMMA), INFRAFRONTIER-I3, Neuherberg-München, Germany), which consists of a bicistronic Cre-activable reporter of two fluorescent proteins whose expression alternates based on cell cycle phase: mCherry-hCdt1 (30/120) (red), expressed in nuclei of cells in G1 phase, and mVenus-hGem (1/110) (green), expressed in nuclei of cells in S/G2/M. We thus obtained triple transgenic mice in a full C57BL/6 background that were genotyped as reported in "Genotyping" paragraph.

In the resulting mice, following doxycycline administration, Pax2 or Pax8 promoter drives the expression of the fluorescent protein mCherry-hCdt1 (30/120) (red) in nuclei of cells in G1 phase, and of the fluorescent protein mVenus-hGem (1/110) (green) in nuclei of cells in S/G2/M phase. Cells can also appear as yellow at the G1/S boundary, but this is a very short event and thus yellow cells are rare.

To induce reporter expression, at 5 weeks of age male mice were treated with 2 mg/ml of doxycycline in 2.5% sucrose water ad libidum, then were kept in a washout period of 1 week (T0, Pax2/FUCCI2aR n = 4, Pax8/FUCCI2aR n = 4). The percentage of induction was defined in healthy mice as the number of Pax8-lineage-positive cells expressing mCherry or mVenus on the total number of tubular cells per field. After the washout period male mice underwent a unilateral IRI of 30 min (same procedure performed in Confetti mice), and were then killed at day 2 (Pax2/FUCCI2aR n = 4, Pax8/FUCCI2aR n = 4) and at day 30 (Pax2/FUCCI2aR n = 9, Pax8/FUCCI2aR n = 9). Animals with identical genotype and similar age were assigned to experimental groups in a blinded manner. To verify if the transgenic system exhibited any type of leakage or non-specific transgene expression, Pax8/FUCCI2aR and Pax2/FUCCI2aR mice without induction were killed at T0 and at day 30 after IRI (n = 3 for each transgenic mouse at each time point).

All animals were killed by $CO_2$ chamber and kidneys were collected and evaluated by confocal microscopy or by flow cytometry. To perform confocal analysis kidneys were processed as described for Confetti mice.

**Genotyping**. Tail biopsies were incubated overnight at 55 °C in lysis reagent (1 M Tris-HCl, pH 8.5; 0.5 M EDTA, 20% SDS, 4 M NaCl, 0.1 mg/ml proteinase K neutralized with 40 mM Tris-HCl, all from Sigma-Aldrich), centrifuged and DNA extracted using isopropanol (Sigma-Aldrich). To distinguish transgene homozygosity from heterozygosity, qRT-PCR were performed by using 5 ng/µl of genomic DNA with LightCycler® 480 SYBR Green I Master (Roche Diagnostics, Rotkreuz, Switzerland). The reactions were performed using a LightCycler® 480 (Roche Diagnostics) with a program consisting of 40 cycles each constituted of an initiation phase at 95 °C for 15 min, annealing phase at 60 °C for 45 min and amplification phase at 72 °C for 60 min. The following primers were used:

**Fig. 7** Pax2 lineage-positive cells proliferate, while other TEC endocycle and are persistently lost after AKI. **a** Schematic procedure. **b–j** FACS analysis shows mCherry+ and mVenus+ cells in total renal cells of Pax8/FUCCI2aR (**b**, **d**) and Pax2/FUCCI2aR mice (**c**, **d**) at T0, of Pax8/FUCCI2aR (**e**, **g**) and Pax2/FUCCI2aR mice (**f**, **g**) at IRI T2 and of Pax8/FUCCI2aR (**h**, **j**) and Pax2/FUCCI2aR mice (**i**, **j**) at IRI T30. A representative experiment out of 4 is shown. **k** Number of total FUCCI2aR cells in Pax8/FUCCI2aR and Pax2/FUCCI2aR mice at T0, IRI T2, IRI T30 (n = 4 in each group). Mann–Whitney test *p < 0.05. **l** Number of new FUCCI2aR cells in Pax8/FUCCI2aR and Pax2/FUCCI2aR mice at IRI T30 in comparison to IRI T2 (n = 4 in each group). Mann–Whitney test NS. **m** Percentage of Pax2/FUCCI2aR over Pax8/FUCCI2aR total cells at T0, IRI T2, IRI T30 (n = 4 in each group). Mann–Whitney test *p < 0.05. **n** Number of total mVenus+ cells mCherry+mVenus+ cells in Pax8/FUCCI2aR and Pax2/FUCCI2aR mice at T0, IRI T2, IRI T30 (n = 4 in each group). Mann–Whitney test NS. **o**, **p** Cell cycle distribution of mCherry+, mVenus+, and mCherry+mVenus+ cells in Pax8/FUCCI2aR (**o**) and Pax2/FUCCI2aR mice (**p**) at IRI T30. A representative experiment out of 4 is shown. **q** Percentage of endocycling cells in Pax2/FUCCI2aR (black column) and Pax8/FUCCI2aR mice (light gray column) at IRI T30 (n = 4 in each group). Mann–Whitney test *p < 0.05. **r** mCherry+ cells (red) and p-H3+ cells (blue) in LTA+ tubules in the cortex (white) of Pax8/FUCCI2aR mice at IRI T30 (n = 5). Arrows show mCherry+p-H3+ cells. **r'** Detail of **r**. **s** mCherry+ cells (red) and mVenus+ cells (green) in LTA+ tubules in the cortex (white) of Pax8/FUCCI2aR mice at IRI T30 (n = 5). Arrow shows mVenus+ cell. **t** Percentage of endocycling cells (mCherry+p-H3+ cells) and cycling cells (mVenus+ cells) in the cortex of Pax8/FUCCI2aR mice at IRI T30 (n = 5). Mann–Whitney test *p < 0.05. **u** Percentage of endocycling (mCherry+p-H3+ cells) and cycling (mVenus+ cells) AQP2− cells in the cortex and in OSOM of Pax8/FUCCI2aR mice at IRI T30 (n = 5). Mann–Whitney test *p < 0.05. **v** Measurement of the cell surface area of mCherry+p-H3+ cells and of mCherry+p-H3− cells after staining with Phalloidin (white) with Image J software in Pax8/FUCCI2aR mice at IRI T30 (n = 4). **w** Cell surface area of mCherry+p-H3+ cells and of mCherry+p-H3− cells in Pax8/FUCCI2aR mice at IRI T30. n = at least 20 cells for each mouse (n = 4), Mann–Whitney test ***p < 0.001. Data are mean ± SEM. Scale bars 20 µm. T0 = day 0, IRI T2 = day 2 after IRI, IRI T30 = day 30 after IRI

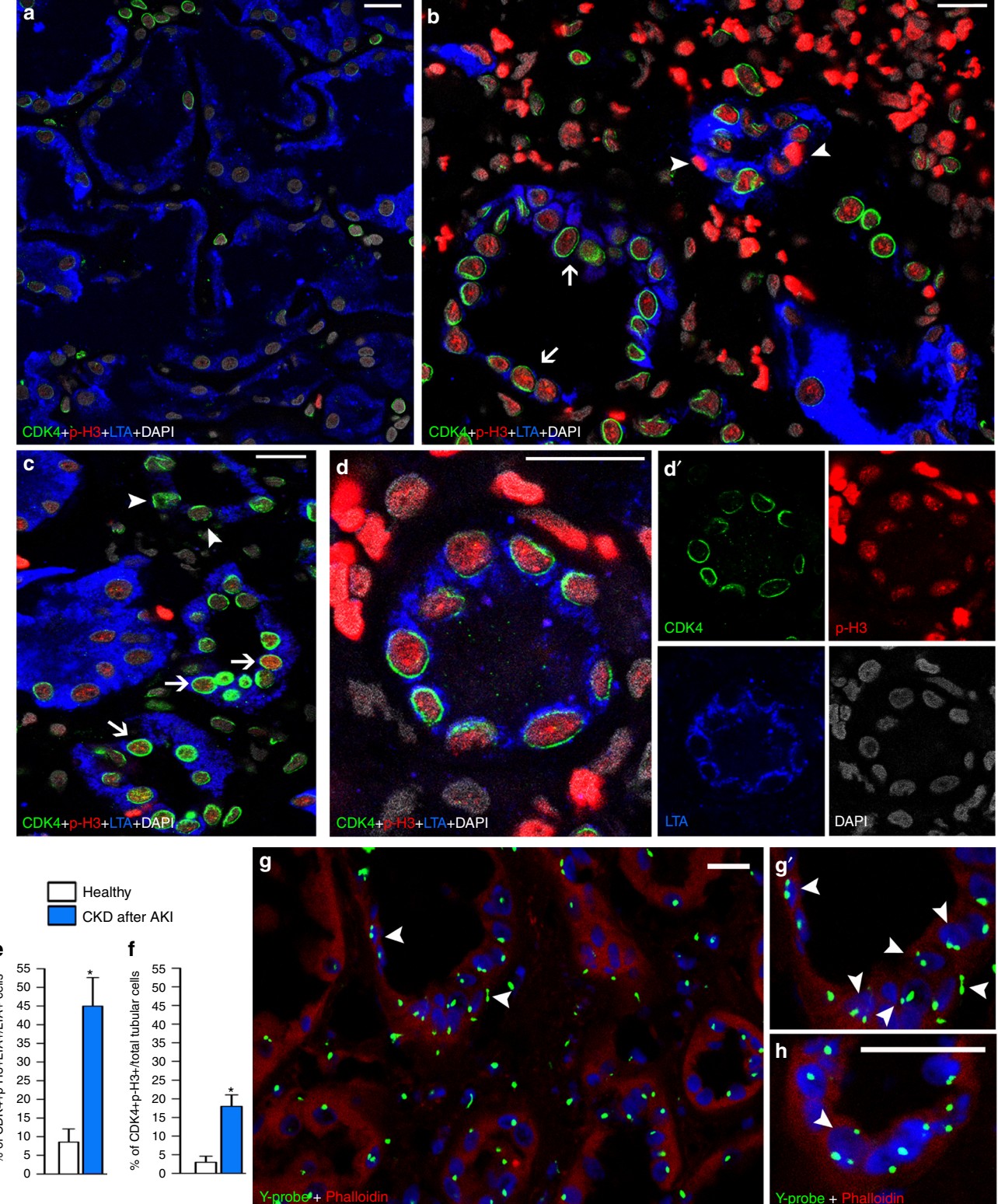

**Fig. 8** High percentage of endocycling TECs in the kidney of patients that developed CKD after AKI. **a** A healthy kidney section showing the expression of CDK4 (green) and p-H3 (red) in LTA+ proximal tubular cells (blue) (*n* = 4). **b–d** A renal biopsy section from a patient that developed CKD after AKI (*n* = 7) showing CDK4+p-H3+ (green and red cells, arrows) and CDK4−p-H3+ (red cells, arrowheads) or CDK4+p-H3− (green cells, arrowheads) in proximal tubular LTA+ cells (blue). **d′** Split images of the tubule shown in **d**. **e** Percentage of CDK4+p-H3+LTA+ cells over total proximal TECs (LTA+) in healthy kidneys (*n* = 4) and in renal biopsies from patients that developed CKD after AKI (*n* = 7). Mann–Whitney test *\*p* < 0.05. **f** Percentage of CDK4+p-H3+ cells over total TECs in healthy kidneys (*n* = 4) and in renal biopsies from patients that developed CKD after AKI (*n* = 7). Mann–Whitney test *\*p* < 0.05. **g, g′** A renal biopsy section from a patient that developed CKD after AKI (*n* = 8) showing the presence of two or three Y chromosome-probe (green) in TECs labeled with Phalloidin (red), at different magnifications. Arrowheads indicate doubled dots for Y chromosome in a nucleus. **h** High magnification of another biopsy field. Arrowhead indicates triple dots for Y chromosome in a nucleus. DAPI counterstains nuclei. Data are mean ± SEM. Scale bars 20 μm

**Table 1 List of patients with CKD after AKI analyzed in this study**

| Patient | Gender | Age (years) | Diagnosis | Number and severity of AKI episode(s) | CKD stage | Time since last AKI episode(s) and biopsy (days) |
|---|---|---|---|---|---|---|
| 1. | M | 10 | TIN | 1× AKIN3 | CKD2 | 45 |
| 2. | M | 5 | IRI | 2× AKIN3 | CKD3 | 180 |
| 3. | M | 59 | IRI | 3× AKIN1 | CKD4 | 365 |
| 4. | M | 56 | Sepsis | 1× AKIN3 | CKD4 | 30 |
| 5. | M | 73 | Light chain ATN | 1× AKIN1, 1× AKIN2 | CKD4 | 365 |
| 6. | M | 57 | IRI | 1× AKIN3 | CKD4 | 730 |
| 7. | F | 55 | IRI | 1× AKIN1, 2× AKIN2 | CKD4 | 270 |
| 8. | F | 30 | IRI | 1× AKIN3 | CKD3 | 1460 |
| 9. | M | 56 | IRI | 1× AKIN3 | CKD4 | 45 |
| 10. | M | 38 | IRI | 1× AKIN2 | CKD3 | 30 |
| 11. | M | 44 | RCC-normal tissue | None | None | |
| 12. | M | 86 | RCC-normal tissue | None | None | |
| 13. | M | 26 | RCC-normal tissue | None | None | |
| 14. | M | 74 | RCC-normal tissue | None | None | |
| 15. | F | 51 | RCC-normal tissue | None | None | |

*TIN* tubulointerstitial nephritis, *IRI* ischemia reperfusion injury, *ATN* acute tubular necrosis, *AKIN* Acute Kidney Injury Network criteria, number indicates stage, *CKD* chronic kidney disease, number indicates stage, *RCC* renal cell carcinoma

Pax2.rtTA and Pax8.rtTA forward 5′-AACGCACTGTACGCTCTGTC-3′ and reverse 5′-GAATCGGTGGTAGGTGTCTC-3′; TetO.Cre forward 5′-TCGCTGCATTACCGGTCGATGC-3′ and reverse 5′-CCATGAGTGAACGAACCTGGTCG-3′. TCRα genomic DNA was used as gene housekeeping for relative quantification and was amplified by using the forward 5′-CAAATGTTGCTTGTCTGGTG-3′ and the reverse 5′-GTCAGTCGAGTGCACAGTTT-3′ primers.

To distinguish FUCCI2aR and Confetti homozygosity from heterozygosity, PCR were performed with the following primers and parameters: Confetti forward 5′-GAATTAATTCCGGTATAACTTCG-3′ and reverse 5′-AAAGTCGCTCTGAGTTGTTAT-3′, 3 min 94 °C, 35 cycles of 30 s 94 °C, 30 s 58 °C, 30 s 72 °C and, finally, 2 min 72 °C; FUCCI-P3 5′-TCCCTCGTGATCTGCAACTCCAGTC-3′, FUCCI-P4 5′-AACCCCAGATGACTACCTATCCTCC-3′ and FUCCI-P4 5′-GGGGGAGGATTGGGAAGACAATAGC-3′; 2 min 96 °C, 35 cycles of 96 °C 30 s, 65 °C 30 s and, finally, 72 °C for 30 s.

**Renal ischemia reperfusion injury.** Male mice were anesthetized by intraperitoneal injection of Ketamine (100 mg/kg)/Xylazine (10 mg/kg, Bio98 S.r.L, Milan, Italia), which produced short-term surgical anesthesia with good analgesia. The animals were kept warm in a 37 °C ventilated heating chamber before surgery and their body temperature was monitored constantly by a rectal probe and maintained in the range of 36.5–37 °C. The mouse was placed on a thermostatic station laying on the right side, shaved and disinfected with Povidone-iodine and an incision of 1–1.5 cm on the skin on the left side was performed, then the muscle layer below was cut and opened. The left kidney was then externalized pushing it out from the cut with sterile cotton swabs to expose the renal pedicle for clamping. The renal artery was clamped to block blood flow to the kidney and cause ischemia (visible by color change of the kidney from red to dark purple). The duration of kidney ischemia starts from the time of clamping. After verification of the color changes, the kidney was returned to the abdomen cavity and the animal returned to the ventilated heating chamber. The duration of ischemia was 30 min, after which the clamp was removed to start the reperfusion, which is indicated by the change of kidney color back to red. The muscle layer was sutured, followed by the closure of the skin wound with metal clips. Immediately after the wound closure, 0.5 ml warm sterile saline (0.9% NaCl) was given subcutaneously to each mouse to rehydrate it. The right contralateral kidney was maintained untouched as a control.

Sham-operated mice underwent the same surgical procedure as above, but did not experienced left renal artery clamping and did not suffered of ischemia.

**Nephrotoxic AKI.** Transgenic female mice were induced for 10 days with doxycycline, held 1 week in washout, following which rhabdomyolysis-induced AKI was performed, by intramuscular injection on day 0 with hypertonic glycerol (8 ml/kg body weight of a 50% glycerol solution; Sigma-Aldrich) into the inferior hind limbs.

**Transcutaneous measurement of glomerular filtration rate.** Measurement of the glomerular filtration rate (GFR) was done as described elsewhere[42]. In brief, mice ($n = 9$ Pax2/Confetti ischemic mice, $n = 13$ Pax8/Confetti ischemic mice, $n = 5$ Pax8/Confetti sham-operated mice, $n = 5$ Pax2/Confetti sham-operated mice, $n = 5$ Pax8/Confetti healthy mice, $n = 5$ Pax2/Confetti healthy mice, $n = 7$ Pax8/Confetti glycerol-treated mice, $n = 8$ Pax2/Confetti glycerol-treated mice, $n = 7$ Pax2/Confetti ischemic mice treated with DMSO, $n = 6$ treated with PBS, $n = 7$ treated with TSA, $n = 6$ treated with PBA) where anesthetized with

isoflurane and a miniaturized imager device built from two light-emitting diodes, a photodiode and a battery (Mannheim Pharma and Diagnostics GmbH, Mannheim, Germany) were mounted via a double-sided adhesive tape onto the shaved animals' neck. For the duration of recording (~1.5 h) each animal was conscious and kept in a single cage. Prior to the intravenous injection of 150 mg/kg FITC-sinistrin (Mannheim Pharma and Diagnostics GmbH), the skin's background signal was recorded for 5 min. After removing the imager device, the data were analyzed using MPD Studio software ver.RC6 (MediBeacon GmbH Cubex41, Mannheim, Germany)[43]. The GFR [µl/min] was calculated from the decrease of fluorescence intensity over time (i.e., plasma half-life of FITC-sinistrin) using a two-compartment model, the animals body weight and an empirical conversion factor. For each time point, GFR value was normalized on the value at baseline and on the sham (for ischemic mice) or healthy mice of comparable age and weight (for glycerol-treated mice) value. Based on the results obtained in preliminary experiments to assess GFR measurement variability, only mice showing a drop of GFR after AKI of at least 20% over baseline were included in the study.

**Estimation of blood urea nitrogen.** Renal function was assessed by collecting a small amount of blood from mice ($n = 5$ Pax8/Confetti ischemic mice, $n = 5$ Pax8/Confetti sham-operated mice, $n = 6$ Pax8/Confetti glycerol-treated mice, $n = 5$ Pax8/Confetti healthy mice, $n = 8$ Pax2/Confetti glycerol-treated mice, $n = 5$ Pax2/Confetti healthy mice) with a metal lancet from submandibular plexus at different time points in order to measure BUN levels. Blood parameters were measured in EDTA anticoagulated plasma samples using Urea FS kit (DiaSys Diagnostic Systems, Holzheim, Germany), according to the manufacturer's protocols.

**Assessment of renal injury.** Healthy and ischemic kidneys stored in 4% buffered formalin were embedded in paraffin and 2–4 µm sections were prepared for periodic acid–Schiff (PAS) staining. Ischemic tubular injury and scaring were evaluated by assessing the percentage of tubules in the outer stripe of outer medulla that displayed cell necrosis, tubular dilatation or cast formation (injury), atubular sections with abundant unorganized parenchyma cells and cell infiltrates (scarring). Twenty-four kidneys from Pax2/Confetti mice and 4 fields (20×) per kidney were analyzed. All assessments were performed by a blinded observer.

**Immunofluorescence and confocal microscopy.** Confocal microscopy was performed on 10 µm sections of renal tissues by using a Leica SP5 AOBS confocal microscope (Leica, Wetzlar, Germany) equipped with a Chameleon Ultra-II two-photon laser (Coherent, Milan, Italy).

The following antibodies were used: anti-aquaporin-1 (AQP1, AB2219, dilution 1:100, Millipore, Darmstadt, Germany), anti-Tamm-Horsfall (THP, CL1032A, dilution 1:20, Cederlane, Burlington, Ontario, Canada), anti-aquaporin-2 (AQP2, C-17, SC-9882, dilution 1:25, Santa Cruz Biotechnology, Saint Louis, USA), anti-megalin (P-20, sc-16478, dilution 1:25, Santa Cruz Biotechnology), biotinylated Lotus Lectin, LTA (B-1325, dilution 1:50, Vector Laboratories, Burlingame, USA), anti-KIM-1 (AF1817, dilution 1:40, R&D Systems, Inc., Minneapolis, USA), anti-Pax2 (71–6000, dilution 1:25, Zymed, Thermo Fisher Scientific, MA USA) anti-Ki-67 (ab15580, dilution 1:50, Abcam, Cambridge, UK), anti-GFP-488 (A21311, dilution 1:100, Life Technologies, Monza, Italy), anti-phosphorylated Histone 3 (p-H3, ab14955, dilution 1:2000, Abcam), anti-cyclin dependent kinase 4 (CDK4, SC-601, dilution 1:50, Santa Cruz Biotechnology) and Phalloidin-633 (A22284, dilution 1:40, Life Technologies). Alexa-Fluor secondary antibodies were obtained

from Molecular Probes (Life Technologies). For the detection of tubules, staining with Alexa Fluor 546 Phalloidin (A22283, dilution 1:40, Molecular Probes, Life Technologies) was performed. For the Pax8/Confetti mice and for Pax2/Confetti mice the acquisition was set in the cyan, green, yellow, and red wavelengths using 405, 488, 514, and 543 nm wavelength excitation, respectively. For Pax8/FUCCI mice, Pax2/FUCCI2, Pax8/FUCCI2aR, and Pax2/FUCCI2aR mice the acquisition was set in green and red wavelengths using 488, and 543 nm excitation, respectively. mCherry emission was collected through 570/620 nm BP. Nuclei were counterstained with DAPI (62248, dilution 1:1000, Life Technologies) excited with Chameleon Ultra-II two-photon laser at 800 nm.

Cell surface area of at least 20 mCherry+p-H3− cells and 20 mCherry+p-H3+ cells for each Pax8/FUCCI2aR mouse were measured using Image J software. One field is: 620 × 620 μm.

**3D reconstruction**. To generate 3D images (Fig. 4g–i) and Supplementary Movie 1, an image processing software from Leica Microsystems "Leica Application Suite X" was used. Z-series stacks were obtained from 40 μm kidney slices, being this thickness the limit to stain all the nuclei with DAPI, permitting cell counting. Images were collected at 1 μm intervals. Clones were observed in every color, but for 3D reconstruction, we chose mostly the yellow ones cause the YFP is distributed in all the cell, making the 3D reconstruction easier to build and appreciate.

To compare 2D with 3D analysis, two blinded observers counted the number of clones in 2D images and in 3D reconstruction. The percentage of clonogenic cells was calculated as reported below (Eq. 2).

**Immunohistochemistry**. Immunohistochemistry for PCNA was performed in paraffin embedded healthy and ischemic kidney sections ($n = 5$ at day 0, $n = 5$ at day 2 and at day 30 after IRI), as detailed elsewhere[44]. Briefly, after removing the paraffin and rehydrating in water, the tissue sections were submerged in 3% $H_2O_2$ for 10 min to block the endogenous peroxidase activity. The target antigen was retrieved by microwaving the sections in sodium citrate buffer (10 mM, pH 6). The samples were heated in a microwave at 800 W for 3 min and then for 7 min at 400 W. The sections were cooled for 30 min at room temperature. After that, the 10-min heating procedure was repeated with fresh buffer. Sections were pre-incubated for 30 min in 1× PBS, 1% bovine serum albumin, 0.05% saponin and subsequently incubated for 15 min at 37 °C and overnight at 4 °C with mouse anti-PCNA (PC10, ab29, dilution 1:300, Abcam). Then, sections were incubated for 1 h at room tememperature with secondary biotinylated-anti mouse IgG antibody, followed by an incubation with avidin-biotin-peroxidase complex (Vectastain ABC kit, Vector Laboratories) for 30 min, and 3-amino-9-ethylcarbazole (AEC, Vector Laboratories) (red color) as peroxidase substrate for 20 min. Sections were counterstained with Eosin G (Bioptica, Milan, Italy).

**Clone frequency analysis**. Single-cell clones and clones with two or more cells were counted in AQP2− tubules of the outer stripe of outer medulla in Pax8/Confetti mice at day 0 (T0), in age-matched controls (T30), in sham operated, in ischemic kidneys (IRI T30) and in nephrotoxic kidneys (Gly T30).

For each clone size, clone frequency analysis showed in Fig. 1j, was assessed as followed:

$$\text{Clone frequency } (\%) = (n° \text{ of clones of } n \text{ cells/total } n° \text{ of clones}) \times 100, \quad (1)$$

where $1 \leq n \leq 11$ in Pax8/Confetti mice because this is the maximum clone size observed.

Same calculations were performed in Pax2/Confetti mice at T0, T30, sham operated and IRI T30 where $2 \leq n \leq 10$ (Fig. 3f).

Clones observed in a total of at least 30 fields of outer stripe of the outer medulla taken from at least five sections of each mouse were counted by two independent blinded observers. Clone size was established by counting the nuclei counterstained with DAPI.

See Supplementary Methods for a representative calculation.

**Clonogenic cell analysis**. To establish the percentage of clonogenic cells in Pax8/Confetti mice we performed the following calculations:

(A) $n°$ of new clones at day 30 after IRI (IRI T30)=

$$\sum_{n=2}^{11} (\text{clones composed of } n \text{ cells at IRI T30} - \text{clones composed of } n \text{ cells at T0}),$$

where $2 \leq n \leq 11$ in Pax8/Confetti mice because the maximum clone size observed is 11 cells.

(B) % of clonogenic cells at IRI T30 in comparison to T0 (vs. T0) showed in Fig. 1l.

$$(A/n° \text{ of Pax8}^+ \text{cells at T0}) \times 100. \quad (2)$$

Same calculations were performed in Pax2/Confetti mice where $2 \leq n \leq 10$ (Fig. 3j).

Same calculations were performed in Pax8 and Pax2 Confetti mice 30 days after nephrotoxic AKI (Gly T30, Fig. 1l, Fig. 3n).

Same calculations were performed for 3D analysis in Pax8 and Pax2 Confetti mice at IRI T30 (Fig. 4f).

All the analysis was performed in tubular cells excluding collecting ducts (AQP2 immunostained) of the outer stripe of outer medulla. All counts were executed by two independent blinded observers.

See Supplementary Methods for a representative calculation.

**Percentage of new cells after ischemic or nephrotoxic AKI**. To establish the percentage of new cells in Pax8/Confetti mice at IRI T30 we performed the following calculations (Fig. 1k):

(C) $n°$ of Pax8+ cells included in clones at IRI T30

$$\sum_{n=2}^{11} [(\text{clones composed of } n \text{ cells at IRI T30} - \text{clones composed of } n \text{ cells at T0}) \times n],$$

where $2 \leq n \leq 11$ in Pax8/Confetti mice because the maximum clone size observed is 11 cells.

(D) $n°$ of new Pax8+ cells at IRI T30 = $n°$ of Pax8+ cells included in new clones–$n°$ of cells that originated new clones*

$$C - A$$

* the $n°$ of cells that originated new clones coincides with the $n°$ of new clones at IRI T30 (A), because each clone is the progeny of one cell.

(E) % of new Pax8+ cells at IRI T30 in comparison to T0

$$(D/n° \text{ of Pax8}^+ \text{cells at T0}) \times 100. \quad (3)$$

Same calculations were performed in Pax8/Confetti mice after nephrotoxic AKI (Fig. 1k) and in Pax2/Confetti mice at day 30 after IRI (Fig. 3g). In Pax2/Confetti mice $2 \leq n \leq 10$.

All the analysis was performed in tubular cells excluding collecting ducts (AQP2 immunostained) of the outer stripe of outer medulla. All counts were executed by two independent blinded observers.

See Supplementary Methods for a representative calculation.

**Percentage of lost cells after ischemic or nephrotoxic AKI**. To establish the percentage of lost cells in Pax2/Confetti mice after ischemic AKI, we performed the following calculations (Fig. 3i):

$$100 - [(n° \text{ of Pax2}^+ \text{cells at IRI T30} - D/n° \text{ of Pax2}^+ \text{cells at T0}) \times 100]. \quad (4)$$

Same calculations were performed in Pax8/Confetti mice after ischemic AKI (Fig. 3i) and in Pax2/Confetti and Pax8/Confetti mice after nephrotoxic AKI (Fig. 3m).

All the analysis was performed in tubular cells excluding collecting ducts (AQP2 immunostained) of the outer stripe of outer medulla. All counts were executed by two independent blinded observers.

See Supplementary Methods for a representative calculation.

**Percentage of variation over T0**. To establish the percentage of variation over T0 in Pax2/Confetti mice after ischemic AKI, we performed the following calculations (Fig. 3k):

$$[(n° \text{ of Pax2}^+ \text{cells at IRI T30}/n° \text{ of Pax2}^+ \text{cells at T0}) \times 100] - 100. \quad (5)$$

Same calculations were performed in Pax8/Confetti mice after ischemic AKI (Fig. 3k) and in Pax2/Confetti and Pax8/Confetti mice after nephrotoxic AKI (Fig. 3o).

All the analyses were performed in tubular cells excluding collecting ducts (AQP2 immunostained) of the outer stripe of outer medulla. All counts were executed by two independent blinded observers.

**Percentage of variation of Pax2+ cells and Pax2− cells over IRI**. To establish the percentage of variation of Pax2+ cells over IRI in Pax2/Confetti mice after ischemic AKI and treatment with TSA or 4-PBA, we performed the following calculations (Fig. 5e):

$$[(n° \text{ of Pax2}^+ \text{cells after treatment at IRI T30}/n° \text{ of Pax2}^+ \text{cells at IRI T30}) \times 100].$$
$$(6)$$

Same calculations were performed for Pax2− cells (Fig. 5e). All counts were executed by two independent blinded observers.

**Frequency of Pax2+ clones**. This analysis was performed in the cortex after megalin immunostaining, in the OSOM after AQP1 and THP immunostaining and in the collecting ducts after AQP2 immunostaining.

To establish the frequency of Pax2+ clones in nephron segments and in collecting ducts, we performed the following calculations (Fig. 4e):

(F) n° of total Pax2+ new clones at day 30 after IRI (IRI T30)=

$$n° \text{ of new megalin}^+ \text{ clones} + n° \text{ of new AQP1}^+ \text{ clones}$$
$$+ n° \text{ of new THP}^+ \text{ clones} + n° \text{ of new AQP2}^+ \text{clones} .$$

The n° of new clones was determined based on the calculation used in formula A.

(G) frequency of Pax2+ clones (%) in each nephron segment and in collecting ducts=

$$(A/F) \times 100. \qquad (7)$$

**Cell cycle analysis by flow cytometry**. Cell cycle analysis was performed on total FUCCI2 cells and on total FUCCI2aR cells (mCherry+ and mVenus+ cells) in Pax8/FUCCI2 mice, in Pax2/FUCCI2 mice, in Pax8/FUCCI2aR mice, and in Pax2/FUCCI2aR mice at 2 days and at 30 days after IRI.

Pax8/FUCCI2, Pax8/FUCCI2aR, Pax2/FUCCI2, and Pax2/FUCCI2aR kidneys were processed in order to obtain a single cell suspension and to perform cell cycle analysis by flow cytometry. To this aim, kidneys were minced using a scalpel. After addition of 1.5 ml of digestion buffer (300 U/ml Collagenase II and 1 mg/ml Pronase E, Sigma-Aldrich) they were incubated at 37 °C for 20 min. The solution was pipetted up and down with a cut 1000 μl pipette tip every 5 min. The digested kidneys were gently pressed through a graded mesh screen (150 mesh, Sigma-Aldrich) and the flow through was washed extensively with HBSS (Life Technologies). After spinning down, the supernatants were discarded and the pellets digested again with digestion buffer at 37 °C for 20 min. During this incubation period the suspensions were sheared with a 27-G needle. Erythrocytes were lysed with NH4Cl 0.8%. Single-cell suspensions were fixed with 1% PFA for 1 h at RT and with 70% ethanol overnight. Then cells were stained as described elsewhere[45]. Incubation with anti-DsRed pAb (632496, dilution 1:25, Clontech, Mountain View, CA, USA) or isotype control was followed by incubation with Alexa Fluor 647 goat anti-rabbit as secondary antibody to detect mCherry+ cells, whereas for detection of mVenus+ cells was used an anti-GFP-488 pAb (A21311, dilution 1:100, Life Technologies). Cells were, then, incubated with DAPI to perform the DNA content analysis. In order to verify if similar results were obtained after depletion of AQP2+ cells, Pax8/FUCCI2 mouse kidneys (n = 3) were processed as above and cells were fixed with 1% PFA for 1 h at RT. An aliquot of cells was incubated with anti-AQP2 pAb (bs-4611R, dilution 1:20, Bioss Inc., Woburn, Massachusetts) or with isotype control and with Alexa Fluor 647 goat anti-rabbit as secondary antibody (Life Technologies). The remaining part of the cells were incubated with anti-AQP2 pAb (bs-4611R, dilution 1:20, Bioss Inc.) followed by anti-rabbit MicroBeads (Miltenyi Biotec S.r.l.), then passed through MS columns (Miltenyi Biotec S.r.l.) accordingly with the manufacturer's protocol and collecting the first flow-through containing unlabeled cells in order to perform a magnetic cell depletion of AQP2+ cells. A small part of both the depleted- or not-depleted AQP2 cell fractions were stained with AQP2 pAb (bs-4611R, dilution 1:20, Bioss Inc.) followed by an Alexa Fluor 647 goat anti-rabbit as secondary antibody and then analyzed by Cytoflex S instrument (Beckman Coulter, Brea, CA) with CytExpert software. AQP2-depleted cell fraction was fixed with 70% ethanol overnight and the day after was stained with anti-DsRed pAb (632496, dilution 1:25, Clontech) and anti-GFP pAb (A21311, dilution 1: 100, Life Technologies) as above to evaluate mCherry+ cells and mVenus+ cells. More than 50,000 events were analysed in each experiment by using a FACS LSRII instrument (BD Biosciences, San Jose, CA) with the FACSDiva software or by using Cytoflex S instrument (Beckman Coulter) with CytExpert software. More than 50,000 events were analysed in the experiments with Pax8/FUCCI2aR and Pax2/FUCCI2aR mice by using a FACS LSRII instrument (BD Biosciences) and a MacsQuant instrument (Miltenyi Biotec S.r.l.). The assessment of the total number of Pax8/FUCCI2aR and Pax2/FUCCI2aR cells was performed by using a MacsQuant instrument (Miltenyi Biotec S.r.l.). Alexa Fluor 647 secondary antibody was excited by a 633 nm laser line, GFP pAb was excited by a 488 nm laser line, DAPI was excited by a UV laser at 355 nm.

**Tissues**. A total of 5 healthy kidneys (4 men and 1 woman, mean age 56.2 ± 11) and 10 renal biopsies from patients (mean age 43.7 ± 7.2) that developed CKD after AKI were analyzed in agreement with the Ethical Committee on human experimentation of the Azienda Ospedaliero-Universitaria Careggi and of the Meyer Children's University Hospital, Florence, Italy. Normal kidney fragments were obtained from the pole opposite to the tumor of patients who underwent nephrectomy for localized renal tumors. Formal consents were obtained by the donors or relatives. The AKI stage was classified according to KDIGO Guidelines[46].

The list of patients with CKD after AKI analyzed in this study is reported in Table 1.

**Fluorescence in situ hybridization analysis**. Fluorescence in situ hybridization analysis was performed using Whole Chromosome Painting Probe for Y chromosome (FITC) (Cytocell Ltd, Tarrytown, NY, USA) following the manufacturer's instructions in combination with Phalloidin-546 (A22283, dilution 1:40, Life Technologies).

**Study approval**. Animal experiments were approved by the Institutional Review Board and by the Italian Ministry of Health and by the Ethical Committee of the "Regierung von Oberbayern" and performed in accordance with institutional, regional, and state guidelines and in adherence to the National Institutes of Health Guide for the Care and Use of Laboratory Animals. Mice were housed in a specific pathogen-free facility with free access to chow and water and a 12-h day/night cycle.

**Statistical analysis**. The results were expressed as mean ± SEM. Comparison between groups was performed by the Mann–Whitney $U$-test, or through the analysis of variance for multiple comparisons (ANOVA for repeated measures) with Tukey post hoc analysis. $p < 0.05$ was considered to be statistically significant.

**Data availability**. The authors declare that all data supporting the findings of this study are available within the article and its Supplementary Information Files or from the corresponding author upon reasonable request.

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

## Acknowledgements

This project has received funding from the European Research Council (ERC) under the European Union's Horizon 2020 research and innovation programme (grant agreement N° 648274) to P.R.. H.-J.A. is supported by the Deutsche Forschungsgemeinschaft (AN372/11-2, 17-1, 23-1, and 24-1) and the BMBF (REPLACE-AKI 031L0071).

## Author contributions

P.R., with the help of E.L., as well as M.L.A. conceived of the study. E.L., M.L.A., F.B. and C.C. designed and performed immunofluorescence and confocal microscopy. E.L., A.P., L.M., C.C., M.E.M., L.D.C. and F.A. designed and performed flow cytometry and cell cycle analysis. A.P., C.C., G.A., E.R., D.L., S.N. and M.E.M designed and performed experiments in the mouse systems. B.M., R.R.G. and A.S. designed and performed in vitro experiments. J.A.M. and H.-J.A. performed GFR analysis. A.B. and B.S. provided the PAX2.rtTA mice. L.L., M.L.A., E.L and P.R. analyzed the data. L.L., M.L.A., E.L. and H.-J.A. critically revised the manuscript. P.R. wrote the paper with input from all authors.

## Additional information

**Competing interests:** The authors declare no competing interests.

