## [Peer Review File · Nature Communications]

Reviewers' comments:

Reviewer #1 (Remarks to the Author):

This is a robust study and paper from one of the leading research groups in the field of kidney regeneration, which addressed cellular and molecular mechanisms of recovery after acute kidney injury. The present work appears to be a logical continuation of the recent study from the same group (Stem Cell Reports 2015), another milestone paper that described Pax2+ progenitor cell-mediated glomerular regeneration. Here, the focus is on the tubular system, on the recovery and regeneration of renal tubular epithelial cells (TECs). Using a variety of experimental approaches including two, clinically relevant kidney injury models in mice and several state-of-the-art progenitor cell, cell fate tracking, and cell cycle analysis methods, the authors made several paradigm-shifting discoveries that are expected to make a significant impact in the nephrology research field.

In contrast to previous work, the present study found substantial and permanent loss of TECs after AKI, with only a small subset (Pax2+ progenitor cells) truly undergoing cell division and clonal expansion. Relevant to potential future therapeutic development, Pax2+ progenitor cell-mediated tubular regeneration was amplified by certain drugs (HDAC inhibitors). Surviving, differentiated TECs were shown to undergo endoreplication (endocycle) and hypertrophy, which was likely critically important in the recovery of overall kidney function. Importantly, the pre-clinical mouse data were successfully translated to the human kidney and disease (AKI and CKD).

This comprehensive and translational study was apparently very carefully designed, and despite the high level of complexity the data are clearly presented, and the paper is easy to follow. The data analysis and statistics are appropriate, very detailed, robust, and to be convincing, the huge datasets were analyzed by multiple individuals independently. Since the results and conclusions are based on multiple, sophisticated, and superior experimental approaches that have been well controlled, the data are compelling and strong even if most results are in disagreement with those in the previous literature. Clearly, these paradigm-shifting new discoveries will be of high impact for the field. I have no major issues, but only a few minor concerns and suggestions to further improve this excellent paper.

1. Appropriately for AKI, most experiments focus on the S3 proximal tubule segment. However, in several places throughout the paper the term "TECs" is used which refers to renal tubular cells of all tubule segments. Sometimes these terms are used interchangeably which could be misleading for a few readers. Precise identification of the renal tubule segments in each figure, for the actual and particular dataset, would be helpful and important. Most Pax2Confetti+ cells after AKI (or TSA and 4-PBA) appear to belong to the loop of Henle (DTL is also AQP1+) and collecting duct in Figs. 3-4, rather than proximal tubules. Also, the Y chromosome probe-positive tubule segments shown in Fig. 7d-e do not appear to be proximal tubules (unusual phalloidin staining with no brush border), but rather distal nephron-collecting ducts.

2. In addition to providing important and novel mechanistic insights for the in vivo cell biology of TECs (the true or "pseudo"-proliferation of TECs), the new data help reveal important technical issues of widely used cell proliferation markers. One drawback of the FUCCI2 R26R lines is that the fluorescence signal is lost during half of the cell cycle. This complicates the tracking of individual cells, which cannot be detected during their "dark" phase. Consistent with this scenario, only a few cells are shown FUCCI positive in Fig. 5. However, this problem can be overcome by using the newer FUCCI 2aR design, which yields higher expression and produces iso-stoichiometric quantities of both FUCCI probes. This newer design has the advantage that both probes are always expressed in the same ratio, which simplifies the detection of green-red transitions.

3. Although each of the applied cell cycle markers have specificity issues, the combined use of p-H3, FUCCI2 system, ploidy analysis, CDK4, and Y chromosome probe provides compelling evidence for the presence of endocycling cells, and their relevance to AKI-CKD.

4. Because of the apparent similarities between Pax2+ and Sox9+ cells that are scattered throughout the tubular system, it would be important and fair to the authors to cite the recent study by Kumar et al (Cell Rep. 2015 PMID: 26279573) either in the introduction or discussion.

Reviewer #2 (Remarks to the Author):

As summarized in the abstract, authors challenge several paradigms regarding tubular cell regeneration after ischemic tubular cell injury in mice: Capacity of tubular cells to regenerate is low, and cellular losses are higher. Widespread proliferation of surviving tubular cells results from a misinterpretation of the methods used and may reflect only cell-cycle entry but not complete cellular divisions (cytokinesis). Clonal divisions occur in only a small subset of tubular cells, the remaining tubular cells undergo only cellular hypertrophy.

Overall this study addresses important and timely questions and makes an effort to challenge several paradigms. However, the study is still limited by a number of methodological issues and the data is not sufficient to support author's conclusions.

Major points to consider

1. The title of the first finding in Results is misleading or at least based on over-interpretation of the data. Irreversible loss of tubular cells is not confirmed. GFR recovery is also not clearly demonstrated. More specific comments are given below and fundamental questions are also raised.
2. The method of quantification for tubular cells should be more clearly described. Given that a large part of the data analysed relies on these analyses, more reliable methods of quantification should be applied (Walton et al. AJP renal 2016).
3. The definition of "new cells" is unclear. Adjacent cells of the same colour share an origin, but which ones are new/old is impossible to determine. The authors should moderate their comments to reflect the findings.
4. Fig 3 b)-d): If Pax2+ cells were progenitor cells, I would expect more tubular cross-sections labelled continuously in just one colour indicating clonal expansion. The labelling pattern looks the same as within Pax8-rtTA mice. Please explain.
5. The use of markers (i.e. Aquaporin) for quantification in scenarios of proximal tubular injury is common practice. However, it is important to remember that cellular injury can certainly lead to transient down-regulation of markers and therefore may have a direct effect on cell numbers. The possibility of transient down-regulation of markers should be considered, tested and discussed. Co-stainings of Pax8-rtTA mice with AQP1 in healthy, sham operated and IRI kidneys at time point T30 would be appreciated. AQP1 staining looks surprisingly heterogeneous after regeneration should be completed. Might this indicate that AQP1 may not be the best marker to count cell loss? Have the authors considered that its expression may have decreased transiently but the cells may not be lost (Fig. 2g,h)? Similar points apply for AQP2 (Fig 2 i-l).
6. The use of the word replace is not clear or is at the very least debatable. In order to claim a cell deficit, proper quantification methods should be applied. At the moment, the data fails to provide sufficient evidence of persistent cell depletion. Furthermore, the authors present percentages in multiple graphs showing 20-35% of cell loss. In multiple analyses, the authors refer to increases in the order of 50% of the 20-35% meaning real increases of 10-18% - for such small differences to be identified properly and confidently, better quantification methods are needed.
7. The use of the word progenitor for PAX2+ cells is debatable. The data presented here does confirm their progenitor state. Please rephrase.
8. One of the main findings of this study is: "Tubular regeneration after AKI is selectively mediated

by PAX2+ progenitor mitosis". The respective experiment confirms that PAX2+ cells are more sensitive to drugs that facilitate mitosis during AKI. If PAX2+ cells are indeed progenitors, why were these drugs employed to show a difference? Did the authors see a difference without the use of these drugs?

Discussion

9. The first two sentences of the first paragraph: The statements presented in this first paragraph are exaggerated or over-interpretations of the actual data.

10. Side-by-side publications by Berger et al. and Kusaba et al. report the potential of tubular cells to acquire the phenotype of scattered tubular cells upon tubular injury. This possibility should be discussed, or at least considered within the interpretation of the data.

Minor

1. Please state whether male or female mice were used (female mice often are more resistant to the ischemia/reperfusion injury).

2. Kidney weights should be normalized to the body weights of the mice (Fig.1g)

3. GFR and kidney weights of post-ischemic mice were available up to 70 days after injury. Why were these time points omitted in the PAS stained or immunofluorescence pictures? (Fig. 1)

4. 5 week-old mice are still growing and analysis of cell numbers and volumes require age-matched controls. The experiments in this study were not designed this way. Therefore age-related bias cannot be ruled out.

5. The use of the contralateral kidney as an internal control or an age-matched control is if anything controversial. The ischaemia of one kidney can have significant effects on the other kidney.

6. Page 5, line 108: "The other 50% remains irreversibly lost." Have the authors excluded the alternative explanation that induction of Pax8 reporter expression may be lowered by giving lower amounts of doxycycline to the mice to create lower recombination rates? Also, as stated above, the use of AQP1 and AQP2 may not be appropriate to measure potential cell loss. In combination with decreased expression of the reporter genes it is not possible to give any evidence that the differentiated tubular cells are not able to repopulate the tubules.

7. Supplemental Figures 3L and 3M show identical images, please correct.

8. It would be interesting to see whether KIM-1 levels in Pax2 mice are comparable to Pax8 mice as well as if both cell populations express KIM-1 similarly. Where there also GFR and BUN level measurements done for Pax2-rtTA mice?

9. Fig. 4: Could it be possible that j) and k) are higher magnifications of panel i)? This should be mentioned. While the staining pattern doesn't look exactly the same this could be serial sections and just the same areas were chosen for the paper, maybe other areas of the kidney are also representative.

10. Cell surface area does not confirm cellular hypertrophy, please rephrase/tone down.

11. Fig. 5: Where the same evaluations of Ki67 and the FUCCI reporters also done for Pax2-rtTA mice? (a-h)

Ki67 is generally believed to label all cells re-entering the cell cycle, that's why the standard method used to evaluate proliferation in animal models is BrdU. In addition, PCNA is often used as proliferation marker. It should be possible to evaluate these markers as well within Pax8/Pax2-FUCCI mice.

Pax8/FUCCI mice show virtually global mCherry expression at T0. Some of these cells should also be Ki67 while they are in G1. Is there still expression of the reporter even if the cells are in G0?

In j/j', there is one cell clearly undergoing cellular division (two nuclei shaped in a way that reminds at dividing spindle apparatuses). Both cells are positive for p-H3 and mCherry, but not mVenus, please explain.

12. Fig. 6: Please add the confocal microscopy pictures (used for the evaluation in n) to the figure or the supplements.

13. Fig. 7: Why was the Y-chromosome FISH not also used for the Pax8/Pax2-rtTA mice to

evaluate hypertrophy and endoreplication of the TECs and the putative progenitor cells?
Co-stainings with tubular compartment markers (LTA, AQP1, etc.) would be recommended. In a) and b) it is not clear which structures are shown.

11. Table 1 only has data of cases of CKD after AKI, but not controls. Demographics and clinical data are important to assess the suitability of these subjects as controls.

12. Comment for all Figures. Pie charts with % of cell populations can be reported in the text and do not need to be part of the Figures.

13. Regarding measurement of GFR, the two-compartment model for FITC-sinistrin introduces unaccounted bias into the changes into t-half life. A recent publication by Friedemann et al (Kidney International 2016) shows that a four-compartment model is more reliable.

Reviewer #3 (Remarks to the Author):

In this paper, Lazzeri and colleagues apply several genetic models to evaluate epithelial repair after acute injury. Using Pax8-rtta; confetti mice, they label the majority of tubular epithelia. They perform clonal analysis, finding a majority of two-cell clones, but rare larger clones of 10 cells, which is consistent with published clonal analyses (Rinkevich et al., Kusaba et al.). They then use a Pax2-rtta and confetti mouse. Here, they claim expression is found in proximal tubule but no convincing evidence is shown to support this claim. Pax2 is known to be expressed in collecting duct, and it is not clear what attempts were made to exclude collecting duct from the analysis of the medulla – since a large proportion of medullary tubules are collecting duct. The authors suggest that only a subset of epithelial cells proliferate, with the rest becoming polyploidy and becoming arrested.

Overall, the lineage tracing experiments are not convincing with many questions about what tubule segments are actually labeled, and no attempts made to correlate genetic labeling with endogenous Pax2 expression.

1. Unilateral IRI is a fibrotic model with ongoing inflammation, so the loss of tubular cells is expected, and not unexpected. Indeed this has already been published (Endo et al., Am J Path, 2015).

2. The Pax2 experiments suffer from a number of major limitations. Unfortunately, there is no convincing labeling of any proximal tubule cells in this mouse. AQP1 is used as a marker, but as the authors point out in Supplementary Figure S3e, AQP1 is also expressed in loop of henle. These are the cells being labeled, which is very clear from the images in Figure 3a and b. There is a complete absence of confetti marking in the larger radius proximal tubule, with only very sparse cortical labeling at all – and these are all cortical collecting ducts. Therefore, these results do not apply to proximal tubule, the major site of injury and proliferation in AKI.

3. No costaining for endogenous Pax2 was performed to validate the Pax2-rtta model and its faithfulness to the endogenous allele. Indeed, two separate reports indicate that in adult mouse kidney, Pax2 expression is limited to collecting duct: Imgrund M, Grone E, Grone HJ et al. Re-expression of the development gene Pax-2 during experimental acute tubular necrosis in mice. Kidney Int 1999; Lindoso RS, Verdoorn KS, Einicker-Lamas M. Renal recovery after injury: the role of Pax-2. Nephrol Dial Transplant. 2009.

We thank reviewer 1 for the enthusiastic and encouraging comments.

Minor points:

1. *Appropriately for AKI, most experiments focus on the S3 proximal tubule segment. However, in several places throughout the paper the term “TECs” is used which refers to renal tubular cells of all tubule segments. Sometimes these terms are used interchangeably which could be misleading for a few readers. Precise identification of the renal tubule segments in each figure, for the actual and particular dataset, would be helpful and important. Most Pax2Confetti+ cells after AKI (or TSA and 4-PBA) appear to belong to the loop of Henle (DTL is also AQP1+) and collecting duct in Figs. 3-4, rather than proximal tubules. Also, the Y chromosome probe-positive tubule segments shown in Fig. 7d-e do not appear to be proximal tubules (unusual phalloidin staining with no brush border), but rather distal nephron-collecting ducts.*

As requested by the reviewer, we now expanded the previous tissue analysis by adding also labelling with the proximal tubule marker megalin, that does not stain thin descending limbs of the loop of Henle. We also more clearly specify each tubule segment in each Figure and added specification of the tubule segment to the word TEC throughout the manuscript. We have also added colabelling of the proximal tubule marker LTA with p-H3 and CDK4 to identify endocycling cells specifically on proximal tubule in human biopsies in new Fig. 8. These co-labelling studies now clearly show that endocycle occurs mostly in the S2 segment of the proximal tubule inside the renal cortex (see new Fig. 8 and page 10, lines 248-254).

2. *In addition to providing important and novel mechanistic insights for the in vivo cell biology of TECs (the true or “pseudo”-proliferation of TECs), the new data help reveal important technical issues of widely used cell proliferation markers. One drawback of the FUCCI2 R26R lines is that the fluorescence signal is lost during half of the cell cycle. This complicates the tracking of individual cells, which cannot be detected during their “dark” phase. Consistent with this scenario, only a few cells are shown FUCCI positive in Fig. 5. However, this problem can be overcome by using the newer FUCCI 2aR design, which yields higher expression and produces iso-stoichiometric quantities of both FUCCI probes. This newer design has the advantage that both probes are always expressed in the same ratio, which simplifies the detection of green–red transitions.*

We thank the reviewer for this excellent suggestion, that allowed us to confirm all the data and extend them through direct quantitation of the number of total Pax8+ and Pax2+ cells as well as of Pax8+ and Pax2+ endocycling or dividing cells. To address the reviewer's concern we conducted a new set of experiments by using two further transgenic mouse lines, based on the FUCCI2aR reporter, producing iso-stoichiometric quantities of both FUCCI probes, making it possible to directly trace and quantify the number of labelled cells by MacsQuant flow cytometry. In this manner, we could provide an ultimate proof that Pax2+ cells are the only tubular cells undergoing mitosis after AKI. (see new Fig. 7 and page 9, lines 202-226).

3. *Although each of the applied cell cycle markers have specificity issues, the combined use of p-H3, FUCCI2 system, ploidy analysis, CDK4, and Y chromosome probe provides compelling evidence for the presence of endocycling cells, and their relevance to AKI-CKD.*

We thank the reviewer for this comment. We now prove the existence of endocycle also with the FUCCI2aR cell line, following the reviewer suggestion (see new Fig. 7 and page 10, lines 229-238).

4. *Because of the apparent similarities between Pax2+ and Sox9+ cells that are scattered throughout the tubular system, it would be important and fair to the authors to cite the recent study by Kumar et al (Cell Rep. 2015 PMID: 26279573) either in the introduction or discussion.*

We have added the recent study by Kumar et al in the discussion (see new ref. 30).

1. *The title of the first finding in Results is misleading or at least based on over-interpretation of the data. Irreversible loss of tubular cells is not confirmed. GFR recovery is also not clearly demonstrated. More specific comments are given below and fundamental questions are also raised.*

We thank the reviewer for raising these important points. In order to address the reviewer's concern we now provide an unbiased quantitation of the absolute remnant tubular cell numbers 30 days after AKI by automated counting using MacsQuant-based flow cytometry and two new transgenic mouse lines, based on the FUCCI2aR reporter. This represents an advancement of the FUCCI2 reporter which produces stoichiometric quantities of both FUCCI probes and labels all the cells with high sensitivity. These new results directly prove persistent tubular cell loss 30 days after IRI, as well as that Pax2+ cells are the only tubular cells that complete mitosis after AKI. These results were added in new Fig. 7 (see page 9, lines 202-226). In addition, GFR recovery is now demonstrated based on a re-analysis of all the data using the new software MPD Studio software ver.RC6 (MediBeacon GmbH Cubex41, Mannheim, Germany), that applies the four-compartment model reported in Friedemann et al (Kidney International 2016), as suggested by the reviewer.

2. *The method of quantification for tubular cells should be more clearly described. Given that a large part of the data analysed relies on these analyses, more reliable methods of quantification should be applied (Walton et al. AJP renal 2016).*

As reported also for the response to point 1, we have added a new direct method of automated quantification of absolute numbers of tubular cells using MacsQuant-based flow cytometry. These results could ultimately prove the tubular cell loss at day 30 and confirm the finding already reported with the Confetti methods (see new Fig. 7 and page 9, lines 202-226).

3. *The definition of "new cells" is unclear. Adjacent cells of the same colour share an origin, but which ones are new/old is impossible to determine. The authors should moderate their comments to reflect the findings.*

As suggested by the reviewer, the definition of new cells was better specified in the text (see Methods section, page 23, lines 545-547). We did not mean to specify which cells are old or new but only that all the cells included in clones that are longer than those observed at time 0 (when all the labelled cells are single and only rare in doublets) can indeed be considered new.

4. *Fig 3 b)-d): If Pax2+ cells were progenitor cells, I would expect more tubular cross-sections labelled continuously in just one colour indicating clonal expansion. The labelling pattern looks the same as within Pax8-rtTA mice. Please explain.*

We agree with this observation and now Figures were replaced by others clearly showing better tubular cross-sections labelled continuously in just one colour indicating clonal expansion and localized in different segments of the tubule (see new Fig. 4). More importantly, we performed 3D-reconstruction of clones and tubule segments. This analysis, demonstrated that Pax2+ cells generate long clones, of up to 98 cells, and

regenerate whole tubule segments (see new Fig. 4 and the 3D Video). Thus, when Pax8 mice are treated with low doses of doxycycline, it is not surprising that they appear similar to Pax2 mice because Pax8+ cells also include Pax2+ cells, suggesting the clones observed in the two models are the same clones. This is now directly confirmed by the experiments with the Fucci2aR transgenic line reported in new Fig. 7, in which we demonstrate that Pax2+ cells are the only TECs that divide after AKI.

5. *The use of markers (i.e. Aquaporin) for quantification in scenarios of proximal tubular injury is common practice. However, it is important to remember that cellular injury can certainly lead to transient down-regulation of markers and therefore may have a direct effect on cell numbers. The possibility of transient down-regulation of markers should be considered, tested and discussed. Co-stainings of Pax8-rtTA mice with AQP1 in healthy, sham operated and IRI kidneys at time point T30 would be appreciated. AQP1 staining looks surprisingly heterogeneous after regeneration should be completed. Might this indicate that AQP1 may not be the best marker to count cell loss? Have the authors considered that its expression may have decreased transiently but the cells may not be lost (Fig. 2g,h)? Similar points apply for AQP2 (Fig 2 i-l).*

We had considered the possibility that expression of AQP1 may have decreased transiently but the cells may not be lost. This is why first of all we counted the total number of Pax8/Confetti-labelled cells in absence of co-labelling with whatever tubular marker. These results were shown in old Fig. 1k as well as in new Fig. 1d, and demonstrate a significant decrease of total tubular cells at day 30 after IRI from 526.13 ± 16.3 to 365.8 ± 14 per field. As the major technical advance over previously used analytical tools lineage tracing excludes subsequent label alterations by tissue injury and persistently marks the labeled cell type as well as its progeny beyond gene regulation, as it e.g. can occur with immunolabeling. These results show a residual loss at day 30 of $30.5 \pm 2.8\%$ of TECs in the unilateral IRI model and of $27.6 \pm 12\%$ in the nephrotoxic AKI model, consistent with results shown in old Fig. 2 and counted on AQP1+ or on AQP2- cells (see new Fig. 1d, g, h and page 4, lines 77-81). In addition, we now further confirm these results with the Fucci2aR transgenic cell line, as already discussed above (see new Fig. 7 and page 9, lines 202-226). All the four methods used indeed, give consistent results and therefore ultimately confirm the relevant tubular cell loss one month after AKI. In addition, as requested by the reviewer, co-stainings of Pax8-rtTA mice with AQP1 in healthy, sham-operated and IRI kidneys at time point T30 were added and are presented in Supplementary Fig. 1g-i.

6. *The use of the word replace is not clear or is at the very least debatable. In order to claim a cell deficit, proper quantification methods should be applied. At the moment, the data fails to provide sufficient evidence of persistent cell depletion. Furthermore, the authors present percentages in multiple graphs showing 20-35% of cell loss. In multiple analyses, the authors refer to increases in the order of 50% of the 20-35% meaning real increases of 10-18% - for such small differences to be identified properly and confidently, better quantification methods are needed.*

As already discussed in the response to point 2, a novel quantification methods was added to the study (see new Fig. 7). The results obtained confirm those already reported with the Confetti transgenic mice (Fig.1, 3).

7. *The use of the word progenitor for PAX2+ cells is debatable. The data presented here does not confirm their progenitor state. Please rephrase.*

As requested we reduced the use of the term progenitor in the manuscript. However, the new results added provide direct proof that Pax2+ cells are the only tubular cells within the nephron that divide and self-renew in response to injury (see new Fig. 7). Self-renewal is now demonstrated also by 3D analysis that revealed regeneration of entire tubule segments by single Pax2+ cells (see new Fig. 4). Finally, the results in Fig. 2i-l show that they represent an homeostatically stable population and not a functional state. Taken altogether, these results directly prove that Pax2+ cells localized within the nephron display progenitor properties as explained in the Discussion section (see page 12, lines 280-285 and page 13, lines 286-287).

8. *One of the main findings of this study is: “Tubular regeneration after AKI is selectively mediated by PAX2+ progenitor mitosis”. The respective experiment confirms that PAX2+ cells are more sensitive to drugs that facilitate mitosis during AKI. If PAX2+ cells are indeed progenitors, why were these drugs employed to show a difference? Did the authors see a difference without the use of these drugs?*

We now provide direct proof that tubular regeneration after AKI is selectively mediated by Pax2+ cells mitosis even in absence of drugs by combining flow cytometry and automated cell quantitation by MacsQuant using two new transgenic cell lines and the FUCCI2aR reporter (Mort RL, et al. Cell Cycle 2014). This represents an advancement of the FUCCI2 model, that avoids the problem of having a quote of black cells among M and G1 phase of cell cycle (Sakaue-Sawano et al. Cell, 2008), as also underlined by reviewer 1, permitting tracing as well as quantitation of absolute numbers of total TECs and Pax2+ cells. These results directly proof that at all time points analysed, Pax2+ cells are the only tubular cells that divide after AKI.

9. *The first two sentences of the first paragraph: The statements presented in this first paragraph are exaggerated or over-interpretations of the actual data.*

Following the request of the reviewer we have carefully adapted the first sentences of the first paragraph of discussion as suggested.

10. *Side-by-side publications by Berger et al. and Kusaba et al. report the potential of tubular cells to acquire the phenotype of scattered tubular cells upon tubular injury. This possibility should be discussed, or at least considered within the interpretation of the data.*

We added in the discussion the possibility that some tubular cells may acquire the phenotype of scattered tubular cells upon injury (see page 12, lines 275-276). However, to test whether Pax2+ cells represent a distinct, predefined and stable TEC population or rather a transient functional state, we had compared two groups of mice, one that was induced with doxycycline for 10 days, then washed out for one week and tracked for the following 30 days, and another that was permanently exposed to doxycycline for an identical period of time (see old Supplementary Fig. 3 f-i, now Fig. 2i-l). Indeed, if the labelled Pax2+ cells after 10 days of doxycycline induction would represent a functional state that can be randomly acquired by TECs in healthy mice, their number should considerably increase when the length of exposure to doxycycline gets at least three times longer. However, no significant difference in Pax2+ TEC number was observed. Thus, these

experiments exclude this possibility and clearly identify Pax2⁺ cells as an homeostatically stable distinct population of tubular cells.

Minor

1. *Please state whether male or female mice were used (female mice often are more resistant to the ischemia/reperfusion injury).*

Only male mice were used for ischemia-reperfusion, and only female mice for nephrotoxic injury, as we had specified in the methods section (see page 19, line 433 and page 19, line 452).
2. *Kidney weights should be normalized to the body weights of the mice (Fig.1g)*

Kidney weight was normalized to the body weight of the mice (see new Supplementary Fig. 1f).
3. *GFR and kidney weights of post-ischemic mice were available up to 70 days after injury. Why were these time points omitted in the PAS stained or immunofluorescence pictures? (Fig. 1)*

In the previous version of the manuscript, GFR measurement data and kidney weights shown in old Fig. 1a-h were obtained in C57BL/6 mice at time points up to 70 days, in order to optimize the unilateral ischemic injury, as clearly stated in old method section. Based on the fact that all the analysed parameters were similar at day 35 and 70 we chose the 30 days time point for the experiments with transgenic mice. Following the reviewer's comment, in the revised manuscript the results obtained in C57BL/6 mice were removed and all the new data of GFR (Fig.1a, Fig.3a), BUN (Supplementary Fig.1e, Supplementary Fig.3b), kidney weights (Supplementary Fig. 1f) and tubular score (Supplementary Fig.1d) are now shown in Pax8/Confetti and Pax2/Confetti mice and are harmonized at the 30 days time point.
4. *5 week-old mice are still growing and analysis of cell numbers and volumes require age-matched controls. The experiments in this study were not designed this way. Therefore age-related bias cannot be ruled out. Age-matched controls were shown in old Fig. 3e and indicate no significant differences with day 0, that's why we can rule out age-related bias.*
5. *The use of the contralateral kidney as an internal control or an age-matched control is if anything controversial. The ischaemia of one kidney can have significant effects on the other kidney.*

The contralateral kidney was removed and only time 0 and age-matched controls are now included (see new Fig 1j, Fig. 3g).
6. *Page 5, line 108: "The other 50% remains irreversibly lost." Have the authors excluded the alternative explanation that induction of Pax8 reporter expression may be lowered by giving lower amounts of doxycycline to the mice to create lower recombination rates? Also, as stated above, the use of AQP1 and AQP2 may not be appropriate to measure potential cell loss. In combination with decreased expression of the*

reporter genes it is not possible to give any evidence that the differentiated tubular cells are not able to repopulate the tubules.

As already pointed out, the results obtained with the Confetti transgenic cell lines were confirmed with an unbiased quantitation of the absolute remnant tubular cell numbers 30 days after AKI by automated counting using MacsQuant and two new transgenic mouse lines, based on the Fucci2aR reporter that do not have the possible bias reported by the authors (see new Fig. 7).

7. *Supplemental Figures 3L and 3M show identical images, please correct.*

We apologize for the mistake, Supplemental Fig. 3l was replaced with the right one (see new Supplementary Fig. 2b).

8. *It would be interesting to see whether KIM-1 levels in Pax2 mice are comparable to Pax8 mice as well as if both cell populations express KIM-1 similarly. Where there also GFR and BUN level measurements done for Pax2-rtTA mice?*

KIM-1 protein was assessed in Pax8 and Pax2 mice. Results are now reported in Supplementary Fig. 7i, m. Interestingly, these results show that KIM-1 expression is largely observed in Pax8+ but not in Pax2+ cells (see page 9, lines 226-228). GFR and BUN measurement data for Pax2 mice were reported in Fig. 3a and in Supplementary Fig. 3b, respectively.

9. *Fig. 4: Could it be possible that j) and k) are higher magnifications of panel i)? This should be mentioned. While the staining pattern doesn't look exactly the same this could be serial sections and just the same areas were chosen for the paper, maybe other areas of the kidney are also representative.*

We added another representative area of the kidney in Fig. 5i.

10. *Cell surface area does not confirm cellular hypertrophy, please rephrase/tone down.*

The sentence was toned down.

11. *Fig. 5: Where the same evaluations of Ki67 and the Fucci reporters also done for Pax2-rtTA mice? (a-h). Ki67 is generally believed to label all cells re-entering the cell cycle, that's why the standard method used to evaluate proliferation in animal models is BrdU. In addition, PCNA is often used as proliferation marker. It should be possible to evaluate these markers as well within Pax8/Pax2-Fucci mice. Pax8/Fucci mice show virtually global mCherry expression at T0. Some of these cells should also be Ki67 while they are in G1. Is there still expression of the reporter even if the cells are in G0? In j/j', there is one cell clearly undergoing cellular division (two nuclei shaped in a way that reminds at dividing spindle apparatuses). Both cells are positive for p-H3 and mCherry, but not mVenus, please explain.*

Evaluations of Fucci2 reporters were also done for Pax2-rtTA mice and added in new Fig. 6 m-p. Analysis of PCNA was also added (see new Fig. 6 e-h). Although PCNA cannot be combined with the immunolabelling for mCherry and mVenus for incompatibility of fixation procedures (while this was instead possible for p-H3), the results obtained with PCNA, as well as with Ki-67, confirm that both cell cycle

markers largely overestimate dividing cells in comparison to FUCCI mice (see Fig. 6 page 7, lines 156-173, page 8, lines 174-181). As also underlined by reviewer 1, the FUCCI2 reporter has a black phase when the cell goes out of mitosis (...“the green fluorescence disappeared rapidly in late M phase and the red fluorescence became detectable in early G1 phase, causing a small gap in fluorescence in newborn daughter cells...” Sakaue-Sawano et al. Cell, 2008). However, cells in G1 are all red, even when they have not entered the cell cycle again and they are in G0 of the cell cycle phase (see Tomura M et al. PLoS One. 2013). However, we would like to underline that this problem is completely bypassed with the new transgenic FUCCI2aR cell lines that confirm all the data and also allowed direct quantitation of absolute cell numbers when combined with MacsQuant analysis (see new Fig. 7).

Finally, it is well known that proximal tubular cells are G1 arrested in healthy kidneys, explaining why they are mCherry+ (red) but do not express Ki67, that instead labels cells in G1 that are primed to cell division (see Lombardi D. et al. Nephrol. Dial. Transplant. 2015). In Fig. 6 r/r' the image may remind a mitotic cell but these are two distinct cells; the positivity for p-H3 and mCherry suggests they are endocycling cells.

12. *Fig. 6: Please add the confocal microscopy pictures (used for the evaluation in n) to the figure or the supplements.*

The evaluation previously shown in old Fig. 6n was now also repeated on FUCCI2aR mice and these are the results included (see new Fig. 7u,v). Moreover, a confocal microscopy representative picture with the mimic of procedure applied to measure the cell surface area is now shown in new Fig. 7u.

13. *Fig. 7: Why was the Y-chromosome FISH not also used for the Pax8/Pax2-rtTA mice to evaluate hypertrophy and endoreplication of the TECs and the putative progenitor cells? Co-stainings with tubular compartment markers (LTA, AQP1, etc.) would be recommended. In a) and b) it is not clear which structures are shown.*

We made efforts to use the Y-chromosome FISH also in Pax8/Pax2-rtTa mice to evaluate endoreplication but could not succeed because the tissue treatments required to allow access of the probe to the nuclear DNA are not compatible with the fluorescent reporter. Following reviewer's suggestion, co-stainings with tubular compartment markers LTA and AQP1 were added and now proximal tubular cells are clearly shown (see new Fig. 4, 7 and 8).

14. *Table 1 only has data of cases of CKD after AKI, but not controls. Demographics and clinical data are important to assess the suitability of these subjects as controls.*

Demographics and clinical data of controls were added to the Table 1.

15. *Comment for all Figures. Pie charts with % of cell populations can be reported in the text and do not need to be part of the Figures.*

Pie charts with % of cell populations were removed from the Figures and the number reported in the text.

16. *Regarding measurement of GFR, the two-compartment model for FITC-sinistrin introduces unaccounted bias into the changes into t-half life. A recent publication by Friedemann et al (Kidney International 2016) shows that a four-compartment model is more reliable.*

All the GFR measurement data were re-analysed using the four-compartment model reported by Friedemann et al (Kidney International 2016); see new Fig. 1, 3, 5, page 20, lines 458-472. In addition, a new experiment of GFR measurement in treated-Pax2/Confetti mice with TSA or DMSO after injury is added and now n=7 for each group of mice analysed.

1. *Unilateral IRI is a fibrotic model with ongoing inflammation, so the loss of tubular cells is expected, and not unexpected. Indeed this has already been published (Endo et al., Am J Path, 2015).*

We agree with the reviewer that Endo et al. reported tubular cell loss in a unilateral IRI model. However, they used an ischemia time of 45 minutes, that is known to cause bad fibrosis and is not considered a physiologic model of IRI. By contrast, we used an ischemia time of 30 min, that is considered as reversible (see for ex. Kusaba et al. Proc Natl Acad Sci U S A 2014, where they use two unilateral IRI models, a 26 min, considered mild, and 35 min, considered severe, but both used to demonstrate tubular regeneration). In addition, we report similar observations in the bilateral glycerol-induced AKI that is a model of nephrotoxic AKI and is also normally considered as fully reversible (see for ex. Bruno et al. J Am Soc Nephrol. 2009).

2. *The Pax2 experiments suffer from a number of major limitations. Unfortunately, there is no convincing labeling of any proximal tubule cells in this mouse. AQP1 is used as a marker, but as the authors point out in Supplementary Figure S3e, AQP1 is also expressed in loop of henle. These are the cells being labeled, which is very clear from the images in Figure 3a and b. There is a complete absence of confetti marking in the larger radius proximal tubule, with only very sparse cortical labeling at all – and these are all cortical collecting ducts. Therefore, these results do not apply to proximal tubule, the major site of injury and proliferation in AKI.*

The selective site of injury in ischemic AKI is the S3 segment of the proximal tubule that corresponds to the thick descending limb of the loop of Henle in the outer stripe of the renal medulla. In the S3 segment, Pax2+ cells represent 6-10% of tubular cells, consistently with the need of tissue regeneration specifically in that area. By contrast, as underlined by the reviewer, in the S2 segment, Pax2+ cells represent less than 2% of all tubular cells. However, the S2 segment is not directly injured in AKI, and the current assumption that it is an area of major proliferation is based only on cell cycle markers, that as clearly demonstrated in our study, are not reliable indicators of cell division. We thus now added new experiments where we performed a detailed analysis of the distribution of endocycle and mitosis in the different segments of the tubule using co-staining with megalin (that labels S1 and S2 segments of the proximal tubule), AQP1 (that labels S1, S2 and S3 segment, including thick and thin descending limbs) and Tamm-horsfall protein (THP), that labels the thick ascending limb of the Henle's loop (TAL). Our results demonstrate that TECs of the S2 segment of the proximal tubule mostly endocycle, while mitosis is instead mostly occurring in the S3 segment of the proximal tubule (Fig. 4f, and 7q-t). This is consistent with the localization within the nephron of the Pax2+ tubular cells, that are the only tubular cells that can efficiently complete mitosis, as also directly proved by new experiments provided in Fig. 7m (see also response to major points of reviewer 2). We also would like to underline that Pax2+ cells within collecting ducts, although numerous, virtually do not divide after IRI, as now clearly shown in Fig. 4f.

3. *No costaining for endogenous Pax2 was performed to validate the Pax2-rtta model and its faithfulness to the endogenous allele. Indeed, two separate reports indicate that in adult mouse kidney, Pax2 expression is*

limited to collecting duct: Imgrund M, Gröne E, Gröne HJ et al. Re-expression of the development gene Pax-2 during experimental acute tubular necrosis in mice. Kidney Int 1999; Lindoso RS, Verdoorn KS, Einicker-Lamas M. Renal recovery after injury: the role of Pax-2. Nephrol Dial Transplant. 2009.

A detailed immunostaining for Pax2 expression was performed on Pax2/Confetti mice under healthy conditions and is now added as Supplementary Fig. 2c-e. While in the collecting ducts Pax2 expression is frequent and intense (Supplementary Fig. 2e), in the tubules of the cortex is rare and weak (Supplementary Fig. 2c), and becomes more frequent in the S3 segment, i.e. the thick descending limb of the Henle's loop (Supplementary Fig. 2d). This is consistent with the labelling observed in the Pax2/Confetti mouse (Supplementary Fig. 2c-e). This explains why immunostaining in scattered proximal tubular cells in addition to collecting ducts is reported in certain studies (see for ex. Winyard PJD et al. JCI, 1996) but not in others, like the ones reported by the reviewer. Expression of Pax2 in scattered proximal tubular cells with a similar distribution to the one we describe in the Pax2/Confetti mouse is also reported in human (see for ex. Ye Y et al. Hum Pathol. 2011). These results show that the Pax2rtTa mouse model fully matches Pax2 immunostaining in healthy mice. However, we would like to underline that a full validation of the Pax2rtTa model has already been previously published (see Lasagni L et al. Stem Cell Reports 2015; Burger A. et al. Genesis 2011), including its faithfulness to the endogenous allele, and this last study was quoted in the Methods section (Burger A. et al. Genesis 2011).

Reviewers' comments:

Reviewer #1 (Remarks to the Author):

In this revised version the authors have carefully considered all major and minor concerns raised by the three reviewers, and conducted a number of additional experiments. These included establishing new triple transgenic (!) mice with the newest version of Fucci2aR design to unequivocally prove their major findings regarding endocycling and proliferating cell populations after AKI/IRI. The additional minor supplementary experiments to the older studies as well as the brand new studies with the new research tools appear to have been very carefully and properly designed and performed. The newly acquired data now provide strong proof and confirmation of the previous results and major claims by the authors. I continue to believe that these paradigm-shifting new discoveries will be of high impact for the field. I have no remaining issues with this robust and exceptionally important paper, and would likely to congratulate the authors on such an excellent study.

Sincerely,

Janos Peti-Peterdi

University of Southern California

Reviewer #2 (Remarks to the Author):

Review Lazzeri et al.

Overall, this rebuttal has not addressed the limitations outlined in the previous round sufficiently. The experimental data does to support the author's conclusions and other interpretations of the data have not been tested or ruled sufficiently.

The paper is very hard to read. One has to spend significant time and effort to search for the relevant information. E.g. when authors claim tubular cell loss in paragraph 1 of the results section one needs to know the method of quantification to be able to judge the quality and evidence of this claim.

Summary of the major findings of the study from the abstract and their shortcomings:

1. regarding: Conditional Pax8/Confetti shows substantial tub cell loss despite function recovery. my concern:

a. inappropriate method of analysis. In Fig 1D, quantification of loss of cells from AKI occurred by comparing the number of genetically labeled cells in the confetti mouse versus all cell in a histological section. This is an inappropriate method because it compares number of tubular cells vs. interstitial cells. After AKI, loss of tubular cells and interstitial fibrosis occurs. Changes in cell number within the interstitium are unpredictable: E.g. if many tubular cells were lost and very little additional interstitial cells accumulated, no loss of tubular cells would be detected. On the other hand, if large amounts of interstitial cells infiltrated a large loss of tubular cells would be detected.

It is suggested, that this prominent TEC loss is compensated mainly by hypertrophy of remaining Pax2 negative cells. These much lower numbers of cells would have to cover the same area of nephrons as previously the 100% of cells, so that they would have about twice the size (unless the interstitial matrix changes). Is this the case? Are they longer and flatter? Can this be truly reproduced in histology?

In general, the method of counting are still unclear and in part described insufficiently (e.g. what is A in the formula on P23L546 and B in the formula on P23L557)? Supplementary data examples showing the counting procedure would be helpful.

In general: How can number of cells be counted in confetti mice which show cytoplasmic reporters?

These methodological problems limit analysis of the drug intervention experiments (suppl. Fig. 5) as well.

- b. down-regulation of markers (see previous review below).
- c. altered kidney tissue integrity limits FACS analysis after acute kidney injury (see blow)
- d. In Fig 1, the IF in T30 vs T0 shows very prominent loss of red color and a large number of clusters of blue and yellow cells. This does not seem to be reflected anyhow in the quantification of clones. Furthermore, the authors conclude, that only Pax2+ cells could proliferate, how do the authors explain the increase in cell clones in Pax8 confetti mice?

2. regarding: Clonal analysis disproved widespread tub cell proliferation, erroneously indicated by proliferation markers unable to ultimately identify cell divisions.

my concerns:

- a. As shown in Figure 1j, only about 1% of all clones contained more than 6 cells after regeneration. This result would allow to conclude that regeneration does not occur from a small predefined progenitor population – nevertheless the authors conclude the opposite. I cannot judge the statistical method used, but I believe that common sense is sufficient here.
- b. Why did the authors use a higher magnification in Fig. 6f? Do they want to create the impression that proliferation markers are expressed in only a few tubular cells after injury? This might be a 'mistake', but it makes me worry and concerned.
- c. Significant polyploidy has not been observed previously in detailed and meticulous studies (see leHir et al). Did the authors consider cell-cycle arrest? This is one of the mechanisms proposed how AKI progresses to CKD, and the provided data do not exclude this possibility.
- d. Did the authors considered counting mitotic figures as the „best“ marker for proliferation? This is also possible in DAPI.

3. regarding: A small subset of Pax2+ tubular progenitors got enriched via higher stress resistance and clonal expansion, regenerating necrotic segments.

my concerns:

- a. I am unconvinced that Pax2 is a suitable marker to identify a predefined population of progenitor cells within the kidney tubule – which is a major argument of this study. I would advise the authors to visit the human protein atlas online. There, authors will see that Pax2 is expressed in virtually all tubular cells in most human biopsies. I suspect that Pax2 is a simple marker for the injury phenotype of tubular cells (scattered tubular cells), which is upregulated in any tubular cells in states of proteinuria or any other type of injury (see the work of Humphreys or Moeller). Authors have even made this observation (as stated on page 6) – but they still adhere to their interpretation. Authors need to characterize the expression of Pax2 in mouse kidney in healthy adult mice (authors used very young mice which are still growing) and after injury. In young mice, Pax2 might be expressed in tubular cells which are under mechanical stress (e.g. the inner turn of a tubule, as reported previously). It is conceivable that these cells will be the first to become STC and proliferate after injury. No need for progenitor cells here. I would predict that Pax2 positive tubular are virtually absent in normal healthy adult mouse kidney (i.e. 12 weeks of age).

5. regarding: Drugs that enhance tub reg confirmed that this occurred only through progenitor mitosis.

my concerns:

- a. Results are limited by an inappropriate method to quantify cell numbers.

regarding the point by point rebuttal:

- 1. The title of the first finding in Results is misleading or at least based on over-interpretation of the data. Irreversible loss of tubular cells is not confirmed. GFR recovery is also not clearly

demonstrated. More specific comments are given below and fundamental questions are also raised.

We thank the reviewer for raising these important points. In order to address the reviewer's concern we now provide an unbiased quantitation of the absolute remnant tubular cell numbers 30 days after AKI by automated counting using MacsQuant-based flow cytometry and two new transgenic mouse lines, based on the Fucci2aR reporter. This represents an advancement of the Fucci2 reporter which produces iso-stoichiometric quantities of both Fucci probes and labels all the cells with high sensitivity. These new results directly prove persistent tubular cell loss 30 days after IRI, as well as that Pax2+ cells are the only tubular cells that complete mitosis after AKI. These results were added in new Fig. 7 (see page 9, lines 202-226). In addition, GFR recovery is now demonstrated based on a re-analysis of all the data using the new software MPD Studio software ver.RC6 (MediBeacon GmbH Cubex41, Mannheim, Germany), that applies the four-compartment model reported in Friedemann et al (Kidney International 2016), as suggested by the reviewer.

my concerns:

How can you be sure that only Pax2+ cells complete mitosis? If the percentage of mCherry+mVenus+ Pax2+ cells is much lower than Pax8+ cells, there could also be a biased interpretation due to the very small amounts of Pax2+ cells overall. A time-dependent analysis of the cell cycle immediately after IRI with more than two time points would be needed. If there were an increase in the number of mVenus+ Pax2+ cells after IRI and a decrease up to day 30 which cannot be seen in Pax8+ cells, it would strengthen your argument. At day 30 there are still nearly half of the Pax2+ cells mVenus+, does that mean they are stuck within the S phase even after a time in which regeneration should be completed?

2. The method of quantification for tubular cells should be more clearly described. Given that a large part of the data analysed relies on these analyses, more reliable methods of quantification should be applied (Walton et al. AJP renal 2016).

As reported also for the response to point 1, we have added a new direct method of automated quantification of absolute numbers of tubular cells using MacsQuant-based flow cytometry. These results could ultimately prove the tubular cell loss at day 30 and confirm the finding already reported with the Confetti methods (see new Fig. 7 and page 9, lines 202-226).

my concerns: The new method used differs a lot from the one I suggested. On day 2 after IRI the kidney is very damaged. The tissue is very susceptible against shear forces like the mechanical disintegration through meshes. Intact but marginal cells could be lost because of the harsh isolation method. A lot of cell debris could stick at intact cells and result in false signals. An additional confirmation with the suggested method would be appreciated.

3. The definition of "new cells" is unclear. Adjacent cells of the same colour share an origin, but which ones are new/old is impossible to determine. The authors should moderate their comments to reflect the findings.

As suggested by the reviewer, the definition of new cells was better specified in the text (see Methods section, page 23, lines 545-547). We did not mean to specify which cells are old or new but only that all the cells included in clones that are longer than those observed at time 0 (when all the labelled cells are single and only rare in doublets) can indeed be considered new.

my concerns: This method will miss cells that just replicate once or twice creating bias towards longer clonogenic tubule segments.

4. Fig 3 b)-d): If Pax2+ cells were progenitor cells, I would expect more tubular cross-sections labelled continuously in just one colour indicating clonal expansion. The labelling pattern looks the

same as within Pax8-rtTA mice. Please explain.

We agree with this observation and now Figures were replaced by others clearly showing better tubular cross-sections labelled continuously in just one colour indicating clonal expansion and localized in different segments of the tubule (see new Fig. 4). More importantly, we performed 3D-reconstruction of clones and tubule segments. This analysis, demonstrated that Pax2+ cells generate long clones, of up to 98 cells, and regenerate whole tubule segments (see new Fig. 4 and the 3D Video). Thus, when Pax8 mice are treated with low doses of doxycycline, it is not surprising that they appear similar to Pax2 mice because Pax8+ cells also include Pax2+ cells, suggesting the clones observed in the two models are the same clones. This is now directly confirmed by the experiments with the FUCCI2aR transgenic line reported in new Fig. 7, in which we demonstrate that Pax2+ cells are the only TECs that divide after AKI.

Regarding the pictures of healthy Pax2 mouse kidneys in Fig. 2 within the OSOM and ISOM a lot of tubule segments show high amount of scattered Pax2+ cells lying nearby and which show very often the same colour (it seems very often the yellow and blue labelling occurs). If these segments are injured in a way that many of these cells start to proliferate one cannot speak of clonogenic tubule segments at all. It cannot be distinguished which of the yellow cells underwent cell division or if all of them just underwent one or two replication cycles. Are the different colours equally distributed among Pax2+ cells in the healthy mice? Were the large clones preferentially labelled in yellow? How many cells per Pax2+ clones can be counted and how much of the lost cells were replaced by Pax2+ cells based on evaluation of the paraffin tissue (comparing the evaluations made for the Pax8 mice in Fig. 1j,k,l)?

The treatment of Pax8 mice with lower doxycycline doses is still not convincing. Of course, there are not so much clonogenic tubule segments seen after a decrease of the labelling efficiency (by the way if these cells could represent also Pax2 expressing cells where are the clones with up to 98 cells?). As well, even with incomplete labelling (due to low labelling efficiency/doxycycline dose) still 54% regeneration of all AQP2- TECs by Pax8+ cells after IRI is seen, the other 46% could still arise from Pax8+ cells which are just not labelled.

5. The use of markers (i.e. Aquaporin) for quantification in scenarios of proximal tubular injury is common practice. However, it is important to remember that cellular injury can certainly lead to transient down-regulation of markers and therefore may have a direct effect on cell numbers. The possibility of transient down-regulation of markers should be considered, tested and discussed. Co-stainings of Pax8-rtTA mice with AQP1 in healthy, sham operated and IRI kidneys at time point T30 would be appreciated. AQP1 staining looks surprisingly heterogeneous after regeneration should be completed. Might this indicate that AQP1 may not be the best marker to count cell loss? Have the authors considered that its expression may have decreased transiently but the cells may not be lost (Fig. 2g,h)? Similar points apply for AQP2 (Fig 2 i-l).

We had considered the possibility that expression of AQP1 may have decreased transiently but the cells may not be lost. This is why first of all we counted the total number of Pax8/Confetti-labelled cells in absence of co-labelling with whatever tubular marker. These results were shown in old Fig. 1k as well as in new Fig. 1d, and demonstrate a significant decrease of total tubular cells at day 30 after IRI from 526.13 ± 16.3 to 365.8 ± 14 per field. As the major technical advance over previously used analytical tools lineage tracing excludes subsequent label alterations by tissue injury and persistently marks the labeled cell type as well as its progeny beyond gene regulation, as it e.g. can occur with immunolabeling. These results show a residual loss at day 30 of $30.5 \pm 2.8\%$ of TECs in the unilateral IRI model and of $27.6 \pm 12\%$ in the nephrotoxic AKI model, consistent with results shown in old Fig. 2 and counted on AQP1+ or on AQP2- cells (see new Fig. 1d, g, h and page 4, lines 77-81). In addition, we now further confirm these results with the FUCCI2aR transgenic cell line, as already discussed above (see new Fig. 7 and page 9, lines 202-226). All the four methods used indeed, give consistent results and therefore ultimately confirm the relevant tubular cell loss one month after AKI. In addition, as requested by the reviewer, co-stainings of Pax8-rtTA mice with AQP1 in healthy, sham-operated and IRI kidneys at time point T30 were

added and are presented in Supplementary Fig. 1g-i.

my concerns: Please state how many AQP1+ cells were lost in Pax2 mice? The images mainly show the ISOM and OSOM and only a small part of the cortex, never the whole cortex. Due to ischemic injury, also the proximal tubule should be damaged (the S1 segment maybe not that much than S3, but still considerable). Similarly regarding Pax2 mice: Did you see clonogenic proximal tubule segments since distinct, scattered Pax2+ cells were also reported in this tubular areas? High resolution images in low magnification to provide an overview of the damage have to be shown in Fig. 4. The same evaluation would be appreciated for the now added nephrotoxic injury model.

6. The use of the word replace is not clear or is at the very least debatable. In order to claim a cell deficit, proper quantification methods should be applied. At the moment, the data fails to provide sufficient evidence of persistent cell depletion. Furthermore, the authors present percentages in multiple graphs showing 20-35% of cell loss. In multiple analyses, the authors refer to increases in the order of 50% of the 20-35% meaning real increases of 10-18% - for such small differences to be identified properly and confidently, better quantification methods are needed.

As already discussed in the response to point 2, a novel quantification methods was added to the study (see new Fig. 7). The results obtained confirm those already reported with the Confetti transgenic mice (Fig.1, 3).

my concerns: As stated above, this method also has technical limitations.

7. The use of the word progenitor for PAX2+ cells is debatable. The data presented here does not confirm their progenitor state. Please rephrase.

As requested we reduced the use of the term progenitor in the manuscript. However, the new results added provide direct proof that Pax2+ cells are the only tubular cells within the nephron that divide and self-renew in response to injury (see new Fig. 7). Self-renewal is now demonstrated also by 3D analysis that revealed regeneration of entire tubule segments by single Pax2+ cells (see new Fig. 4). Finally, the results in Fig. 2i-l show that they represent an homeostatically stable population and not a functional state. Taken altogether, these results directly prove that Pax2+ cells localized within the nephron display progenitor properties as explained in the Discussion section (see page 12, lines 280-285 and page 13, lines 286-287).

my concerns: As stated above, the Pax2+ cells are not the only cells that undergo mitosis. Furthermore, the 3D analysis reveals that regeneration occurs, but while a lot of Pax2+ cells in healthy mice already are lying nearby within a tubule segment expressing the same colour of labelling, it is not substantiated by the experimental data that these tubule segments with large areas of the same colour arise from one of these cells and are not formed by replication of all of these clusters of cells. 3D reconstruction should also be done for injured Pax8 mice to confirm the absence of such long clonogenic tubule segments.

8. One of the main findings of this study is: "Tubular regeneration after AKI is selectively mediated by PAX2+ progenitor mitosis". The respective experiment confirms that PAX2+ cells are more sensitive to drugs that facilitate mitosis during AKI. If PAX2+ cells are indeed progenitors, why were these drugs employed to show a difference? Did the authors see a difference without the use of these drugs?

We now provide direct proof that tubular regeneration after AKI is selectively mediated by Pax2+ cells mitosis even in absence of drugs by combining flow cytometry and automated cell quantitation by MacsQuant using two new transgenic cell lines and the FUCCI2aR reporter (Mort RL, et al. Cell Cycle 2014). This represents an advancement of the FUCCI2 model, that avoids the problem of having a quote of black cells among M and G1 phase of cell cycle (Sakaue-Sawano et

al. Cell, 2008), as also underlined by reviewer 1, permitting tracing as well as quantitation of absolute numbers of total TECs and Pax2+ cells. These results directly proof that at all time points analysed, Pax2+ cells are the only tubular cells that divide after AKI.

my concern: Why were the experiments using mitosis-facilitating drugs not done with the Pax8 mice?

9. The first two sentences of the first paragraph: The statements presented in this first paragraph are exaggerated or over-interpretations of the actual data.

Following the request of the reviewer we have carefully adapted the first sentences of the first paragraph of discussion as suggested.

Okay.

10. Side-by-side publications by Berger et al. and Kusaba et al. report the potential of tubular cells to acquire the phenotype of scattered tubular cells upon tubular injury. This possibility should be discussed, or at least considered within the interpretation of the data.

We added in the discussion the possibility that some tubular cells may acquire the phenotype of scattered tubular cells upon injury (see page 12, lines 275-276). However, to test whether Pax2+ cells represent a distinct, predefined and stable TEC population or rather a transient functional state, we had compared two groups of mice, one that was induced with doxycycline for 10 days, then washed out for one week and tracked for the following 30 days, and another that was permanently exposed to doxycycline for an identical period of time (see old Supplementary Fig. 3 f-i, now Fig. 2i-l). Indeed, if the labelled Pax2+ cells after 10 days of doxycycline induction would represent a functional state that can be randomly acquired by TECs in healthy mice, their number should considerably increase when the length of exposure to doxycycline gets at least three times longer. However, no significant difference in Pax2+ TEC number was observed. Thus, these experiments exclude this possibility and clearly identify Pax2+ cells as an homeostatically stable distinct population of tubular cells.

my concerns: Regarding the sentence another subset of possible tubular progenitors characterized by Sox9 expression. Is this another subset of cells? Do Pax2+ cells co-express Sox9 in healthy and injured kidneys, respectively?

The doxycycline treatment is started with an age of 5 weeks. The mice are very young and still growing which can result in an increase of Pax2+ cells. A comparison with T0 is missing.

Minor

4. 5 week-old mice are still growing and analysis of cell numbers and volumes require age-matched controls. The experiments in this study were not designed this way. Therefore age-related bias cannot be ruled out.

Age-matched controls were shown in old Fig. 3e and indicate no significant differences with day 0, that's why we can rule out age-related bias.

These control kidneys were the contralateral kidneys of the IRI treated mice. These kidneys could also be harmed by the injury of the other kidney. Therefore, a slight increase of Pax2+ cells could be expected. Age-matched, sham-operated Pax2 mice would be appropriate controls.

5. The use of the contralateral kidney as an internal control or an age-matched control is if anything controversial. The ischaemia of one kidney can have significant effects on the other kidney.

The controlateral kidney was removed and only time 0 and age-matched controls are now included

(see new Fig 1j, Fig. 3g).

But now still the age-matched controls are missing.

6. Page 5, line 108: "The other 50% remains irreversibly lost." Have the authors excluded the alternative explanation that induction of Pax8 reporter expression may be lowered by giving lower amounts of doxycycline to the mice to create lower recombination rates? Also, as stated above, the use of AQP1 and AQP2 may not be appropriate to measure potential cell loss. In combination with decreased expression of the reporter genes it is not possible to give any evidence that the differentiated tubular cells are not able to repopulate the tubules.

As already pointed out, the results obtained with the Confetti transgenic cell lines were confirmed with an unbiased quantitation of the absolute remnant tubular cell numbers 30 days after AKI by automated counting using MacsQuant and two new transgenic mouse lines, based on the Fucci2aR reporter that do not have the possible bias reported by the authors (see new Fig. 7).

As mentioned above, it still is not unbiased.

8. It would be interesting to see whether KIM-1 levels in Pax2 mice are comparable to Pax8 mice as well as if both cell populations express KIM-1 similarly. Where there also GFR and BUN level measurements done for Pax2-rtTA mice?

KIM-1 protein was assessed in Pax8 and Pax2 mice. Results are now reported in Supplementary Fig. 7i, m. Interestingly, these results show that KIM-1 expression is largely observed in Pax8+ but not in Pax2+ cells (see page 9, lines 226-228). GFR and BUN measurement data for Pax2 mice were reported in Fig. 3a and in Supplementary Fig. 3b, respectively.

This answer is ok. The finding of rare KIM-1 co-expression in Pax2+ cells could be discussed in more detail.

Regarding the figures:

Fig 1: the Authors counted Aqp2 cells, but seem not to provide pictures of Aqp2 staining in Pax8 confetti mice.

Suppl. Fig 1: i is a different part of kidney than g and h

Fig 2g: for Pax8 confetti: %of Pax8+Aqp2- cells were counted by field and showed loss of Pax8+ cells after injury, what about Pax2+ cells? Here it was calculated as percentage of all Aqp2- cells, why not per field like for Pax8?

Fig 4b and c: Is both named Aqp1 containing, but on page 6 is Aqp1 and 2

Fig 6r': showed clearly anaphase of mitosis of an interstitial(?) cell, but it's also cherry positive (which is a marker for G1) and without any green color? This cannot be endocycling! How do the authors explain this?

Fig 6u: most venus positive cells seem to be interstitial, but not tubular cells. So at T2 the amount of proliferative Pax8 and Pax2 + cells are equal and on T30 the pictures showed in both no proliferative tubular cells.

Fig 6u': Dapi does not fit the other pictures, it is not the same area

Fig 7q': nuclei clearly positive for mCherry are negative for pH3. Some nuclei are weakly mCherry positive is this unspecific background or is it the transition from M (ph3+) to G1 (mCherry positive)

Fig 7r: venus+ seems not to be a nucleus?

Fig 8: pH3 looks unspecific. Every cell is positive and also CDK-, pH3 positive cells showed no classical PH3 chromatin pattern for G2 or M phase

Reviewer #3 (Remarks to the Author):

The authors have added substantial data in the revision. They have included prior lineage analysis work (Ref. 32, 33) but since these approaches excluded endocycling cells because the the CreER approaches used, better integration of why the current results differ so drastically from those prior studies is required. For example, if all S3 proliferation occurs from the 3.5% of Pax2+ 'progenitors,' then the authors must explain how clonal analysis of cells in S1 and S2 illustrate that over 50% of these cell clones undergo proliferative expansion during repair (not endocycle, since these were lineage analysis approaches). It is not clear how these studies are 'erroneous' or 'misinterpreted' as the authors write.

Point-by-point response to the reviewers' comments

Reviewer 2

Overall, this rebuttal has not addressed the limitations outlined in the previous round sufficiently. The experimental data does not support the author's conclusions and other interpretations of the data have not been tested or ruled sufficiently.

The paper is very hard to read. One has to spend significant time and effort to search for the relevant information. E.g. when authors claim tubular cell loss in paragraph 1 of the results section one needs to know the method of quantification to be able to judge the quality and evidence of this claim.

Summary of the major findings of the study from the abstract and their shortcomings:

1. regarding: Conditional Pax8/Confetti shows substantial tub cell loss despite function recovery.

my concern:

a. inappropriate method of analysis. In Fig 1D, quantification of loss of cells from AKI occurred by comparing the number of genetically labeled cells in the confetti mouse versus all cells in a histological section. This is an inappropriate method because it compares number of tubular cells vs. interstitial cells. After AKI, loss of tubular cells and interstitial fibrosis occurs. Changes in cell number within the interstitium are unpredictable: E.g. if many tubular cells were lost and very little additional interstitial cells accumulated, no loss of tubular cells would be detected. On the other hand, if large amounts of interstitial cells infiltrated a large loss of tubular cells would be detected.

These claims about our data analysis are incorrect. As clearly mentioned in the Results (see page 4, lines 76-80) in Fig. 1d we counted the total number of genetically labelled cells per field (and not percentage of confetti labelled cells vs. all cells in a histological section), exactly as performed in previously published literature, including for example in the study quoted by the reviewer as a reference (Berger et al).

In addition, we repeated all the same assessments also in the FUCCI2aR model where we automatically counted the absolute number of FUCCI2aR (only tubular) cells in the kidney (Fig. 7 k-n), with identical conclusions. These results are consistent and cannot in any manner be influenced by other cells.

It is suggested, that this prominent TEC loss is compensated mainly by hypertrophy of remaining Pax2 negative cells. These much lower numbers of cells would have to cover the same area of nephrons as previously the 100% of cells, so that they would have about twice the size (unless the interstitial matrix changes). Is this the case? Are they longer and flatter? Can this be truly reproduced in histology?

Yes, the cells increased in size and all requested data were already reported in the manuscript (see pages 10 and 11, lines 255-258 and Fig. 7 v, w). Indeed, endocycling cells are $38.8 \pm 5.5\%$ larger as compared to the other tubular cells. They do not have to cover the initial kidney size because the weight of the kidney anyway decreased upon injury as shown in Supplementary Fig. 1.1 f.

In general, the method of counting are still unclear and in part described insufficiently (e.g. what is A in the formula on P23L546 and B in the formula on P23L557)? Supplementary data examples showing the counting procedure would be helpful.

The definitions of A and B were already reported in the manuscript (see page 23, lines

585-589). Following the suggestion of the reviewer we have now added Supplementary data examples illustrating the counting procedure in detail in a new file of Supplementary Methods.

In general: How can number of cells be counted in confetti mice which show cytoplasmic reporters?

As reported in the methods section, all the counts were done on sections where the nuclei were stained with DAPI (page 23, lines 579-580). In many figures we decided to show the images without the nuclei because for a non-expert reader this means adding a further colour that may be confounding. Anyway, figures are available also with nuclei.

These methodological problems limit analysis of the drug intervention experiments (suppl. Fig. 5) as well.

See the responses to each single point above.

b. down-regulation of markers (see previous review below).

We respectfully disagree with the reviewer. We provide quantitation in Confetti mice as well as in the Fucci2aR mice in presence (Fig. 1d, Suppl. Fig. 6-2, respectively) and absence (Fig. 1d and Fig 7k-n, respectively) of immunolabelling for the tubular marker AQP2. All the four methods gave identical results. Therefore, we can absolutely exclude downregulation of markers to affect our results.

c. altered kidney tissue integrity limits FACS analysis after acute kidney injury (see below)

We respectfully disagree with the reviewer. FACS analysis is considered the gold standard to reproducibly quantify cell numbers in cell suspensions obtained from digesting injured solid organ tissues (Wang B. et al, Nature 2015; Ding B-S et al, Nature 2010) including the injured kidney (Li Y. et al, Nat Med 2010, that uses it in 3 different AKI mouse models). In addition, false signals can be excluded because we have an internal control. See below for further technical explanations (query 7).

d. In Fig 1, the IF in T30 vs T0 shows very prominent loss of red color and a large number of clusters of blue and yellow cells. This does not seem to be reflected anyhow in the quantification of clones.

This point is irrelevant because all counts are based on the variation of the number of adjacent cells of the same colour over time of all colours of the Confetti system in this study just like in those published by many other authors (Snippert HJ et al. Cell 2010; Schepers AG et al, Science 2012; Ritsma L. et al, Nature 2014; Jamieson PR et al, Development 2017). We wonder whether the reviewer interpreted the random colours used in the clone frequency analysis graph shown in Fig. 1j as representations of the true colours of the clones. Obviously this is not the case and to avoid confusion we have eliminated red, green, blue and yellow from the graph.

Furthermore, the authors conclude, that only Pax2+ cells could proliferate, how do the authors explain the increase in cell clones in Pax8 confetti mice?

The Pax8 promoter is widely used in the field to target all tubular cells. Therefore, Pax8+ tubular cells labelled with Confetti (see page 4, lines 64-67), include the Pax2+ population as an anyway small subset, and the clones seen in Pax8/Confetti mice are most likely generated by the Pax2+ subpopulation.

2. regarding: Clonal analysis disproved widespread tub cell proliferation, erroneously indicated by proliferation markers unable to ultimately identify cell divisions.

my concerns:

a. As shown in Figure 1j, only about 1% of all clones contained more than 6 cells after regeneration. This result would allow to conclude that regeneration does not occur from a small predefined progenitor population – nevertheless the authors conclude the opposite. I cannot judge the statistical method used, but I believe that common sense is sufficient here.

a. We respectfully disagree with the reviewer for the following reasons:

-clonal analysis shown in Fig. 1j quantifies the percentages of clones containing different number of cells on survived clones at day 30. Given the massive cell loss, clone frequency at day 30 does not mirror the percentage of cells that truly divided as explained in the added scheme in Supplementary Figure 1.2.

-we demonstrate that 2D analysis underestimates the true clone length and represents a sensitivity-related technical artifact. Indeed, 3D analysis reveals clones long up to 98 tubular cells originating from a single (yellow) tubular cell that survived the injury phase of AKI (Video).

b. Why did the authors use a higher magnification in Fig. 6f? Do they want to create the impression that proliferation markers are expressed in only a few tubular cells after injury? This might be a 'mistake', but it makes me worry and concerned.

In Fig. 6f the magnification is the same, as indicated by the scale bar. Hence, the images document that PCNA labels numerous cells rather than “few”. The point is that such labels when referred to as “proliferation markers” lead to wrong conclusions and disease concepts. These markers indicate nothing else but cell cycle activation, which becomes obvious in comparison with our clonal lineage tracing analysis that overts this discrepancy and disproves the misconception of widespread stochastic cell proliferation in response to AKI.

c. Significant polyploidy has not been observed previously in detailed and meticulous studies (see leHir et al).

Polyploid cells result either from endocycle (single nucleus) or from endomitosis or cell fusion (multiple nuclei). The standard techniques used by Le Hir et al. may detect endomitosis but are unable to detect endocycle. This becomes only possible with the advanced technical approach used by us (see Ganem et al. Cell, 2014 for technical explanations), and hence represents an important new discovery, which we are glad was readily recognized by the other reviewers. Indeed, this difference between the historical and the new technical approach was explained in the Discussion (see page 13, lines 319-320). To minimize possible misunderstandings we added it once more in the Results (see page 9 lines 207-210).

Did the authors consider cell-cycle arrest? This is one of the mechanisms proposed how AKI progresses to CKD, and the provided data do not exclude this possibility.

Endocycling cells may even be growth-arrested, we do not question this point. The point is that these cells do not undergo mitosis as widely erroneously concluded from immunolabeling with so-called “proliferation markers”.

d. Did the authors considered counting mitotic figures as the „best“ marker for proliferation? This is also possible in DAPI.

Post-AKI mitotic figures have been counted extensively 30 years ago and were found to be very rare, which supports our conclusions and is rather in sharp contrast to the current dogma based rather on the widespread positivity of “proliferation markers” (Cuppge FE et al., Am J Pathol 1967; Phillips TL et al., Cancer Research 1967). In our view it is time to admit that in AKI the conclusions taken from immunolabeling are misleading, while the historical counts of mitotic figures (spot mitosis only occurring in the moment of fixation) and our results (spotting all mitotic events that have occurred in a 4 weeks period) are absolutely consistent.

3. regarding: A small subset of Pax2+ tubular progenitors got enriched via higher stress resistance and clonal expansion, regenerating necrotic segments.

my concerns:

a. I am unconvinced that Pax2 is a suitable marker to identify a predefined population of progenitor cells within the kidney tubule – which is a major argument of this study. I would advise the authors to visit the human protein atlas online. There, authors will see that Pax2 is expressed in virtually all tubular cells in most human biopsies. I suspect that Pax2 is a simple marker for the injury phenotype of tubular cells (scattered tubular cells), which is upregulated in any tubular cells in states of proteinuria or any other type of injury (see the work of Humphreys or Moeller). Authors have even made this observation (as stated on page 6) – but they still adhere to their interpretation.

We respectfully disagree. In the healthy kidney only rare proximal tubular cells stain positive for Pax2, a population with distinct morphology and functional properties, as mentioned also by reviewer 3. The same we observed in our mouse model. Biopsies from injured kidneys can diffusely express Pax2 in tubular cells because Pax2 is readily upregulated upon injury. For this reason Pax2 immunostaining is absolutely useless in fate mapping of the Pax2+ population present in healthy kidneys. The lineage tracing is a technique that was invented to finally overcome this limitation of immunostaining. For more details we refer to our recent detailed review on this topic (Romagnani, et al. Nat Rev Nephrol 2015). For this reason, only Pax2 lineage tracing activated before onset of injury can allow the fate tracing of the Pax2+ population present in healthy kidneys. In our experiments, we perform a lineage tracing and we can unequivocally distinguish amplification of the Pax2+ population from Pax2 upregulation. We had carefully discussed this point inside our manuscript and explained why this is fully consistent with our data (see page 6, lines 134-138).

4. Authors need to characterize the expression of Pax2 in mouse kidney in healthy adult mice (authors used very young mice which are still growing) and after injury. In young mice, Pax2 might be expressed in tubular cells which are under mechanical stress (e.g. the inner turn of a tubule, as reported previously). It is conceivable that these cells will be the first to become STC and proliferate after injury. No need for progenitor cells here. I would predict that Pax2 positive tubular are virtually absent in normal healthy adult mouse kidney (i.e. 12 weeks of age).

The reviewer raises a valid concern, but we can absolutely exclude it, because the Pax2+ cell population is stable along life time in mice, so also at 12 weeks of age. These data were available and have now been added to the revised version of the manuscript (see page 6, lines 117-120)

5. regarding: Drugs that enhance tub reg confirmed that this occurred only through progenitor mitosis.

my concerns:

a. Results are limited by an inappropriate method to quantify cell numbers.

See responses to identical comment raised in point 1.

regarding the point by point rebuttal:
my concerns:

6. How can you be sure that only Pax2+ cells complete mitosis? If the percentage of mCherry+mVenus+ Pax2+ cells is much lower than Pax8+ cells, there could also be a biased interpretation due to the very small amounts of Pax2+ cells overall. A time-dependent analysis of the cell cycle immediately after IRI with more than two time points would be needed. If there were an increase in the number of mVenus+ Pax2+ cells after IRI and a decrease up to day 30 which cannot be seen in Pax8+ cells, it would strengthen your argument.

As we explained in the manuscript, (see page 4, lines 64-67), the marker Pax8 is expressed by all tubular cells, hence, Pax8+cells include the Pax2+ cell subset. So we can exclude the reviewer's concern. Even if the percentage of Pax8+FUCCI2aR+ cells is considerably higher than that of Pax2+FUCCI2aR+ cells over the total cells of the kidneys, the absolute number of mVenus+ cells is always identical (Fig. 7n). Since Pax2+ cells are a subset of Pax8+ cells, this unequivocally proves that they are the same cells as explained in page 10, lines 240-246).

At day 30 there are still nearly half of the Pax2+ cells mVenus+, does that mean they are stuck within the S phase even after a time in which regeneration should be completed?

Our results show that Pax2+ cells don't get arrested in the S phase because they increase in absolute numbers (Fig. 7m), while other tubular cells decrease (Fig. 7m), means they keep proliferating.

7. my concerns: The new method used differs a lot from the one I suggested. On day 2 after IRI the kidney is very damaged. The tissue is very susceptible against shear forces like the mechanical disintegration through meshes. Intact but marginal cells could be lost because of the harsh isolation method. A lot of cell debris could stick at intact cells and result in false signals. An additional confirmation with the suggested method would be appreciated.

See response to 1c and the following:

-The false signals suggested by the reviewer in the quantitation provided in Fig. 7 by labelling of mCherry and mVenus fluorochromes with FACS analysis can definitely be excluded by our internal control. Indeed, in Pax2FUCCI2aR mice, mCherry perfectly matches with cells in G1 phase and mVenus with cells in s/G2-M, as depicted by DNA content, validating the specificity of the signal. The fact that in Pax8FUCCI2aR that undergoes exactly the same digestion procedure of the tissue, endocycle can instead be detected, further underlines the specificity of the signal.

In addition, this reviewer had asked us a better quantitation of cells suggesting a method that cannot be technically performed in Confetti mice. More in detail:

-the method of Walton et al. requires paraffin-embedded sections and thus cannot be applied to Confetti mice because the fluorochromes will be lost upon paraffin inclusion.

-in addition, Walton et al. clearly write in the discussion section that "In settings of pathology, expression of markers may be diminished to an undetectable level, reducing the validity of this method of tubule identification" (see attached image of their

statements), therefore this method doesn't solve the problem raised by the reviewer (marker downregulation after injury), so we cannot understand why he/she proposed it.

8. my concerns: This method will miss cells that just replicate once or twice creating bias towards longer clonogenic tubule segments.

The reviewer got wrong on how we calculated the clones. In our clonal analysis, we also considered all the cells that replicate just once or twice. Indeed, we considered all clones observed at T30, including those made of only two cells. At T0 doublets are extremely rare, so all the clones of even two cells can be considered. In addition, even if they are a rare finding, doublets present at T0 are subtracted from the clones observed at day 30. To avoid that this misunderstanding may happen to another reader, we have now added a detailed example of all the calculations as a Supplementary Method file.

9. Regarding the pictures of healthy Pax2 mouse kidneys in Fig. 2 within the OSOM and ISOM a lot of tubule segments show high amount of scattered Pax2+ cells lying nearby and which show very often the same colour (it seems very often the yellow and blue labelling occurs). If these segments are injured in a way that many of these cells start to proliferate one cannot speak of clonogenic tubule segments at all. It cannot be distinguished which of the yellow cells underwent cell division or if all of them just underwent one or two replication cycles.

We show that Pax2+ cells in the cortex and OSOM, the areas analysed in our study, are always single in these mice. Only collecting ducts have adjacent cells of the same colours (Fig. 2c) and they were excluded from our calculations by immunolabelling for AQP2, as clearly stated (see page 6, lines 127, 132, 134). Adjacent cells of the same colour in healthy mice represent only $2.2 \pm 0.2\%$ of all Pax2+AQP2- cells at Time 0. To make it even clearer, and avoid that this misunderstanding may happen to another reader, we decided to add a new graph directly showing the clone frequency at time 0, in age-matched controls, in sham operated mice as well as in mice with AKI at day 30 after injury in Pax2+ mice (see new Fig. 3f and related results description page 6, lines 127-132).

10. Are the different colours equally distributed among Pax2+ cells in the healthy mice?

This information was already included in the manuscript. It is well known and reported since its first description (see Snippert et al. Cell, 2010), and we have clearly stated in the results section of the first version of the manuscript (now moved to the methods section, see page 15, lines 361-363), that GFP is much less likely than the other three colours that instead have about the same frequency. However, this is irrelevant for the analysis as also explained in our response to point 1d.

11. Were the large clones preferentially labelled in yellow?

No, large clones were observed in every colour as shown in Fig. 3c and 4b-c. For 3D reconstruction, we chose only the yellow ones because the YFP is distributed in all the cell, it is particularly bright and gives a highly defined signal, making the 3D reconstruction easier to build and appreciate. This specification was added to the Methods section (see page 22, lines 547-551).

12. How many cells per Pax2+ clones can be counted and how much of the lost cells were replaced by Pax2+ cells based on evaluation of the paraffin tissue (comparing the evaluations made for the Pax8 mice in Fig. 1j,k,l)?

We did not evaluate paraffin section because the Confetti fluorochromes cannot stand paraffin inclusion. In addition, the number of cells in Pax2+ clones was already reported in the manuscript, based both on the 2D (“clones were made of up to 10 cells”, see page 7, line 152) and 3D analysis (“clones were made up to 98 cells”, see page 7, line 153). Anyway, we understand that the reviewer wants to see the data not only in percentages but also in numbers. As an answer, we added three new graphs showing:

- the clone frequency of Pax2+ cells at all time points analysed (Fig 3f).
- the number of Pax2+ cells per field before and after injury (Fig 3e);
- the percentage of new Pax2+AQP2- cells generated at day 30 after injury (Fig. 3g).

All these analysis further confirm the different behaviour of Pax2+ cells in comparison to other TECs that was already shown in percentages in old Fig. 3h,j.

13. The treatment of Pax8 mice with lower doxycycline doses is still not convincing. Of course, there are not so much clonogenic tubule segments seen after a decrease of the labelling efficiency (by the way if these cells could represent also Pax2 expressing cells where are the clones with up to 98 cells?).

Since Pax2+ cells represent about 8-10% of Pax8+ cells, when lower doses of doxycycline are used in Pax8+ mice, also the probability to label Pax2+ cells will decrease in the same way, i.e. long clones can be found also in Pax8 mice but are much more rare upon dilution being also more difficult to find than in Pax2 mice. This provides another proof that long clones come only from Pax2+ cells.

14. As well, even with incomplete labelling (due to low labelling efficiency/doxycycline dose) still 54% regeneration of all AQP2- TECs by Pax8+ cells after IRI is seen, the other 46% could still arise from Pax8+ cells which are just not labelled.

We can exclude this option because all data are expressed as percentages of cells stochastically labelled cells, independently from the labelling efficiency. Thus, the percentage of regeneration will be 54% in labelled as well as in unlabelled cells.

15. my concerns: Please state how many AQP1+ cells were lost in Pax2 mice?

Percentages of lost AQP1+ cells in Pax2 mice is comparable to the percentage of lost AQP1+ cells already reported in Fig. 1d for Pax8 mice (30.2±9.3% in Pax2 mice, 32.5±7.1% in Pax8 mice, respectively).

16. The images mainly show the ISOM and OSOM and only a small part of the cortex, never the whole cortex. Due to ischemic injury, also the proximal tubule should be damaged (the S1 segment maybe not that much than S3, but still considerable).

We can exclude this possibility. In Fig. 1b,c we showed the whole kidney cortex up to the border of the section. We now enlarged also Fig. 3b, showing a wide area of the kidney to include the whole cortex also there.

17. Similarly regarding Pax2 mice: Did you see clonogenic proximal tubule segments since distinct, scattered Pax2+ cells were also reported in this tubular areas?

The data requested by the reviewer were already included in the paper. Data on frequency of Pax2+ clones in cortex (S1-S2 segments) were already reported in Fig. 4f and showed that only a minority of Pax2+ clones are observed in the S2 segments.

18. High resolution images in low magnification to provide an overview of the damage have to be shown in Fig. 4.

Figure 3b already included a high resolution image in low magnification with an overview of the damage. We now enlarged it to include also all the cortex.

19. The same evaluation would be appreciated for the now added nephrotoxic injury model.

The data requested by the reviewer were already included in the paper. The nephrotoxic model together with a careful quantitation of all the data on AQP2- cells were shown in old Fig. 3l, m, n and described in page 6, lines 142,143

20. my concerns: As stated above, this method also has technical limitations.

This is a reiteration of the criticism in points 1a and 1c (see response to point 1a, c and 7).

21. my concerns: As stated above, the Pax2+ cells are not the only cells that undergo mitosis. Furthermore, the 3D analysis reveals that regeneration occurs, but while a lot of Pax2+ cells in healthy mice already are lying nearby within a tubule segment expressing the same colour of labelling, it is not substantiated by the experimental data that these tubule segments with large areas of the same colour arise from one of these cells and are not formed by replication of all of these clusters of cells. 3D reconstruction should also be done for injured Pax8 mice to confirm the absence of such long clonogenic tubule segments.

The point raised is identical to point 4 so see the response to point 4.

22. my concern: Why were the experiments using mitosis-facilitating drugs not done with the Pax8 mice?

Drug experiments were not done in Pax8 mice because, by using Pax2Confetti mice we were able to compare the behaviour of Pax2+ cells with the other tubular cells within the same model. This allowed us to establish that Pax2+ cells proliferate in response to the drugs, while other TEC do not.

23. my concerns: Regarding the sentence another subset of possible tubular progenitors characterized by Sox9 expression. Is this another subset of cells? Do Pax2+ cells co-express Sox9 in healthy and injured kidneys, respectively?

For technical reasons we cannot assess if this is the same population of cells because the two anti-Sox9 antibodies reported in the literature do not work with the fixation procedures that are needed to maintain fluorochromes in our Pax2 mice.

24. The doxycycline treatment is started with an age of 5 weeks. The mice are very young and still growing which can result in an increase of Pax2+ cells. A comparison with T0 is missing.

Comparisons with T0 were already included in the paper (old Fig. 3g). We saw no statistically significant difference in the number of Pax2+ cells at T0 and T30, this is why the comparison with healthy mice at time 0 or age matched control T30 will obviously give the same results. In addition, as already reported in the response to point 4, the Pax2+ cell population is stable along life time in mice, so also at 12 weeks of age. These data were available and have now been added to the revised version of the manuscript (see page 6, lines 117-120 and new Supplementary Fig. 2f)

Minor

1. These control kidneys were the contralateral kidneys of the IRI treated mice. These kidneys could also be harmed by the injury of the other kidney. Therefore, a slight increase of Pax2+ cells could be expected. Age-matched, sham-operated Pax2 mice would be appropriate controls.

In addition to controlateral kidneys (that we had removed in our previous revision following this reviewer's request), we were also showing age-matched controls in all the previous versions of the paper (see old Fig 3g). As stated above, we see no statistically significant difference in the number of Pax2+ cells in healthy mice at T0 and healthy mice traced up to day 30 (age-matched controls, T30). We have now added even the sham-operated controls traced up to day 30 (new Fig. 3 e, f) that were not initially included because they show no statistically significant difference with T0 and T30.

2. But now still the age-matched controls are missing.

The age-matched controls (T30) were already there in the previous version and are still there as stated above (New Fig.1j and Fig. 3e, f, h).

3. As mentioned above, it still is not unbiased.

This is a reiteration of the criticisms already commented in point 1a, 1c.

4. This answer is ok. The finding of rare KIM-1 co-expression in Pax2+ cells could be discussed in more detail.

We discussed the finding in more detail (see page 12, lines 298,299).

5. Fig 1: the Authors counted Aqp2 cells, but seem not to provide pictures of Aqp2 staining in Pax8 confetti mice.

Representative Pax8+AQP2-stained sections were now added (see new Fig1i)

6. Suppl. Fig 1: i is a different part of kidney than g and h.

The Figures showed the same areas, but sections are sagittal in g and h and cross in i. For a better comparison we now show all cross sections.

7. Fig 2g: for Pax8 confetti: %of Pax8+Aqp2- cells were counted by field and showed loss of Pax8+ cells after injury, what about Pax2+ cells? Here it was calculated as percentage of all Aqp2- cells, why not per field like for Pax8?

Pax8+ cells were counted by field only in Figure 1d to prove absolute total cell loss.

However, in all the other Figures, Pax8+ cells are expressed as a percentage same as Pax2+ cells to be directly compared (old Fig. 3g).

As required by this reviewer, we have now added also three new graphs where we show all the data in numbers/field also for the Pax2+ cells (new Fig. 3e-g). As the reviewer can see data expressed in percentage or numbers are similar.

We have now added also detailed explanation and an example of all the calculations in Supplementary Methods (see pages 17-19 of Supplementary Informations).

8. Fig 4b and c: Is both named Aqp1 containing, but on page 6 is Aqp1 and 2

All the markers, AQP1, THP and AQP2 were immunolabelled in order to establish the frequency of Pax2+ clones in different tubule segments as reported in Methods and in Fig. 4f. However, in Fig. 4b and c only two representative AQP1+ clones are shown, as specified in Figure legend (page 40, lines 900-902)

9. Fig 6r: showed clearly anaphase of mitosis of an interstitial(?) cell, but it's also cherry positive (which is a marker for G1) and without any green color? This cannot be endocycling! How do the authors explain this?

Fig 6u: most venus positive cells seem to be interstitial, but not tubular cells. So at T2 the amount of proliferative Pax8 and Pax2 + cells are equal and on T30 the pictures showed in both no proliferative tubular cells.

Not a single interstitial cell can be shown in these Figures, because both the Pax8 and the Pax2 promoters are selectively active in tubular epithelial cells, as already reported by other studies (Kaminsky MM, Nature Cell Biology 2016, Naiman N. Developmental Cell 2017). We see 100% labelling of FUCI in tubular epithelial cells in our transgenic mice (as now shown in sections costained with phalloidin, new Fig 6i-k and 6m-o) and the system has no leakage (Supplementary Fig 6.1 l-q). Since this is a lineage tracing performed with a conditional system, the possibility that interstitial cell labelling occurs at some time points after injury can absolutely be excluded.

We think this misunderstanding of the reviewer may have been induced by the fact that at T30 the structure of the tissue is quite damaged because of the residual cell loss and the border of many tubules are not clearly detectable like at T0. To make it clearer, we have replaced Fig. 6i-k and 6m-o with new ones including colabelling with the cytoskeleton marker Phalloidin not only at day 0, but also at day 2 and day 30 after IRI, which shows that at T0 FUCI labelling is selectively found in tubular cells while at day 30 after IRI this is hard to determine because many tubular structures are hardly recognizable. These explanations were added in the Results page 8, lines 190-192

In addition, although this is not an anaphase of a mitosis as speculated by the reviewer, we replaced Fig. 6r with one that does not induce the possibility of such interpretations.

10. Fig 6u': Dapi does not fit the other pictures, it is not the same area

We apologise for the mistake. The correct DAPI panel was now added.

11. Fig 7q': nuclei clearly positive for mCherry are negative for pH3. Some nuclei are weakly mCherry positive is this unspecific background or is it the transition from M (ph3+) to G1 (mCherry positive)

The system has no unspecific background, as shown in Supplementary Fig. 7 g,h. Cells going out of mitosis are still green. Cells intensely red are in late G1. Cells with less intense red are in early G1, so both light red or intense red cells can be costained with p-H3. To make it clearer, we are now showing another area of the same image in the splitted pictures where the red colabelling with blue is more intense.

12. Fig 7r: venus+ seems not to be a nucleus?

Venus+ is a nucleus. However, we have replaced this image with another one where the nucleus is more clearly recognizable.

13. Fig 8: pH3 looks unspecific. Every cell is positive and also CDK-, ph3 positive cells showed no classical PH3 chromatin pattern for G2 or M phase

We show a healthy kidney as a comparison where only a single pH3+ cell is shown (Fig. 8a). In addition, even in biopsies some cells are only red or only green, as also clearly shown by a lower power magnification of old Fig. 8c. This excludes the possibility of unspecific labelling of pH3.

In addition, the statement of the reviewer on the pH3 pattern is incorrect. Intense red labelling is characteristic of cells in mitosis; by contrast, in G2 or in cells with an open chromatin, the pattern is less intense and has a dots-like appearance (Yang L. Nat Med. 2010, Crosio C. Molecular and Cellular Biology 2002).

Reviewer 3

The authors have added substantial data in the revision. They have included prior lineage analysis work (Ref. 32, 33) but since these approaches excluded endocycling cells because of the CreER approaches used, better integration of why the current results differ so drastically from those prior studies is required. For example, if all S3 proliferation occurs from the 3.5% of Pax2+ 'progenitors,' then the authors must explain how clonal analysis of cells in S1 and S2 illustrate that over 50% of these cell clones undergo proliferative expansion during repair (not endocycle, since these were lineage analysis approaches). It is not clear how these studies are 'erroneous' or 'misinterpreted' as the authors write.

This reviewer has no further technical criticism on our revised manuscript.

The only comment of this reviewer addresses the discrepancy of our results with the study of Kusaba et al. and Berger et al. But this discrepancy can be easily explained.

In detail:

-Berger et al did not perform clonal analysis and cannot conclude on how many cells truly divided.

-Kusaba et al instead performed a clonal analysis, as correctly stated by the reviewer. To count clones, Kusaba et al dilute labelling by administering low doses of doxycycline as we did in the Pax8Confetti mice, albeit using a single fluorochrome. They calculated a clone frequency at end of their lineage tracing after 2 weeks. They apply two models:

-a 35 min IRI model, where they observe a clone frequency of over 50%, as reported by the reviewer. In such a model, injury is extremely severe so these numbers cannot be taken as a comparison to the ones we report.

-a 26 min unilateral IRI model, where after a lineage tracing period of 2 weeks, they observe a clone frequency of 27.4% in the inner cortex, and don't observe clones at all in the outer cortex, where most of the S1 and S2 segments are localized. This is very similar to our results. We apply a 30 min IRI model and observe at the end an overall clone frequency of 21.1% (see Fig. 1j).

Kusaba et al. would conclude that 21% of TECs generated clones, but this is not what truly happened. Indeed, the 4 clones observed at day 30 after IRI, represent the progeny of

only 8.3% (see Fig. 1I) of TECs labeled at T0 (not the 3.5% reported by the reviewer, that we observed in the nephrotoxic AKI model, where both kidneys are injured), as shown below.

Indeed, without considering the possibility of cell loss, Kusaba et al. assumed that the clone frequency calculated on survived clones at the end of the tracing mirrors the percentage of dividing cells upon injury. However, since irreversible cell loss enriches dividing clones this leads them to overestimate dividing cell numbers.

In summary, clone frequency at the end of the tracing period cannot be used as a quantification of clonogenic TECs.

Maybe this schematic cartoon illustrates the problem better so we added it as a new Supplementary Fig.1.2.

Finally, both the authors, with the lineage tracing technique they use cannot detect or exclude endocycle. Indeed, detection of mononucleated endocycling cells requires the combination of techniques that we used in our study (see Ganem et al. Cell 2014). In addition, endocycling cells are labelled by cell cycle markers such as PCNA or KI67, representing a further cause of overestimation, that the authors could not consider without knowing the existence of endocycle.

We now added in the discussion a careful explanation that the results of Kusaba and Berger are not wrong. Simply, they were the only possible conclusions without considering the occurrence of massive cell loss and knowing the existence of endocycle (page 13, lines 306-316).

Addendum to response to Reviewer 2, point 7

Quotations from: Walton SL, Moritz KM, Bertram JF, Singh RR. Lengths of nephron tubule segments and collecting ducts in the CD-1 mouse kidney: an ontogeny study. *Am J Physiol Renal Physiol.* 311: F976-F983, 2016.

“The aim of this project was to develop methodology to estimate lengths of specific segments of nephron tubules and collecting ducts in the CD-1 mouse kidney using a combination of immunohistochemistry and design-based stereology”.

“In settings of pathology, expression of markers may be diminished to an undetectable level, reducing the validity of this method of tubule identification.”

“A caveat to our approach was that it necessitated the use of paraffin-embedded tissues and thus our estimates of length are most likely underestimates given that paraffin processing results in significant tissue shrinkage.”

Reviewers' comments:

Reviewer #4 (Remarks to the Author):

This paper argues that kidney tubular epithelial cell regeneration after injury is caused by the proliferation of a small subset of tubular cells, which are Pax2-positive and divide to form clones after injury, rather than by proliferation of most or all TECs. I believe that there are problems with the quantitative analysis of the number of clonogenic cells. The evidence that many TECs undergo endocycle rather than complete mitosis after injury is stronger. The paper is very difficult to follow in many places, and needs extensive editing.

Major Points

Figure 1 and associated text: I believe that the clone size analysis, which is used to calculate the percentage of "clonogenic cells" (Fig 1I), is flawed. They are counting colored cells in single sections, while many clones are expected to be oriented such that the section contains only a subset of cells in a clone; in some cases, only a single cell of a clone may be present in the section, so it is not counted as a multi-cell clone. This underestimates the numbers of multi-cell clones, which is used to calculate the number of clonogenic Pax8+ cells. I am aware that a similar method of clone-counting was used by Kusaba et al. (who cite an earlier paper by Driessens et al. to justify its use), but they did not use it for similar calculations. In the present paper, underestimating the number of Pax8-lineage clones will also lead to an underestimate of the number of clonogenic cells.

The same issue applies to Figure 3, where Pax2-lineage clones are analyzed.

Figure S2, c-e. If the mice "display consistent Pax2 expression" (line 106-107), why do the Pax2 antibody-stained cells ("white") not correspond to the Pax2-lineage labeled cells (yellow, red, blue, green)? The results shown suggest that Pax2 expression is intermittent or changing over time. This seems to contradict the results of Figure 2 i-l, which show that the number of Pax2-lineage cells is stable over time.

Minor Points

Line 33, 38, 323, 342. I object to the words "disproved" and "disprove". It is fairer to say "clonal analysis argued against..." and "These results challenge current paradigms".

Line 75. "apparently reconstituted TECs and single coloured clones in outer stripe of the outer medulla (OSOM) (Fig. 1b,c)."

I don't know what they mean by "reconstituted". They define "clones" as anything from one to 11+ cells, including single cells. Single colored "clones" don't mean anything other than that Pax8 was expressed in the cell at some earlier point – they don't show that anything was "reconstituted". I think what they are trying to point out is the presence of multi-cell, single color clones, so they should be more specific.

Figure 2 legend: "confetti reporter shows Pax2 expression" –

No, it shows cells (or the daughters of cells) that expressed Pax2-rtTA at any time in the past when doxycycline was present.

Line 81 "Identical data" – change to "similar data"

Line 77, Line 84, line 101 – "irreversible", "irreversibly lost" – they don't show that the loss of TECs is irreversible, only that it is not reversed within 30 days. They could say "a substantial and sustained TEC loss".

Line 90 "the percentage of single cell clones at T0 decreased from $92.4 \pm 0.9\%$ to $78.9 \pm 0.9\%$ at day 30 after IRI".

Change to "the percentage of single cell clones decreased from $92.4 \pm 0.9\%$ at T0 to $78.9 \pm 0.9\%$ at day 30 after IRI"

Figure 1d. They are not measuring "Pax8+ cells" as the axis labeling and legend stat, but rather Pax8-lineage-positive cells, i.e., cells that expressed Pax8.rtTA at the earlier time when Dox was administered; they may no longer express Pax8 at the time of imaging. To call them Pax8+ is confusing and should be corrected. I agree with the rebuttal, that they are not counting "tubular cells. vs. interstitial cells", as Reviewer 1 stated.

Figure 1l – they don't define the meaning of the gray vs. white areas of the bar.

Supplementary Figure 1.1 – there is no legend for panels l and m.

Line 111-114: "Immunolabelling with megalin, that stains proximal TECs of the S1 and S2 but not S3 segment, showed that Pax2+ cells represented 1-3% of TECs in S1 and S2 segments (Fig. 2h). Labelling with AQP1 showed that they represented 6-10% in S3 segment (Fig. 2a,e) and labelling with THP showed that they represented 8-12% of distal TECs (Fig. 2b,f)."

Are these numbers estimates, not actual counts? Why are they presented as ranges (1-3%, 6-10%, etc.) rather than actual averages +/- SD?

Figure 3i,j,k,m,n,o. It is not obvious how these values were calculated – please explain in the figure legends.

Figure 4f. The values of the 4 bars do not add up to 100%.

Line 154: "3D analysis revealed single coloured-Pax2+ clones up to 98 cells long regenerating entire tubule segments".

How do they know that this is an entire tubule segment?

Line 161: "Treatment starting not before 24 hours after IRI"

This tells us when treatment did not start, but when did it start? At 24 hours after IRI? Please clarify.

Figure 5f. The ordinate is labeled (No of Px2+ cells included in clones"

Do they mean "No of PAX2+ cells per field included in clones"?

Again, these are Pax2-lineage+ cells, not Pax2+ cells.

Figure 5. The legend for g-k says "arrows indicate single-color clones". First, their definition of a clone is a group of 2 or more single-color cells in contact with each other, so all clones are "single-color". But then, the two arrows in h point to tubules with cells of more than one color – so this is confusing.

Line 188 "the number per field was comparable to that of the Pax8/FUCCI mice (Fig. 6n vs j)."

What are the numbers per field?

Line 194-195. "a lower percentage of TECs 195 at day 2 was p-H3+ in comparison to Ki-67 ($11.9 \pm 1.1\%$ vs. $47.1 \pm 9.2\%$, $p < 0.05$)."

Are these numbers plotted in any Figure?

Fig 6s. The legend on the ordinate "% of Pax8/FUCCI2 cells over p-H3+ FUCCI2 cells" does not match the legend, which says "Percentage of p-H3+mCherry+ cells and p-H3+mVenus+ cells in Pax8/FUCCI2 mice....."

Line 214: "Rather, after IRI $35 \pm 4.8\%$ of Pax2+ cells were in the S phase"
Change to "...2 days after IRI....."

Line 227. "Both transgenic lines showed a percentage of induction higher than 90%."
Please define "percentage of induction", and where is this data plotted?

Line 298: "This was further underlined by their irrelevant expression of the TEC injury marker KIM-1"

What do they mean by "irrelevant" ?

Line 304-305 "Pax2+ cells can give rise to all the cells along the tubule, they are not replaced by unlabeled Pax2- cells. That is, Pax2+ cells self-renew, a distinctive property of progenitor cells"
I don't understand this argument – if some unlabeled Pax2- cells had replaced some "Pax2+" , they would not be seen as they are not labeled. How can they rule out that some Pax2- cells replaced some Pax2+ cells?

Line 306 "conflictual" – I don't believe this is a word in English. Replace with "to conflict" or "in conflict".

Line 325 "endorses" – improper word usage. Maybe they mean "enhances"?

Line 344: "identify Pax2+ progenitors as the cellular source of a limited intrinsic regenerative capacity of kidney tubules than previously anticipated"
This sentence seems to be missing some words?

Reviewer #5 (Remarks to the Author):

As a new reviewer of this manuscript, I have considered the primary data, previous reviewers comments and the responses of the author available to me. It is my opinion that one of the previous reviewers has misunderstood aspects of the approach and the analysis being presented. The data is arguing that there is loss of epithelial cells in response to AKI, there is little evidence of genuine proliferation to recover from the injury state and that the majority of cells present, particularly in S1 and S2, undergo an alternate cell cycle progression referred to as endocycling where the nuclear content increases without mitosis or nuclear division. It is my opinion that the data in the manuscript does support these claims. A second reviewer has asked for a better description of how this paradigm shift can be explained based upon previously published data in two publications, including a fairly recent analysis from Kusaba et al. The response provided, which argues primarily a difference in the degree of injury and secondarily that the data are no inconsistent in the more mild IRI model, is reasonable and has not been incorporated into the manuscript.

Università degli Studi di Firenze

***CENTER OF EXCELLENCE FOR RESEARCH, TRANSFER AND HIGH EDUCATION
DENOTHE, UNIVERSITY OF FLORENCE, ITALY***

Point by point response to the referees' comments

Reviewer #4 (Remarks to the Author):

This paper argues that kidney tubular epithelial cell regeneration after injury is caused by the proliferation of a small subset of tubular cells, which are Pax2-positive and divide to form clones after injury, rather than by proliferation of most or all TECs. I believe that there are problems with the quantitative analysis of the number of clonogenic cells. The evidence that many TECs undergo endocycle rather than complete mitosis after injury is stronger. The paper is very difficult to follow in many places, and needs extensive editing.

Major Points

Figure 1 and associated text: I believe that the clone size analysis, which is used to calculate the percentage of “clonogenic cells” (Fig 1f), is flawed. They are counting colored cells in single sections, while many clones are expected to be oriented such that the section contains only a subset of cells in a clone; in some cases, only a single cell of a clone may be present in the section, so it is not counted as a multi-cell clone. This underestimates the numbers of multi-cell clones, which is used to calculate the number of clonogenic Pax8+ cells. I am aware that a similar method of clone-counting was used by Kusaba et al. (who cite an earlier paper by Driessens et al. to justify its use), but they did not use it for similar calculations. In the present paper, underestimating the number of Pax8-lineage clones will also lead to an underestimate of the number of clonogenic cells.

The same issue applies to Figure 3, where Pax2-lineage clones are analyzed.

To address the issue raised by the reviewer, we counted the number of clones in 3D images. To this aim, an image processing software from Leica Microsystems “Leica Application Suite X” was used. Z-serie stacks were obtained from 40 μm kidney slices stained with DAPI. Images were collected at 1 μm intervals. To compare 2D with 3D analysis, two blinded observers counted the number of clones in 2D images and in 3D reconstruction.

The 2D analysis did not underestimate the percentage of clonogenic cells. Rather, the 2D analysis slightly overestimated the number of clones, being unable to detect long clones that were instead revealed by 3D analysis (Fig. 4f-i and Supplementary video). However, the difference between the 2D analysis and 3D analysis was not statistically significant (Figure 4f and page 12 lines 300-303, page 13 lines 304-306).

We would like to underline that the concept that only few TECs divide, in our study is confirmed also by assessments in the FUCCI2aR model where we counted the absolute number of FUCCI2aR cells by unbiased Coulter counter analysis in the kidney using the MacsQuant (Fig. 7 k-n), with identical conclusions. Therefore, based on these multiple technical validations we can be absolutely sure that only a small subset of TECs divides after AKI and this is represented by Pax2-lineage cells.

Figure S2, c-e. If the mice “display consistent Pax2 expression” (line 106-107), why do the Pax2 antibody-stained cells (“white”) not correspond to the Pax2-lineage labeled cells (yellow, red, blue, green)? The results shown suggest that Pax2 expression is intermittent or changing over time. This seems to contradict the results of Figure 2 i-l, which show that the number of Pax2-lineage cells is stable over time.

In healthy mice Pax2-lineage labelled cells fully corresponded to Pax2-antibody-stained cells and this is stable over time. The images questioned by the reviewer were replaced by two others (see new images in Supplementary Figure 2d,e) better representing this concept that has already been addressed in previous publications (see Burger A. et al. Genesis 2011; Lasagni L et al. Stem Cell Reports 2015). However, as already discussed, (see page 6, lines 129-132), following AKI, Pax2 can be upregulated also by other TECs, and immunolabeling does not fully correspond to Pax2-lineage labelled cells anymore. Therefore, the Pax2 lineage tracing applied in our study is the only technical method to study the behaviour of Pax2-lineage labelled cells after AKI.

Following reviewer's suggestion, we also edited the manuscript to make it easier to read.

Minor Points

Line 33, 38, 323, 342. I object to the words “disproved” and “disprove”. It is fairer to say “clonal analysis argued against...” and “These results challenge current paradigms”.

Following the reviewer's suggestion, we replaced the words “disproved” with “challenge” in all the manuscript.

Line 75. “apparently reconstituted TECs and single coloured clones in outer stripe of the outer medulla (OSOM) (Fig. 1b,c).”

I don't know what they mean by “reconstituted”. They define “clones” as anything from one to 11+ cells, including single cells. Single colored “clones” don't mean anything other than that Pax8 was expressed in the cell at some earlier point – they don't show that anything was “reconstituted”. I think what they are trying to point out is the presence of multi-cell, single color clones, so they should be more specific.

We modified the sentence as suggested by the reviewer.

Figure 2 legend: “confetti reporter shows Pax2 expression” –

No, it shows cells (or the daughters of cells) that expressed Pax2-rtTA at any time in the past when doxycycline was present.

We changed the sentence in the legend of figure 2 to better specify that this is tracing and not expression.

Line 81 “Identical data” – change to “similar data”

We changed “identical” to “similar” as indicated.

Line 77, Line 84, line 101 – “irreversible”, “irreversibly lost” – they don't show that the loss of TECs is irreversible, only that it is not reversed within 30 days. They could say “a substantial and sustained TEC loss”.

We changed the sentence as indicated.

Line 90 “the percentage of single cell clones at T0 decreased from 92.4±0.9% to 78.9±0.9% at day 30 after IRI”.

Change to “the percentage of single cell clones decreased from 92.4±0.9% at T0 to 78.9±0.9% at day 30 after IRI”

We changed the sentence as indicated.

Figure 1d. They are not measuring “Pax8+ cells” as the axis labeling and legend stat, but rather Pax8-lineage-positive cells, i.e., cells that expressed Pax8.rtTA at the earlier time when Dox was administered; they may no longer express Pax8 at the time of imaging. To call them Pax8+ is confusing and should be corrected. I agree with the rebuttal, that they are not counting “tubular cells. vs. interstitial cells”, as Reviewer 1 stated.

We changed in each figure legend Pax8+ cells in Pax8-lineage positive cells as suggested.

Figure 1I – they don't define the meaning of the gray vs. white areas of the bar.

We added the explanation for gray and white areas of the bar.

Supplementary Figure 1.1 – there is no legend for panels l and m.

We added the legend for panels l and m to Supplementary Figure 1.1.

Line 111-114: “Immunolabelling with megalin, that stains proximal TECs of the S1 and S2 but not S3 segment, showed that Pax2+ cells represented 1-3% of TECs in S1 and S2 segments (Fig. 2h). Labelling with AQP1 showed that they represented 6-10% in S3 segment (Fig. 2a,e) and labelling with THP showed that they represented 8-12% of distal TECs (Fig. 2b,f).”

Are these numbers estimates, not actual counts? Why are they presented as ranges (1-3%, 6-10%, etc.) rather than actual averages +/- SD?

Ranges were replaced with averages +/- SD (page 5, lines 107-110).

Figure 3i,j,k,m,n,o. It is not obvious how these values were calculated – please explain in the figure legends.

We added a sentence in the Figure legend to directly recall the part of the Method section where calculations are reported (page 40, lines 919,920).

Figure 4f. The values of the 4 bars do not add up to 100%.

We apologise, there was a mistake in copying a number. The graph was corrected.

Line 154: “3D analysis revealed single coloured-Pax2+ clones up to 98 cells long regenerating entire tubule segments”.

How do they know that this is an entire tubule segment?

We replaced the word “entire” with “long” (page 7, line 150).

Line 161: “Treatment starting not before 24 hours after IRI”

This tells us when treatment did not start, but when did it start? At 24 hours after IRI? Please clarify.

Treatment started at 24 hours after IRI. This was clarified in the text (page 7, line 158).

Figure 5f. The ordinate is labeled (No of Px2+ cells included in clones”

Do they mean “No of PAX2+ cells per field included in clones”?

Yes, we mean “No of PAX2+ cells per field included in clones”, so this was corrected accordingly.

Again, these are Pax2-lineage+ cells, not Pax2+ cells.

We explained in each figure legend that “Pax2+ cells” means “Pax2-lineage+ cells” as suggested.

Figure 5. The legend for g-k says “arrows indicate single-color clones”. First, their definition of a clone is a group of 2 or more single-color cells in contact with each other, so all clones are “single-color”. But then, the two arrows in h point to tubules with cells of more than one color – so this is confusing.

We deleted the words “single colour” and pointed arrows in h to better representative clones.

Line 188 “the number per field was comparable to that of the Pax8/FUCCI mice (Fig. 6n vs j).”

What are the numbers per field?

Number per fields were added (page 8, lines 186-187).

Line 194-195. “a lower percentage of TECs 195 at day 2 was p-H3+ in comparison to Ki-67 (11.9±1.1% vs. 47.1±9.2%, p<0.05).”

Are these numbers plotted in any Figure?

No, they were only reported as numbers in the text (page 8, line 193).

Fig 6s. The legend on the ordinate “% of Pax8/FUCCI2 cells over p-H3+ FUCCI2 cells” does not match the legend, which says “Percentage of p-H3+mCherry+ cells and p-H3+mVenus+ cells in Pax8/FUCCI2 mice.....”

We homogenized the graph and the legend.

Line 214: “Rather, after IRI 35±4.8% of Pax2+ cells were in the S phase”

Change to “...2 days after IRI.....”

We changed the sentence as suggested (page 9, line 212).

Line 227. “Both transgenic lines showed a percentage of induction higher than 90%.”

Please define “percentage of induction”, and where is this data plotted?

The definition of “percentage of induction” is detailed in the method section (page 17 lines 409-411, and page 18 lines 441,442).

Line 298: “This was further underlined by their irrelevant expression of the TEC injury marker KIM-1”

What do they mean by “irrelevant” ?

We meant that Pax2+ cells were usually negative for KIM-1. This was better clarified (page 12, line 296).

Line 304-305 “Pax2+ cells can give rise to all the cells along the tubule, they are not replaced by unlabeled Pax2- cells. That is, Pax2+ cells self-renew, a distinctive property of progenitor cells”

I don't understand this argument – if some unlabeled Pax2- cells had replaced some “Pax2+”, they would not be seen as they are not labeled. How can they rule out that some Pax2- cells replaced some Pax2+ cells?

We changed the sentence to make it clearer (page 13, lines 304-306) .

Line 306 “conflictual” – I don't believe this is a word in English. Replace with “to conflict” or “in conflict”.

We replaced “conflictual” with “in conflict” (page 13, line 307).

Line 325 “endorses” – improper word usage. Maybe they mean “enhances”?

The word “endorses” was replaced with “enhances” (page 13, line 328).

Line 344: “identify Pax2+ progenitors as the cellular source of a limited intrinsic regenerative capacity of kidney tubules than previously anticipated”

This sentence seems to be missing some words?

Yes, we apologise, a word was missing and was now added (page 13, line 346).

Reviewer #5 (Remarks to the Author):

As a new reviewer of this manuscript, I have considered the primary data, previous reviewers comments and the responses of the author available to me. It is my opinion that one of the previous reviewers has

misunderstood aspects of the approach and the analysis being presented. The data is arguing that there is loss of epithelial cells in response to AKI, there is little evidence of genuine proliferation to recover from the injury state and that the majority of cells present, particularly in S1 and S2, undergo an alternate cell cycle progression referred to as endocycling where the nuclear content increases without mitosis or nuclear division. It is my opinion that the data in the manuscript does support these claims. A second reviewer has asked for a better description of how this paradigm shift can be explained based upon previously published data in two publications, including a fairly recent analysis from Kusaba et al. The response provided, which argues primarily a difference in the degree of injury and secondarily that the data are no inconsistent in the more mild IRI model, is reasonable and has not been incorporated into the manuscript.

We thank the reviewer for his comments and for the careful reading and appreciation of the data. We extended our discussion of the reasons for the possible discrepancies with the analysis of Kusaba et al., that are summarized in the scheme shown in Supplementary Fig 1.2 (page 13, lines 307-315).

REVIEWERS' COMMENTS:

Reviewer #4 (Remarks to the Author):

I am satisfied with the authors responses to my comments, and related changes to the manuscript.